# The role of large-scale dynamics in an exceptional sequence of severe thunderstorms in Europe May/June 2018

Susanna Mohr[1], Jannik Wilhelm[1], Jan Wandel[1], Michael Kunz[1,2], Raphael Portmann[3], Heinz Jürgen Punge[1], Manuel Schmidberger[1], Julian F. Quinting[1], and Christian M. Grams[1]

[1]Karlsruhe Institute of Technology (KIT), Institute of Meteorology and Climate Research (IMK), Karlsruhe, Germany
[2]Center for Disaster Management and Risk Reduction Technology (CEDIM), Karlsruhe, Germany
[3]Institute for Atmospheric and Climate Science, ETH Zurich, Switzerland

**Correspondence:** Susanna Mohr (mohr@kit.edu)

**Abstract.** Over three weeks in May and June 2018, an exceptionally large number of thunderstorms hit vast parts of western and central Europe, causing precipitation of up to 80 mm in one hour and several flash floods. During this time, the large-scale atmospheric circulation, which was characterized by a blocking situation over northern Europe, influenced atmospheric conditions relevant for thunderstorm development. Initially, the southwesterly flow on the western flank of the blocking anticyclone induced the advection of warm, moist, and unstably stratified air masses. Due to a low-pressure gradient associated with the blocking anticyclone, these air masses were trapped in western and central Europe, remained almost stationary and prevented a significant air mass exchange. In addition, the low-pressure gradient led predominantly to weak flow conditions in the mid-troposphere and thus to low vertical wind shear that prevented thunderstorms from developing into severe organized systems. Due to the weak propagation speed in combination with high rain rates, several thunderstorms were able to produce torrential heavy rain that affected local-scale areas and triggered several flash floods.

Atmospheric blocking also increased the upper-level cut-off low frequency on its upstream regions, which was up to 10 times higher than the climatological mean. Together with filaments of positive potential vorticity (PV), the cut-offs provided the meso-scale setting for the development of a large number of thunderstorms. During the 22-day study period, we found that more than 50 % of lightning strikes can be linked to a nearby cut-off low or PV filament. The exceptional persistence of low stability combined with weak wind speed in the mid-troposphere over three weeks has not been observed during the past 30 years.

**Keywords:** Europe, thunderstorms, severe convective storms, heavy rain, flash floods, atmospheric blocking, weather regimes, cut-off lows, potential vorticity

## 1 Introduction

Historically, the period from May to mid of June 2018 was among the most active periods of severe convective storms associated with heavy rain, hail, convective wind gusts and even tornadoes over large parts of western and central Europe (WetterOnline, 2018a, b, c; DWD, 2018a). More than 1,500 reports of hazardous weather events were documented by the European Severe Weather Database (ESWD; Dotzek et al., 2009). Rainfall totals of up to 90 mm within a few hours caused (pluvial) flash floods

in various municipalities. Gust speeds of up to $30\,\mathrm{m\,s^{-1}}$ led to numerous fallen trees and severely damaged buildings. For example, from 26 May to 1 June 2018, thunderstorms caused insured losses of about 300 million USD and overall losses of about 430 millions USD according to Munich Re's NatCatSERVICE (Munich Re, 2019). Thus, it was the costliest convective storm event in western Europe that year.

In general, the development of convective storms results from scale interactions of different processes in the atmosphere. It is well known that deep moist convection depends on three necessary but not sufficient ingredients (e.g., Johns and Doswell, 1992; Trapp, 2013): (i) any kind of instability (conditional, latent, potential) over a layer of sufficient depth and (ii) sufficient moisture in the lower troposphere. These requirements are usually controlled by processes on the synoptic scale. The third ingredient is (iii) a suitable lifting mechanism for the triggering of convection, which can occur at different scale ranges. For example, lifting mechanisms on the mesoscale include orographic lifting, horizontal convective rolls, or gravity waves (e.g., Wilson and Schreiber, 1986; Browning et al., 2007; Barthlott et al., 2010), whereas large-scale lifting can be related to drylines or cold fronts (e.g., Bennett et al., 2006; Kunz et al., 2020). A further relevant condition for the evolution of deep moist convection is the vertical wind shear, which is decisive not only for the organizational form, the longevity and thus the severity of the convective storms (e.g., Weisman and Klemp, 1982; Thompson et al., 2007; Dennis and Kumjian, 2017), but also for their propagation (Corfidi, 2003).

The general synoptic situation during the thunderstorm episode 2018 investigated in this study was similar to that prevailing over a 15-day period in May/June 2016, where also an exceptionally large number of thunderstorms caused several flash floods, primarily in Germany (Piper et al., 2016; Bronstert et al., 2018; Ozturk et al., 2018). During the episode in 2016, a blocking anticyclone over the North Sea and Scandinavian region prevented an exchange of the dominant unstably stratified air masses over several days. In addition, low wind speeds throughout the troposphere caused the thunderstorms to be almost stationary with the effect of torrential rain accumulations in several small regions (Piper et al., 2016, hereinafter referred to as PIP16).

Atmospheric blocking, with a typical lifetime of several days to weeks, is a quasi-stationary, persistent flow situation that modulates the large-scale extratropical circulation (Rex, 1950a, b; Barriopedro et al., 2006; Woollings et al., 2018). Such blocks typically occur either in a *dipole configuration* with an accompanying cut-off low on the equatorward side (Rex, 1950a; Tibaldi and Molteni, 1990) or they adopt an *omega-shape* with cut-off lows forming at the flanks of the blocked region (Dole and Gordon, 1983). In the potential vorticity (PV) framework, a cut-off low is an upper-level closed anomaly of stratospheric high PV air (e.g., Wernli and Sprenger, 2007; Nieto et al., 2007a, 2008). PV anomalies, in general, have a far-field impact on the meteorological conditions in their surroundings (cf. Hoskins et al., 1985). Below the positive PV anomaly, isentropes bend upward, resulting in reduced static stability and increased lifting. Due to an induced cyclonic circulation anomaly, the positive PV anomaly favours isentropic gliding up and thus ascent along the isentropes that usually bend upward towards the pole. Finally, when the positive PV anomaly propagates, air masses ascend isentropically at the PV anomalies' upstream side. These three mechanisms associated with lifting are intrinsic to upper-level positive PV anomalies in general. Additionally, at the flanks of a mature PV cut-off, small meso-scale filaments of positive PV often separate and are advected away, particularly when the PV cut-off gradually decays (Portmann et al., 2018). When such a positive PV filament moves over air masses that are conditionally or potentially unstably stratified, they trigger lifting and thereby release – if the air parcel reaches its level

of free convection – convective available potential energy (CAPE) and facilitate/cause deep moist convection (cf. Grams and Blumer, 2015). The effect of large-scale PV anomalies accompanied by cut-off lows on deep moist convection (in relation to severe precipitation events) has already been observed in other studies showing for Europe that this is an important mechanism for convection due to the associated patterns of advection and vertical motion (Roberts, 2000; Morcrette et al., 2007; Browning et al., 2007; Russell et al., 2012). But the effect is complex and not full understood.

At first, atmospheric blocking was primarily known for its conjunction to extreme weather events such as cold spells and heatwaves (and associated droughts; e.g., Pfahl and Wernli, 2012a; Bieli et al., 2015; Schaller et al., 2018; Röthlisberger and Martius, 2019). But in peripheral locations upstream and downstream of the blocks can also create environmental conditions conducive for deep moist convection development. Thus, the link to heavy precipitation events (including flood events) has already been intensively investigated in past years (e.g., Martius et al., 2013; Grams et al., 2014; Piaget et al., 2015; Sousa et al., 2017; Lenggenhager et al., 2018; Lenggenhager and Martius, 2019). A new study by Mohr et al. (2019) now shows a statistical relationship between convective activity (based on lightning data) and specific blocking situations in the European sector. They found a block over the Baltic Sea frequently associated with increased thunderstorm occurrences because of southwesterly advection of warm, moist and unstable air masses on its western flank. In addition, such situations are usually associated with weak wind speed at mid-tropospheric levels and thus weak vertical wind shear over the thunderstorm area with the consequence that thunderstorms become often stationary and rarely develop into large organized convective systems. Recently, Tarabukina et al. (2019) also demonstrate a correlation between the annual variation of summer lightning activity in Yakutia (Russia) and the frequency of atmospheric blocking in Western Siberia.

The primary objective of this paper is to examine the conditions and processes that made this particular thunderstorm episode in 2018 unique. We focus on the process interaction across scales, i.e., from the large-scale dynamics such as atmospheric blocking to meso-scale PV cut-off lows and/or small meso-scale PV filaments to modifications of the convective environment to local-scale thunderstorm occurrences. Further objectives are to highlight the synoptic setting during the thunderstorm episode, to demonstrate the severity of the events, and to place the event in a historical context.

The paper is structured as follows: Section 2 presents the different data sets and the methods used. Section 3 starts with a description of the thunderstorm episode in 2018 and their accompanying phenomena by investigating different observation data such as lightning information, hazardous storm reports, rain gauge measurements, and radar-based storm tracks estimating the propagation speed. Subsequently, the synoptic situation prior and during the examined thunderstorm episode is investigated by analyses of the large-scale flow situation, backward trajectories, accompanied weather regimes, and environmental conditions such as instability, moisture, or mid-tropospheric wind speed). Furthermore, we examine the role of PV cut-off and PV filaments on the development of deep moist convection. The next Section 4 puts the results in a historical context, whereby the exceptional nature of the thunderstorm episode is assessed by relating the observed rainfall totals, the prevailing environmental conditions, and the occurrence of cut-off systems to long-term data records. Finally, Section 5 and Section 6 discuss and summarize the main results and draw conclusions.

## 2 Data and methods

The study area includes parts of central and western Europe – France, Benelux (Belgium, Netherlands, Luxembourg), Germany, Switzerland and Austria (see Fig. 1). The study period extends over three weeks from 22 May to 12 June 2018, where most of the thunderstorms and secondary effects such as heavy rain, hail and convective wind gusts occurred (see Sect. 3). To highlight the synoptic situation prior to the episode and to emphasise that severe convection during the study period was embedded in a longer lasting unusual large-scale flow situation, we considered an extended study period from 1 May to 20 June 2018. For the purpose of climatological comparison, the 30-year period from 1981 to 2010 (1 May to 30 June) was the reference period (unless otherwise indicated).

### 2.1 Observation data

For the description of the thunderstorm episode in 2018, we use different observation data. Lightning data offer the best spatially homogeneous coverage for a full thunderstorm detection, but does not distinguish according to severity. For this purpose, we use eyewitness reports of the ESWD and precipitation observation (station-based and gridded-based). Radar-based storm tracks allow to investigate the propagation speed of the convective cells. Some investigation are limited to Germany, for which data were available (storm tracks, REGNIE), but enable a deeper insight into the exceptional nature of the phenomenon. Additionally, the atmospheric conditions are examined with data from various sounding stations. Some data are also available consistently and homogeneously over long-term periods, which allow us to compare the episode with historical conditions/events.

#### 2.1.1 Lightning data

Lightning data are obtained from the ground-based low-frequency lightning detection system of Siemens part of the EUCLID network (EUropean Cooperation for LIghtning Detection; Drüe et al., 2007; Schulz et al., 2016; Poelman et al., 2016). Available for the whole study domain, the data are projected on an equidistant grid of $10 \times 10 \, \text{km}^2$ and accumulated over 6-hour periods centered around the times in ERA-Interim (e.g., for the 06 UTC reanalysis the lightning period is $03 - 09$ UTC). This allows the data to be linked to the cut-off lows (see Sect. 2.5). We consider all types of flashes including cloud-to-ground, cloud-to-cloud, and intra-cloud flashes, whereas polarity or peak current are not investigated.

#### 2.1.2 ESWD reports

We use reports of heavy rain, hail (diameter $\geqslant 2 \, \text{cm}$), and convective wind gusts $\geqslant 25 \, \text{m} \, \text{s}^{-1}$ from the European Severe Weather Database (ESWD; Dotzek et al., 2009; Groenemeijer et al., 2017). The ESWD is a step-by-step quality controlled (four levels) database providing detailed information about severe convective storms in Europe, mainly based on reports from storm chasers, eyewitnesses, voluntary observers, meteorological services, and news media. We consider all records with a quality level above QC0+. Using a homogeneous data format, these observations contain information about hazardous weather events such as

location, time, intensity, and damage-related information. For a detailed description of the event reporting criteria see ESSL
(2014).

### 2.1.3 Rainfall totals

Daily rainfall totals of 232 stations distributed across the domain ($41°N – 58°N\ 4°W – 20°E$) were collected from the European
Climate Assessment and Dataset (ECA&D), a database of daily meteorological station observations across Europe (Klein Tank
et al., 2002). In addition, hourly and daily data were obtained from the Météo-France (1223/1935 stations with hourly/daily
data), the Royal Netherlands Meteorological Institute (KNMI; 50/322), the German Weather Service (DWD; 958/810), Me-
teoSwiss (952/0), and the Central Institution for Meteorology and Geodynamics (ZMAG; 254/0). For statistics of hourly and
3-hour extreme rainfall events, we applied the same severity thresholds used in the ESWD (ESSL, 2014), which amount to 35
and 60 mm, respectively (Wussow, 1922; Nachtnebel, 2003). Note that the 24-hour criterion of 170 mm was not measured at
any of the stations.
Statistical return periods of single heavy precipitation events are estimated using regionalized precipitation data (*REGion-*
*alisierte NIEderschläge*, REGNIE) provided by DWD (DWD, 2018b). REGNIE is a gridded data set of 24-hour totals (from
06 UTC to 06 UTC on the next day) based on approximately 2,000 climate stations more or less evenly distributed across
Germany (the so-called RR collective). The REGNIE algorithm interpolates the measurement data to a regular grid of $1\,km^2$
considering altitude, exposure, and climatology (Rauthe et al., 2013). The data covering only Germany are available since
1951. The long-term availability of REGNIE over almost 70 years is the decisive advantage compared to other data sets such
as RADOLAN (merger between radar and station data; DWD, 2019), which have a higher spatial and temporal resolution but
are only available for 20 years. Note that the REGNIE time series are affected by temporal changes in the number of rain
gauges considered by the regionalization (Rauthe et al., 2013). For our purpose, the homogeneity of the data are sufficient.
Statistical return periods of REGNIE totals are quantified using the Generalized Extreme Value (GEV) distribution (e.g.,
van den Besselaar et al., 2013; Ehmele and Kunz, 2019). The Fisher-Tippett Type I distribution, also known as the Gumbel
distribution (Gumbel, 1958; Wilks, 2006), has been extensively used in various fields including hydrology for modelling
extreme events, i.e. to estimate statistical return periods or return values (Sivapalan and Blöschl, 1998; Rasmussen and Gautam,
2003). The Gumbel cumulative distribution function (CDF) for the precipitation totals $R$ is given by:
$$F(R) = \exp\left[-\exp\left(\frac{\zeta - R}{\beta}\right)\right], \tag{1}$$
with $\zeta$ and $\beta$ as location and scale parameters. For their estimation, we use the Method of Moments (Wilks, 2006, Chap. 4) and
considered the 67-year period from 1951 to 2017 (summer half-year from April to September):
$$\beta = \frac{\sigma\sqrt{6}}{\pi} \qquad \& \qquad \zeta = \bar{R} - \delta \cdot \beta, \tag{2}$$
with $\sigma$ as the standard derivation, $\bar{R}$ as the mean of the REGNIE sample and $\delta$ as the Euler-Mascheroni constant ($\approx .0.5772$).
The return period $t_{RP}$ is directly related to the probability of occurrence of the threshold $P(R \geqslant R_{trs}) = t_{RP}^{-1}$ so that the CDF
is given by $F(R) = 1 - t_{RP}^{-1}$. The resulting equation to estimate the return period $t_{RP}$ is:
$$t_{RP}(R) = \left[ 1 - \exp\left( -\exp\left( \frac{\zeta - R}{\beta} \right) \right) \right]^{-1}.$$  (3)
**2.1.4    Storm tracks computed from radar reflectivity**
Storm motion vectors are computed from three-dimensional (3D) radar reflectivity data from the radar network of DWD.
The data, which includes 17 radar stations with dual-polarization Doppler radars, are combined and interpolated into a radar
composite with a spatial resolution of $1 \times 1 \, \mathrm{km}^2$ (Cartesian grid). The temporal resolution of the individual scans is 15 minutes.
Radar reflectivity is available on 12 equidistant vertical levels with a distance of 1 km (lowest level is 1 km above ground). For
the whole period between 2005 and 2018, which is used to relate the storm motions computed for the investigation period to
the climatology (Sect. 4.1), data were stored in six reflectivity classes only. The two highest classes, which are considered here,
range from 46 to 55 dBZ and $\geqslant 55$ dBZ.

To identify storm tracks, the cell-tracking algorithm TRACE3D (Handwerker, 2002) was adapted to the DWD radar com-

posite in Cartesian coordinates. Once the algorithm detects a convective cell core, it can be re-detected in the consecutive time
steps and merged into an entire cell track. Storms are defined by having a minimum reflectivity core of 55 dBZ (corresponding
to the highest class) and a vertical extent of at least 1 km. Thus, only severe convective storms frequently associated with
hazardous weather are considered. Thunderstorms above the 55 dBZ threshold usually form a well-defined core of high reflec-
tivity that can be easily and reliably tracked. Based on TRACE3D, information about width, length, duration, and propagation
speed, as well as direction, is available for each individual thunderstorm track. Note that we mainly use tracking to estimate
the propagation speed and direction of the cells (Sect. 3 and Sect. 4.1). Even if weaker cells are not detected using the 55 dBZ
thresholds, it can be assumed that they cannot move at higher speeds. More details about data and the tracking method can be
found in Puskeiler et al. (2016) and Schmidberger (2018). Due to a lack of 3D radar data for France in 2018, our investigation
refers only to severe convective storms that occurred in Germany.
**2.1.5    Sounding stations**
Atmospheric conditions are estimated from vertical profiles of temperature, moisture, wind speed and direction at seven sound-
ing stations provided by DWD and the Integrated Global Radiosonde Archive (IGRA) from the National Climatic Data Center
(Durre et al., 2006). These stations are distributed over the entire domain: Bordeaux (44.83°N 0.68°W) and Trappes (48.77°N
2.00°E) in France; Essen (51.41°N 6.97°E), Stuttgart (48.83°N 9.20°E), and Munich (48.24°N 11.55°E) in Germany; Payerne
(46.82°N 6.95°E) in Switzerland, and Vienna (48.23°N 16.37°E) in Austria (see Fig. 1). Other sounding stations could not be
used because of multiple gaps in the time series.

Atmospheric stability can be estimated, for example, by CAPE as well as by the surface-based Lifted Index (SLI; Galway,

1956). The latter, which we use in the following, has proven to be as suitable parameter as CAPE for estimating instability
in several studies (e.g., Huntrieser et al., 1997; Sánchez et al., 2009; Westermayer et al., 2017; Rädler et al., 2018). There are
studies, in which SLI has even shown a better prediction skill than CAPE (e.g., Haklander and van Delden, 2003; Manzato,
2003; Kunz, 2007; Mohr and Kunz, 2013). In addition to the SLI, we also investigate the horizontal wind speed in 500 hPa
(V500). Both variables are analysed at 12 UTC, since thunderstorms in central Europe usually peak during the late afternoon
(Wapler, 2013; Piper and Kunz, 2017; Enno et al., 2020).

## 2.2 Model data

We use the European Centre for Medium-Range Weather Forecasts (ECMWF) high-resolution operational analysis data and
ECMWF ERA-Interim reanalysis (Dee et al., 2011) to describe the large-scale meteorological conditions and to calculate
weather regimes (see Sect. 2.3), kinematic backward trajectories (see Sect. 2.4), and cut-off lows (see Sect. 2.5). ECMWF anal-
ysis is available 6-hourly interpolated to a regular grid with $0.125°$ horizontal resolution. ERA-Interim used for the historical
analysis is available 6-hourly interpolated to a regular grid at $1.0°$ horizontal resolution. Beside the atmospheric stability (based
on SLI), we examine in the study V500, the bulk wind shear (BWS; directional shear) as wind difference between 10 m and
500 hPa calculated by vector subtraction (e.g., Thompson et al., 2007), 500 hPa geopotential height (Z500) and the vertically
integrated water vapor (IWV).

## 2.3 North Atlantic-European weather regimes

The large-scale flow conditions in the Atlantic-European region are characterized in terms of a definition of seven year-round
weather regimes based on 10-day low-pass-filtered 500 hPa geopotential height anomalies (Z500'; Grams et al., 2017). The
regimes are identified by k-means clustering in the phase-space spanned by the seven leading empirical orthogonal functions
(EOFs). Based on these seven clusters, an active weather regime life-cycle is derived from the normalized projection of each
6-hourly anomaly in the cluster mean following Michel and Rivière (2011). Thereby, time steps with weak projections are
filtered out (no regime). An active regime life-cycle persists for at least 5 days but fulfills further criteria as described in Grams
et al. (2017).
Our weather regime definition is in line with 'classical' concepts of four seasonal regimes for Europe (e.g. Vautard, 1990;
Michelangeli et al., 1995; Ferranti et al., 2015), but reflects important seasonal differences. Three of the seven regimes are
dominated by a negative Z500' and enhanced cyclonic activity (see Supplementary Fig. 1). These are the *Atlantic Trough (AT)*
regime with a trough extending towards western Europe, the *Zonal regime (ZO)* with cyclonic activity around Iceland, and the
*Scandinavian Trough (ScTr)* regime with a trough shifted towards the east. The remaining four regimes are characterized by a
positive Z500' centered at different locations and therefore referred to as 'blocked regimes'. These are the *Atlantic Ridge (AR)*
regime, with a blocking ridge over the eastern North Atlantic and an accompanying trough extending from eastern Europe into
the central Mediterranean, the *European Blocking (EuBL)* regime, with a blocking anticyclone extending from Western Europe
to the North Sea, *Scandinavian Blocking (ScBL)*, with high-latitude blocking over Scandinavia, and *Greenland Blocking (GL)*
with a blocking ridge over the Greenland-Icelandic region.

## 2.4 Lagrangian Analysis Tool

The path of the air masses during the thunderstorm period from 22 May to 12 June is traced by calculating 10-day kinematic backward trajectories from ERA-Interim using the Lagrangian Analysis Tool (LAGRANTO, Wernli and Davies, 1997; Sprenger and Wernli, 2015). The trajectories are initialised 6-hourly on each day of the study period from the five ERA-Interim grid points surrounding the seven sounding stations (Fig. 1 yellow squares). In order to represent the Lagrangian history of moist, low-tropospheric air masses that contributed to the severe thunderstorms, trajectories are initialised every 50 hPa between 950 and 600 hPa, where the air masses relevant for the thunderstorm development are located.

## 2.5 Identification of PV cut-off low and matching with lightning data

We identify upper-level cut-off lows based on PV on the 325 K isentropic surface from ERA-Interim using the algorithm of Wernli and Sprenger (2007) and Sprenger et al. (2017). The optimal level for the inspection of weather systems on isentropic surfaces depends on the season. The specific level of 325 K used here is motivated by the literature (cf. Röthlisberger et al., 2018) and the inspection of isentropic PV charts for our case. The algorithm searches for closed areas of PV larger than 2 PVU, which are disconnected from the main PV reservoir that expands across the North Pole.

Following earlier approaches to match weather objects with surface weather (e.g., cyclones and precipitation; Pfahl and Wernli, 2012a, b), the identified PV cut-off lows (including their PV filaments) are then related to thunderstorm events using lightning data on the $10 \times 10\,\text{km}^2$ grid cells. We utilize the smallest distance approach to link a grid cell in the lightning data set to a grid point in the PV cut-off data set. The different grid sizes between the model and observation data sets require matching multiple grid cells (lightning data) to one PV cut-off grid point. This means if a grid point shows the presence of a PV cut-off, all flashes from the associated grid cells are linked to it.

To account for the far-field impact of lifting and destabilization by a PV cut-off, we expand the PV cut-off mask by a buffer. This scale is estimated from the typical Rossby radius of deformation

$$L_R = \frac{N \cdot H}{f_0} \tag{4}$$

associated with a PV cut-off. Here, $N$ is the Brunt-Väisälä frequency, $H$ is the scale height, and $f_0$ is the Coriolis parameter. For characteristic values in mid-latitudes with $N = 0.01\,\text{s}^{-1}$ and $f_0 = 10^{-4}\,\text{s}^{-1}$, $N/f_0$ is typically in the order of 100. A scale height of 10 km leads to a Rossby deformation radius of 1,000 km, which is typical for synoptic scales. We assume that some of the PV cut-offs during the study period have a vertical extent of less than 10 km. Therefore, we chose a conservative deformation radius (buffer) of about 500 km. The robustness of the chosen deformation radius is investigated both qualitatively and quantitatively. We found that a change in the radius of 100 km, for example, leads to an increase or decrease of around 10 % in the total amount of lightning strikes associated with a PV cut-off during our study period (see Supplement Sect. 2). Such small changes do not affect the qualitative interpretation of our results.

## 2.6 Persistence analysis

Days with constant atmospheric conditions tend to form temporal clusters of certain weather events (here thunderstorms) with a lifetime of several days. This behavior can be described statistically by the concept of persistence. The cluster length or event persistence $n$ of a specified event is defined as the sequence of days (between 1 and $x$ days) with the binary parameter with values of 1 (event day = criterion fulfilled) or zero (non-event day = criterion not fulfilled). Within a cluster of seven event days, we allow one day to be a non-event one (skip day), which is not considered in the total length $n$. For example, clusters with a length of up to 7 (14/21) days may contain at most 1 skip day (2/3 skip days). For more information on the concept see PIP16.

In the study, we investigate the co-occurrence of low stability (using SLI) and low mid-tropospheric wind speeds (using V500). For this purpose, the same thresholds as in PIP16 are chosen, which were used to investigate the exceptional atmospheric conditions of a similar thunderstorm episode. We employ the basic criterion, which is fulfilled if both conditions apply: $SLI < 0\,K$ and $V500 < 10\,m\,s^{-1}$ ($TH_{BC}$). In addition, we also discuss our results in context with the strict criterion, which is fulfilled with $SLI < -1.3\,K$ and $v_{500hPa} < 8\,m\,s^{-1}$ ($TH_{SC}$). Both thresholds were originally determined by choosing the maximum of the daily minima in the case study to capture the prevailing (exceptional) atmospheric conditions.

## 3 Description of the thunderstorm episode 2018

The period from the first of May to mid-June 2018 was characterized by a large number of thunderstorms that spread across the study area, several of which were associated with heavy rainfall, hail, and strong wind gusts (Fig. 2a). More than 1,500 severe weather reports were collected and archived by the ESWD in our study area during that period. Lightning strikes were recorded on each day, and the affected area ranges between $100\,km^2$ on 19 June and $1,140,000\,km^2$ on 27 May (accumulations of the $10 \times 10\,km^2$ grids).

The three-week period from 22 May until 12 June was the most active thunderstorm episode in May/June 2018 with a total of 868 heavy rain, 144 hail, and 145 convective wind gust reports based on the ESWD. The highest number (152 reports) was issued on 29 May, followed by 31 May (137 reports), most of them reporting heavy rainfall leading to a couple of flash floods and landslides, which destroyed buildings, vehicles, streets and even railway tracks (DWD, 2018a; WetterOnline, 2018b). On average, an area of $758,000\,km^2$ – an area twice the size of Germany – was affected by lightning per day, with the result that thunderstorms covered the entire study area. As shown in Figure 2b, most of the severe weather reports during the episode came from the western part of France, Benelux, central and southern Germany, and the easternmost part of Austria. While the spatial distribution of the ESWD reports shows several regional gaps due to an under-representation of eyewitness reports, for example, in Central and southeastern France (cf. Groenemeijer et al., 2017; Kunz et al., 2020), thunderstorm days are observed throughout the study area (see Supplementary Fig. 4). The extraordinarily large number of thunderstorms, several of them severe, and the unusual persistence of that situation over three weeks motivated us to select that time frame as the study period.

Figure 3 summarizes the evaluation of hourly (1 h) and 3-hour (3 h) rain gauge measurements in the study area exceeding the ESWD heavy rain criteria of 35 mm and 60 mm, respectively. The 1 h criterion was fulfilled during the study period 167

times (Fig. 3a) and an average of about 7.6 stations per day with a variability between one and 20 stations. This highest number of stations belongs to the day with the second most ESWD severe weather reports (31 May). The 3 h criterion was reached 38 times, with a maximum of at least 5 stations on three days. The location of the respective stations shows heavy rain events in all of the countries under consideration without any clustering (Fig. 3b,c).

During the episode, the thunderstorms developed mainly as isolated cells and clusters of several cells, the latter preferably in the early evening and night. Only on a few days (e.g., on 22 May or 1 June) larger mesoscale convective systems (MCS) formed, which persisted during the night and early morning. Animated images of radar reflectivity can be found in the Video Supplement for two representative days: 27 and 31 May. The two animations show a large number of both isolated thunderstorms with a short lifetime of approximately 30 min (radar visibility) and cell clusters persisting over several hours. Most cells moved very slowly or even remained stationary during the two days.

The slow movement of the convective cells, a prominent feature of the entire thunderstorm episode, was mainly due to the low wind speed at mid-tropospheric levels (cf. Sect. 3.1.2). According to the cell tracking (Germany only; see Sect. 2.1.4), about half of all cells reaching a radar reflectivity of at least 55 dBZ had a propagation speed of less than $5\,\mathrm{m\,s^{-1}}$ (47.3 % from 480 cells); only a few cells (1.5 %) had a speed above $15\,\mathrm{m\,s^{-1}}$ (Fig. 4). Mean (standard deviation) and median values are $5.9\,\mathrm{m\,s^{-1}}$ ($\pm 2.9\,\mathrm{m\,s^{-1}}$) and $5.2\,\mathrm{m\,s^{-1}}$, respectively, which is almost half of the long-term values (cf. Sect. 4.1). The predominant propagation direction was from southeast to northwest (26.3 % of the detected cells). However, several cells moved in completely different directions on the same day – a clear sign that the propagation was not only determined by the (weak) mid-tropospheric wind, but also by internal dynamical effects induced by cold pools or by pressure disturbances (Markowski and Richardson, 2010; Houston and Wilhelmson, 2012). Examples of different track directions of neighbouring cells can be seen in the radar animation on 27 May (14 to 15 UTC, at the coordinates: $x \sim 250\,\mathrm{km}$ & $y \sim 600\,\mathrm{km}$) or on 31 May (21 to 22 UTC; $x \sim 400\,\mathrm{km}$ & $y \sim 700\,\mathrm{km}$).

A detailed look at the chronological sequence during the episode (Fig. 2b) shows that thunderstorms associated with heavy rainfall and small hail with diameters of around 2 cm were restricted to Benelux and western Germany on 22 May. Some entries report on flash floods and mudslides, for example in the Heilbronn area (SW Germany). Two days later, on 24 May, the federal state of Saxony (east Germany), the east of Austria, and parts of Belgium were hit by torrential rain accumulations. The German station Bad Elster-Sohl in Saxony (see Fig. 1) on the border to the Czech Republic, for example, measured a record of 86.3 mm / 3 h and 154.9 mm / 24 h. On 26 May, several wind reports with gust speeds of up to $29\,\mathrm{m\,s^{-1}}$ (Poitiers, France; see Fig. 1) and hail reports indicating hailstones with a diameter of up to 5 cm were recorded in the French coastal region of the Bay of Biscay.

The subsequent time frame from 27 May to 1 June was the most active both in terms of the area affected by lightning and the number of ESWD reports (Fig. 2a). Widespread thunderstorms were observed mainly in Benelux, Germany, and France, but also sporadically in Switzerland and Austria, many of them associated with large rain accumulations and subsequent flooding, hail between 2 and 4 cm in diameter, and damaging wind reports. Many of record-breaking 1 h and 3 h rainfall totals occurred within this period (Table 1). For example, the weather station Bruchweiler (see Fig. 1), located in the west of Germany, measured a 24 h rain accumulation of 145.0 mm on 27 May (Note that the station only provides reports for the full 24 hours).

However, this rain amount was fallen within 3 h, while a rain rate of more than 60 mm was observed within 50 min alone (see
also Supplementary Fig. 5a). The corresponding track, derived from TRACE3D, has a length of 21 km and a propagation speed
of $5.7\,\mathrm{m\,s^{-1}}$ (Table 1). A second example is on 31 May the conspicuously high 1 h rain accumulation of 85.7 mm measured at
Dietenhofen close to Nuremberg in the south of Germany (see also Fig. 3b), listed high in the ranking of highest 3 h rainfall
totals as well. The station was fully hit by an isolated system, which was relatively stationary. The rain rate above 60 mm was
present over 35 min (see also Supplementary Fig. 5b and Video Supplement).
In the first half of June, some hail stones and heavy rainfall were still reported almost daily somewhere in the study domain,
though less frequently than before. Towards the end of the study period, convective activity increased again. Especially on the
last day of the study period, on 12 June, the proportion of gust reports indicating wind speeds between 25 and $31\,\mathrm{m\,s^{-1}}$ to
all reports was very large. After the convectively most active period, when environmental conditions became more stable (cf.
Sect. 3.1), thunderstorms rarely occurred. The area affected by lightning decreased considerably and no further severe weather
reports were archived in the ESWD.
As we will show later (Sect. 3.1.2), the wind shear values over the study area were predominantly very low. Individual cases
with hail stones of 5 cm were feasible, because close to the border of our study area over the Pyrenees to the east two times
regions with high shear (up to $20\,\mathrm{m\,s^{-1}}$) were advected, which led to the large hail in southwest France (26 May/9 June) or
southern Germany (11 June). However, these were exceptional cases.

## 3.1   Synoptic overview

The synoptic situation prior to the thunderstorm episode in 2018 was embedded in a longer lasting unusual large-scale flow
situation. At the beginning of the extended study period, a large-scale mid-tropospheric area of high geopotential stretched out
from the Azores over central Europe and the Baltic to western Russia (Fig. 5a), attended by a corresponding prolonged lower-
level high-pressure system (not shown). This configuration was associated with the advection of warm and relatively dry air
masses over large parts of Europe. In the second week of May, the pattern transitioned into a blocked situation over Europe (see
Sect. 3.1.1). The geopotential height at 500 hPa depicts the typical *Omega*-like structure with high geopotential over central
Scandinavia, flanked by one pronounced trough upstream over the Northern Atlantic and one downstream over Western Russia
(Fig. 5b). Subsequently, the two troughs turned into enclosed cut-off lows filled with relatively cold air and finally merged
into one system located over central Europe on 15 May (not shown). In the third week of May, the cut-off moved slowly
northeastward on an erratic track while gradually dissipating over central and eastern Europe, leaving a moderately warm and
dry air mass with weak gradients over central Europe (Fig. 5c).
The study period from 22 May to 12 June was characterized by a rather stationary and persistent synoptic situation with a
pronounced blocking ridge stretching from Iceland over the North Sea to Scandinavia and Northeast Europe (Fig. 6a). As a
consequence of the synoptic setting during this episode, the mid-tropospheric flow was weak over most parts of Europe (see
Sect. 3.1.2). On average, the ridge was flanked by long-wave troughs: one on the western side with the axis pointing from Baffin
Bay to Newfoundland, the other on the eastern side stretching from the Barents Sea to Kazakhstan, while the ridge remained
relatively stationary centered over the North Sea region (Fig. 5c-f).

A noticeable feature in the mean 500 hPa geopotential height for this episode is a locally enclosed geopotential minimum over the Bay of Biscay and its surroundings (Fig. 6a) that emerges from repeating/transient cut-off lows forming on the upstream side of the blocking ridge. On 25 May (Fig. 5d), a cut-off low (C1a) approached Iberia – which merged in the next days with the cut-off located over the Celtic Sea (C1b) – and triggered several storms, first in France and then in Benelux and Germany (cf. Fig. 2). In the following days, a new cut off (C2; not shown) formed west of Spain, which subsequently influenced the weather there and disappeared relatively quickly. On 1 June, another cut-off (C3) advanced from the Atlantic (Fig. 5e), with some impact on convective activity over France, and then developed into a shallow low-pressure zone in central Europe. Several convergence lines were formed in that zone. In addition, this situation provided very moist air (IWV well above $30 \, \mathrm{kg \, m^{-2}}$ over large areas) until 9 June in eastern France and central Europe (Fig. 5e,f). In the end phase of the study period, the next cut-off low (C5) with its associated fronts and convergence lines affected the western half of France and central and southern Germany and lasted until 12 June (Fig. 5f). Simultaneously, a cut-off (C6) over the British Isles influenced the weather in northern Europe.

The geopotential anomalies at the 500 hPa level, calculated as the deviation from the climatological mean (1981 – 2010), exhibit for the study period significant positive values of up to 200 gpm west of Norway (Fig. 6). In contrast, the area over southwestern Europe is reflected by negative geopotential anomalies of more than 50 gpm. Qualitatively similar anomaly patterns are seen in the sea-level pressure distribution (not shown). Simultaneously, the IWV (Fig. 6b) showed distinct positive anomalies of up to $9 \, \mathrm{kg \, m^{-2}}$ with a 22-day average of $24 - 28 \, \mathrm{kg \, m^{-2}}$. This finding is in line with the sequential progression of several cut-off lows approaching southwestern Europe and leading to repeating the advection of warm and moist air masses towards central and western Europe during the study period.

### 3.1.1 North Atlantic-European Weather Regimes

In terms of the North Atlantic-European weather regimes, the large-scale flow situation in May was dominated by simultaneously active life cycles of a Zonal regime (ZO; dark red in Fig. 7a) and European Blocking (EuBL; green). Climatologically, the Zonal regime is characterized by a negative 500 hPa geopotential height anomaly centered over southern Greenland and Iceland, accompanied by a weak positive anomaly over central Europe (cf. Supplementary Fig. 1). The climatological European Blocking regime is characterized by a strong positive geopotential height anomaly over the North Sea region, and a weak negative anomaly over Baffin Bay.

The strong projection in both regimes in May suggests that both the cyclonic anomaly in the Icelandic region and the positive anticyclonic anomaly over Europe were pronounced but altered in their intensities – as discussed in the previous section. The alternating dominance of either regime in the first three weeks of May (Fig. 7a) reflects the change of zonal to meridional circulation and the persistent blocking situation during our study period. It is striking that enhanced convection and thunderstorm activity over Europe co-occurred with a weakening of the projection in the Zonal regime (see Section before). Specifically, the first period of widespread thunderstorms (9 – 16 May; cf. Fig. 2) coincides with a weakening of zonal conditions and a dominance of European Blocking from 11 to 18 May. This is interrupted by more zonal conditions from 19 to 21 May, leading to a substantial weakening of convective activity. The convectively most active period from 26 May to 1 June co-occurs with a very

strong projection into European Blocking and ends when the blocking decays. On 3 June, a transition into the Atlantic Ridge
regime occurs, with blocking shifting into the Northeast Atlantic and western Europe, which coincides with the last episode of
an increased number of convective events from 6 to 12 June.

### 3.1.2 Local-scale environmental conditions

During the entire May/June period, atmospheric stability was very low over large parts of the study domain as indicated by
sounding data (Fig. 7b). The SLI values reached negative values almost every day at 12 UTC at one sounding station at least.
During the first thunderstorm episode from 9 to 16 May with several heavy rain and hail events (cf. Fig. 2), several stations
already show negative SLI values at some days. During the study period, all soundings (with a few exceptions) exhibit per-
manently negative SLI values; most of the time the values are far below the basic/strict criterion of PIP16 (cf. Sect. 2.6). For
example, the median of the SLI during the study period was lower than –3.0 K for Stuttgart, Munich, Vienna, Trappes, and
Payerne. Such low values represent very conducive conditions for thunderstorm formation (e.g., Haklander and van Delden,
2003; Manzato, 2003; Sánchez et al., 2009; Kunz, 2007; Mohr and Kunz, 2013). In the ECMWF analysis (Fig. 8a), the SLI
average over the study period (12 UTC) was negative for most parts of the domain except for northern Germany, where thun-
derstorms occurred infrequently. Furthermore, over large parts of the study domain, the strict criterion was also reached. Due
to the upcoming westerly flow at the end of the study period, instability decreased significantly and SLI returned to positive
values less conducive for deep moist convection (Fig. 7b).
Due to the low-pressure gradient that prevailed during the study period (Fig. 6), horizontal wind speed in the mid-troposphere
was likewise exceptionally low. During the first half of May, 500 hPa wind speed (V500) was already low in the sounding
data with values rarely exceeding $15 \, \mathrm{m \, s^{-1}}$ (Fig. 7c), but further dropped significantly at the beginning of the study period.
Averaged over the entire study period, median V500 was $7 \, \mathrm{m \, s^{-1}}$ at the Essen sounding station; at Stuttgart, Munich, and
Vienna values were even lower at around $5 \, \mathrm{m \, s^{-1}}$. At the other three stations in France and Switzerland, the median was
between 8 and $10 \, \mathrm{m \, s^{-1}}$. The observations are in line with ECMWF analysis, where V500 was between 5 and $10 \, \mathrm{m \, s^{-1}}$ on
average (particularly low in large parts of Germany and Austria; Fig. 8b).
Due to the very low wind speed near the surface, V500 is almost similar to BWS from ECMWF analysis (12 UTC; Fig. 8c).
Mean values of BWS between 5 and $10 \, \mathrm{m \, s^{-1}}$ across the study area (except of the Pyrenees region) are a strong indication that
the majority of storms did not developed into highly organized convective systems such as squall lines, MCS or supercells. The
following analyses are relying on V500 instead of BWS, especially because of the very unusually low wind speed at 500 hPa.
It should be noted that the values for the deep layer shear (speed shear) are even lower compared to BWS ($3-9 \, \mathrm{m \, s^{-1}}$; not
shown).

### 3.2 Air mass origin and paths during the event

The investigation of sounding data revealed an exceptional air mass, which conserved its key properties conducive to convection
in the entire study period. This finding together with the low-pressure gradient associated with the blocking anticyclone over
the European sector (Fig. 6) suggests that the air mass was relatively stationary in western and central Europe during the study
period. To test this hypothesis, 10-day kinematic backward trajectories are calculated to investigate the Lagrangian history
and paths of moist, lower-tropospheric air masses. Though backward trajectories are started from all six sounding stations,
Bordeaux, Stuttgart, and Vienna are chosen as representative locations for the following analysis.
The median trajectory pathways during the entire study period 22 May to 12 June consistently show that air masses originated
west of the sounding stations and reached those in a southwesterly flow (Fig. 9a). Already ten days prior to reaching the area
of the sounding stations, two thirds of the air masses were located over the Atlantic-European sector. Though about 50 %
of the air masses were transported over a distance of 5,000 km (Fig. 9b), the median distance from their initial location (i.e.,
Bordeaux, Stuttgart, Vienna) never exceeded more than 2,000 km (Fig. 9c). This clearly indicates that air masses re-circulated
while approaching the area of the sounding stations. Five days prior to arriving at the location, trajectories were mostly located
over Europe (bold ellipses) and within a radius of 1,000 km around the sounding stations.

## 3.3 Thunderstorms related to cut-off lows

After having shown that a quasi-stationary air mass, which was conducive to convection prevailed over vast parts of central
Europe during the study period, we now explore cut-off low activity as potential trigger for thunderstorms. The blocking
situation over central Europe and the North Sea during the study period was accompanied by a negative geopotential height
anomaly over the Iberian Peninsula (Fig. 6), which corresponds well with a significantly enhanced frequency of PV cut-offs of
more than 50 % in the Bay of Biscay region (Fig. 10). This region of enhanced PV cut-off frequencies expands over much of
Spain, western France and some parts of the British Isles with frequencies often above 25 %, but does not reach Germany or
eastern Europe. The fact that relatively high PV cut-off frequencies expand over a larger region of western Europe (Fig. 10)
underlines that multiple individual PV cut-offs form on the upstream flank of the blocking ridge (see Fig. 5), and intermittently
move across Iberia, France, the British Isles, the North Sea, and Germany.
In such a configuration, filaments of positive PV that separate from the main PV cut-off may favour lifting on their down-
stream flank and help to trigger deep moist convection over larger areas. This relation is exemplified by a 2-day period from
31 May to 1 June representing the end of the period with the most lightning activity and ESWD reports. Here, more than
700,000 lightning strikes were measured over the study domain (black bars in Fig. 11) and more than 70 % of these can be
attributed to PV cut-off activity (light grey bars). On 31 May, in the early afternoon, thunderstorms primarily affected Bel-
gium and the Netherlands first (Fig. 12a), before lightning activity re-emerged over central and northern France, Switzerland,
and various parts of Germany (Fig. 12b). Several of these events were documented by heavy rain reports in the ESWD (cf.
Fig. 2). During the following night, the slow-moving multicellular system moved from Switzerland northwards affecting the
southwestern and the western parts of Germany (Fig. 12c,d; cf. Sect. 3). While the system dissipated in the late morning over
the border region of Germany and Belgium, severe thunderstorms developed again over eastern and northern Germany, Czech
Republic, western Poland, and the Pyrenees (Spain; Fig. 12e,f). The link to upper-level PV filaments becomes apparent by
carefully investigating the 6-hour evolution of the identified cut-off low masks (Fig. 12; cf. Sect. 2.5). Additionally, the area of
negative $\omega$ values indicates upward vertical motion over larger areas (light blue). Generally, this point is expected downstream
of a trough/PV cut-off due to vertically increasing advection of PV in combination with layer thickness advection and destabi-

lization underneath the high PV air, which is well represented in our example. On 31 May, a narrow trough accompanied by the cut-off low (C3) approached from the Atlantic to Iberia (cf. Fig. 5e). The areas of ascent on 31 May (Fig. 12a) correspond well with the regions of thunderstorm development in southeastern Germany, central France and the Netherlands (Fig. 12b). From 12 UTC until 18 UTC the next day, this trough narrowed while moving gradually northeastward accompanied by enhanced lightning activity moving from Central France and southern Germany to northeastern Germany and Poland (Fig. 12e,f). It is especially apparent that the multicellular system, which developed in the evening hours of 31 May (Section 3), emerged in a region of negative $\omega$ values ahead of the trough (Fig. 12c). On 1 June ascent occurs further to the east over Austria, the Czech Republic and northeastern Germany (Fig. 12e), which agrees well with the location of thunderstorm initiation.

The above discussion of PV filament evolution and lightning activity from 31 May to 1 June revealed an apparent link of this feature with lighting activity confined to the downstream side of PV filaments, where lifting is favoured. Considering the entire study period, we found 54 % of the lightning linked to a nearby PV cut-off (Fig. 11). Examining individual days reveals that on the day with the highest number of lightning detections (29 May) over 85 % of these events can be linked to a PV cut-off. Six out of eight days with the highest number of lightning flashes were between 27 May to 1 June. During this period, more than 75 % of the lightning strikes can be connected with one of the PV cut-offs. We conclude that cut-off low activity provided the necessary environment that favoured lifting within the prevailing unstable air mass and thus helped to trigger widespread thunderstorm activity in western and central Europe during this period.

## 4 Historical context

In this section, we assess the exceptional nature of the thunderstorm event, by relating the observed rainfall totals, the prevailing environmental conditions, and the occurrence of cut-off systems to the long-term data record.

### 4.1 Return periods of rainfall and propagation speed of convective cells

To estimate the severity of the rainfalls with respect to the rainfall climatology, we computed return periods (RPs) for each day during the study period in the REGNIE long-term record based on Equation (3). Afterward, we determined the highest RP (largest 24-hour rain total) for each grid point. Because long-term ($> 50$ years), highly-resolved ($1 \, \mathrm{km}^2$) and area-wide precipitation data are available only for Germany, we restrict our analysis to this area. REGNIE data derived from measurements at climate stations certainly underestimate precipitation peaks, but this is the case both for the study period and the 67-years reference period.

Extreme precipitation generally occurred locally, and only a few smaller regions were affected by high rainfall totals exceeding RPs of 5 years (Fig. 13). RPs in excess of 10 years were restricted to the southern parts of Germany (south of 52°N), except for a few grid points south of Berlin. Most of the precipitation fields with higher RPs occurred as clusters; for example, those near the border to France in Rhineland Palatinate and the Saarland (near Saarbruecken), northeast of Stuttgart, around Bad-Elster Sohl, or north of Munich. Several local maxima have RPs of up to 50 years, but a few hot spots, unevenly distributed in southern Germany, reach values in excess of 200 years (e.g., the observation in Bad Elster-Sohl; cf. Sect. 3). For

those locations, of course, precipitation was extreme, partly with new all-year records. Several hot spots have an almost circular
shape with the highest value located in the center. This is not an artefact of insufficient gauge density (i.e. only one station is
considered), since most fields take several precipitation stations into account (not shown).
This characteristic likely reflects the very slow propagation of the thunderstorms, which was substantially lower during the
study period compared to climatology (Fig. 4). Generally, convective storms detected between 2005 and 2017 (May/June: 3,428
cells) show significantly higher values of $10.2 \pm 4.9\,\mathrm{m\,s^{-1}}$ (mean $\pm$ std) and $9.5\,\mathrm{m\,s^{-1}}$ (median) compared to $5.9 \pm 2.9\,\mathrm{m\,s^{-1}}$
and $5.2\,\mathrm{m\,s^{-1}}$ in the study period. Only 14.4 % of all detected cells show values below $5\,\mathrm{m\,s^{-1}}$, which differs significantly from
the proportion in the study period with 47.3%. 15.5 % of the events propagated with a speed of at least $15\,\mathrm{m\,s^{-1}}$ (study period
only 1.5%; cf. Sect. 3.1.2).

## 4.2 Environmental conditions

We begin the analysis of the environmental conditions by comparing the SLI and V500 values observed at the seven sounding
stations during the study period with comparably low values during a 30-year period. The latter is represented by the annual
minimum of 22-day (same duration as study period) running mean values for May and June during 1981 and 2010. The box-
and-whisker plots (Fig. 14) on the left represent conditions during our study period (all 22 daily values) and on the right the
historical situation (in sum 30 values). Recall that the low values for both SLI and V500 were the peculiarity during the 2018
thunderstorm episode. By doing so, each of the 30 values taken into account in the right box-plot of each station has the same
temporal dimension (running mean of a 22-day period) as the median in the left box-plot of each station.
Both for atmospheric stability and mid-tropospheric flow speed, the interquartile range (the middle 50 % of all values) of
the left box-plot is mostly lower than the interquartile range of the right box-plot, illustrating the exceptional environmental
conditions of the 2018 thunderstorm episode. This applies in particular to the stations in Germany and Austria; stations in
France and in Switzerland tend to overlap (slightly) between the two interquartile ranges. As already mentioned in Sect. 3.1.2,
a large portion of SLI and V500 values during the event (left box-plot) are well below the basic and strict thresholds (cf.
Sect. 2.6).
To elaborate on both the peculiarity of the co-occurrence of low stability and weak mid-tropospheric flow and the persistence,
we investigate the probability of concurrent events (CE) by following the methodology of PIP16 (see Sect. 2.6) using the same
basic criterion. The event persistences of CE for each of the seven sounding stations during the extended study period in 2018
varies between 5 (Trappes) and 28 days (Munich; cf. legend in Fig. 15). At all three German stations, the defined concurrent
conditions prevailed over an extraordinarily long period (Essen: 17 days incl. 3 skip days; Stuttgart: 21 days incl. 1 skip days;
Munich 28 days incl. 3 skip days).
In order to assess the occurrence probability of event persistences of CE with long duration, we compare the event per-
sistences for the 2018 thunderstorm episode with a frequency analysis of CE between 1981 and 2017 (May/June; Fig. 15).
In doing so, the different amount of a certain event persistence with the length $n$ from the past between 1981 and 2017 are
determined for each sounding station. Subsequently, the relative frequency of the event persistence $n$ per station in Figure 15
is determined by dividing the absolute number of event persistence by the total number of all events. For example, the total
number of all events is approximately 100 for Trappes, Bordeaux, and Essen, approximately 150 for Stuttgart and Payerne, and
approximately 200 for Munich and Vienna reflecting the climatological distribution (north-to-south and west-to-east gradient)
of atmospheric stability (Mohr and Kunz, 2013).
The exceptional nature of the atmospheric conditions in 2018 is supported by the fact that, for example, the maximum event
persistence of 19 days between 1981 and 2017 (observed in Vienna) was exceeded in 2018 by two of the considered sounding
stations (Stuttgart, Munich). Additionally, when examining the individual stations, it can be seen that the CE event persistences
of 2018 at the stations Stuttgart, Essen, Munich and Payerne have never been observed since 1981. The same applies to the
Stuttgart sounding compared with the results in PIP16, where so far a maximum CE event persistence of 16 days (1960 – 2014,
but summer half-year) has been calculated. Furthermore, the relative frequency of CE at the other stations (Trappes, Bordeaux,
Vienna) is also rare (0.5 – 2 %).
## 4.3 Cut-off lows
In May and June, cut-off lows particularly affected southern Europe and the Mediterranean region. The highest frequency
during the climatological period from 1981 to 2010 is found over Portugal and Turkey but with values of around 4 % (contour
in Fig. 16; cf. Nieto et al., 2007b; Wernli and Sprenger, 2007). This means that during a 22-day period (same time horizon of
the study period) in May and June an average of 0.9 days (4 % of 22 days) with PV cut-off can be expected. During the 2018
thunderstorm episode, the anomaly of the PV cut-off frequency from the climatological mean was exceptionally large with
maximum values of around 40 % confined to northern Iberia and the Bay of Biscay in western Europe. This means that in 2018
a PV cut-off was additionally up to 10 times higher than the climatological mean, resulting in 9 additional days. The region
of anomalous PV cut-off activity expands northward over the British Isles and the adjacent Atlantic Ocean and the North Sea,
still with an excess of 20 % (additional 4 days compared to climatological mean). In other regions, PV cut-off occurrence was
similar to the climatological mean. As an orientation, note that the standard deviation of the cut-off low frequency between
1981 to 2010 (May/June) is 3 % over northern Iberia and the Bay of Biscay and between 1 and 2 % over the British Isles
(not shown). We conclude that the unusual blocking situation over Europe effectively caused cut-off formation on its upstream
flank, which then supported a (synoptic) lifting mechanism – the third ingredient for thunderstorm development, together with
instability and available moisture.
# 5  Discussion
In this study, we investigated the synoptic characteristics of an unusual three-week period of thunderstorm activity in central
Europe in May/June 2018. Interestingly, atmospheric blocking was key to providing the large-scale setting conducive for
convective in its vicinity. Because of the influence of large-scale mechanisms related to the block and affecting the entire
continent, a very high number of thunderstorms affected large parts of western and central Europe during an unusually long
period of three weeks. At the beginning of the thunderstorm period, southwesterly flow induced the advection of warm and
moist air masses into central Europe. Several studies have identified such a flow to provide convection-favouring conditions
(e.g., van Delden, 2001; Kapsch et al., 2012; Mohr, 2013; Merino et al., 2014; Wapler and James, 2015; Nisi et al., 2016; Piper
et al., 2019; Mohr et al., 2019). Subsequently, the low-pressure gradient associated with the blocking anticyclone over the
(adjacent) European sector prevented a significant air mass change. Thus, moist and conditionally unstable stratified air masses
were trapped in a stationary flow on the southern flank of high pressure for more than three weeks (and were re-circulated).
A few authors have already identified atmospheric blocking as a relevant influencer for widespread thunderstorms. PIP16, for
example, showed that the exceptional thunderstorm episode in 2016 in Germany was related to the sequence of Scandinavian
and European Blocking. Santos and Belo-Pereira (2019) identified a blocking-like dynamical structure in addition to a Western
European and a Scandinavian trough to be responsible for approximately three-quarters of all hail events across Portugal. By
combining ERA-Interim reanalysis and lightning detections over a 14-years period, Mohr et al. (2019) found that the presence
of a block over the Baltic Sea is frequently associated with increased odds of thunderstorm occurrence due to convection-
favouring conditions on its western flank (southwesterly advection of warm, moist and unstable air masses).
Upper-level cut-off lows or filaments of high PV that separate from the main PV cut-off were key in creating conditions
conducive for convective activity on the meso-scale. Associated lifting can increase CAPE and reduce Convective inhibition
(CIN) or can generate instability, if an entire column is lifted bodily until complete saturation in case of potential instability
(Markowski and Richardson, 2010). On several days during the peak thunderstorm activity, we found that the majority of
thunderstorms (based on lightning detections) can be related to a PV cut-off that favours lifting on its downstream flank. The
large positive anomaly in PV cut-off frequency, which seems to be relevant for the exceptionally high number of thunderstorms
during the study period, in turn was also related to atmospheric blocking. The latter repeatedly lead to the elongation of troughs
on its upstream flanks, which finally led to several cut-off lows. The general flow patterns consisting of this spatially extended
ridge flanked by troughs persisted over a period of three weeks.
Heavy rain events are a result of continuously high rain rates, whereby the duration of an event is linked to its propagation
speed and the size of the convective system (Doswell et al., 1996). In addition, a high concentration of water vapour at low
levels in the presence of strong updrafts, high environmental relative humidity, significant cloud depth below the freezing level
contribute to maximize rain accumulations, and potentially weak vertical wind shear, which tend to be correlated with weak
mid-tropospheric winds (Markowski and Richardson, 2010, Chap. 10.4). Due to the low propagation speeds, which contributes
to long rainfall duration during the thunderstorm episode in 2018, and high rain rates ($60\,\mathrm{mm\,h^{-1}}$ continuously over $50\,\mathrm{min}$),
some of the thunderstorms were able to produce torrential amounts of rain. Furthermore, the stagnant flow at mid-tropospheric
levels and thus the low vertical wind shear as a consequence of the blocking (cf. PIP16; Mohr et al., 2019) were also conducive
and frequently prevented most thunderstorms from developing into organized systems such as large MCS or supercells (cf.
Weisman and Klemp, 1982; Doswell and Evans, 2003; Markowski and Richardson, 2010). Most of the thunderstorms formed
as short-lived isolated cells or slow-moving multicellular clusters.

# 6 Summary and Conclusions

In our study, we investigated an exceptionally large number of thunderstorms in western and central Europe over a three-week period, mid-May to mid-June 2018, using a combination of observational data and model data to gain a more holistic view of the prevailing dynamical and thermodynamical conditions and the decisive trigger mechanisms for this unusual thunderstorm episode. Additional data over a climatological period helped to place the event in its historical context. The 2018 thunderstorm episode was exceptional due to several reasons: (i) the unusual large number of several thousand thunderstorms that caused more than 5 million lightning strikes (all types) in the study area; (ii) the combination of low stability (negative Lifted Index) and low wind speed at mid-tropospheric levels ($\leqslant 5\,\mathrm{m\,s^{-1}}$ at some locations) that prevailed almost every day during the 22-day period; (iii) the large cut-off low frequency that was responsible for the majority of convection triggering; and (iv) the high rainfall totals with several new records (e.g., Dietenhofen 86 mm / 1 h) mainly as a consequence of the low propagation speed of the storms in combination with high rain rates leading to several pluvial flash floods.

The other main conclusions drawn from our analyses are:

- Atmospheric blocking, albeit frequently associated with heatwaves and droughts, provided large-scale environmental conditions favouring convection in its vicinity when unstably stratified air masses are advected into Europe and/or become entrapped in stagnant flow.

- In the present paper, blocking is accompanied by a high cut-off frequency on its upstream side, which together with filaments of high PV provided the meso-scale setting for deep moist convection. Compared to climatology, the number of cut-off lows in parts of the study area during the study period was up to 10 times higher.

- The exceptional persistence of low stability combined with weak wind speed in the mid-troposphere prevailing over more than three weeks in some regions, especially in Germany and Austria, has never been observed during the past climatological period of 30 years. This situation was similar to the 2016 thunderstorm episode documented by PIP16, but with a much longer persistence.

- Blocking often associated with low mid-tropospheric wind speeds/low wind shear (cf. Mohr et al., 2019) reduces the development in severe organized convective systems. However, because of the low propagation speed of the storms related to the low-pressure gradient within the block, torrential rainfalls can occur on a local scale.

A growing understanding of the relationship between atmospheric blocking and deep moist convection can enhance – due to the associated persistence – the forecast horizon of thunderstorms on sub-seasonal time scales beyond the classical weather forecast time scale of a few days. This may, for example, help with disaster management, large outdoor activities, and the agriculture sector. It is only helpful, however, if blocked areas are correctly predicted. Recent studies show that this remains a challenge for present numerical weather prediction and climate models (Ferranti et al., 2015; Grams et al., 2018), which, for example, underestimate the blocking frequency in the Atlantic-European sector (Quinting and Vitart, 2019; Attinger et al., 2019).

In future, we intend to investigate statistically some of this study's results, such as the relationship between blocking, cut-off lows, air mass transport, and thunderstorm probability. Furthermore, we want to distinguish between different hazard types (hail, heavy rain, gusts) and associated types of thunderstorms and blocking regimes that reveal possible differences in atmospheric processes (e.g., jet stream).

*Acknowledgements.* The authors thank the various national weather service (DWD; MeteoSwiss; Météo-France; Royal Netherlands Meteorological Institute, KNMI; Zentralanstalt für Meteorologie und Geodynamik; ZAMG), the European Climate Assessment and Dataset (ECA&D) project, the Blitz-Informationsdienst von Siemens (BLIDS; namely Stephan Thern), the Integrated Global Radiosonde Archive (IGRA) and the European Severe Storms Laboratory (ESSL) for providing different observational data sets. In addition, we thank the European Centre for Medium-Range Weather Forecasts (ECMWF) for providing the operational analysis and the ERA-Interim reanalysis data. Furthermore, we thank Michael Sprenger (ETH Zurich) for compiling the ERA-Interim PV cutoff climatology and Florian Ehmele (KIT) for the post-processing of the REGINE data (return periods). The contributions of CMG, JFQ, and JaWa were funded by the Helmholtz Association as part of the Young Investigator Group „Sub-Seasonal Predictability: Understanding the Role of Diabatic Outflow" (SPREADOUT; grant VH-NG-1243). We acknowledge the constructive comments two anonymous reviewers, which helped to improve the quality of the paper.

*Data availability.* REGNIE (doi:10.1127/0941-2948/2013/0436), German precipitation data, and 3D radar data used in this paper are freely available for research and can be requested at DWD. Tracks of severe convective storms were calculated from the DWD radar data and are not freely available, but can be made available on request to Michael Kunz for research. Data from ECA&D can be downloaded via the project website (https://www.ecad.eu), from Météo-France via https://donneespubliques.meteofrance.fr/?fond=rubrique&id_rubrique=26, from MeteoSwiss via https://www.meteoswiss.admin.ch/home/services-and-publications/beratung-und-service/datenportal-fuer-lehre-und-forschung.html, and from ZMAG via https://www.zamg.ac.at/cms/de/klima/produkte-und-services/daten-und-statistiken/messdaten. Sounding data are available from the Integrated Global Radiosonde Archive (https://www.ncdc.noaa.gov/data-access/weather-balloon/integrated-global-radiosonde-archive) and data from the ESWD can be obtained via https://www.eswd.eu (see terms and conditions for academic or commercial use). Lightning data are not freely available, but can be requested from the Blitz-Informationsdienst von Siemens (http://blids.de). ECMWF ERA-Interim reanalysis and operational analysis are also online available via https://apps.ecmwf.int/datasets/data/interim-full-daily and the TIGGE webpage (control forecast step 0; https://apps.ecmwf.int/datasets/data/tigge). The methods to detect cut-off lows based on these data are given in Wernli and Sprenger (2007) and Sprenger et al. (2017) and for weather regimes in Grams et al. (2017).

*Author contributions.* All KIT authors jointly conceived the research questions of the study, continuously discussed the results and wrote the text passages for their respective contribution. SM analysed the ESWD data and together with JaWi the environmental conditions during the thunderstorm episode and in a historical context. In addition, SM wrote the introduction part together with CMG and the discussion/summary part of the paper together with MK and prepared the final draft version of the paper. JaWi also described the synoptic overview and the rainfall statistics in 2018, which were produced by HJP. The return periods of rainfall were investigated by MK, who also examined the lightning data.

Based on LAGRANTO, JQ performed backward trajectory analysis. MS contributed with the analyses of the storm track data (propagation
speed of convective cells). RP generated the PV cut-off data and its relationship to lightning activity was analysed by JaWa and CMG. In
addition, CMG contributed with the analysis of the weather regimes. Finally, all co-authors edited the final draft and provided substantial
comments and constructive suggestions for scientific clarification and further improvements.
*Competing interests.* The authors declare that they have no conflict of interest.
*Supplement.* The supplement related to this article is available online at: https://doi.org/10.5194/jn-0-1-2020-supplement.
*Video supplement.* Video supplement related to this paper is available from the Repository KITopen at:
https://doi.org/10.5445/IR/1000118571 and https://doi.org/10.5445/IR/1000118574.

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

**Table 1.** Top list of 1 h, 3 h, and 24 h rainfall totals (in UTC) within the study domain during the study period (AT = Austria, FR = France, GE = Germany). Note that 24 h value means precipitation between 00 and 00 UTC on the next day. Note that some stations only provide reports for the full 24 hours (e.g., Bruchweiler; Mauth-Finsterau). Further analyses regarding rain duration (RD), track length (in km), propagation speed (in m s$^{-1}$), and total track area (in km$^2$) are limited to Germany due to data availability. RD3 means a rain duration with a rain rate $> 3$ mm h$^{-1}$, RD35 $> 35$ mm h$^{-1}$, and RD60 $> 60$ mm h$^{-1}$. Note two tracks for the German events could not be identified by TRACE3D due to the overlapping of several cells, which were relatively quasi-stationary.

| Period | Location (Country) | Coordinates | Rainfall | Time | RD3 | RD35 | RD60 | Length | Speed | Area |
|---|---|---|---|---|---|---|---|---|---|---|
| 1 h | Dietenhofen (GE) | 49.4°N 10.7°E | 85.7 mm | 31 May 19 h | 1 h | 45 min | 35 min | | | |
| 1 h | Rohr-Dechendorf (GE) | 49.3°N 10.9°E | 71.0 mm | 09 June 15 h | 1 h | 40 min | 15 min | 84 | 15 | 608 |
| 1 h | Labécède-Lauragais (FR) | 43.4°N 2.0°E | 64.4 mm | 10 June 17 h | | | | | | |
| 1 h | Hohenberg an der Eger (GE) | 50.1°N 12.2°E | 61.4 mm | 31 May 18 h | 1 h | 55 min | 30 min | 30 | 6.6 | 215 |
| 1 h | Lenzkirch-Ruhbühl (GE) | 47.9°N 8.2°E | 59.8 mm | 31 May 20 h | 40 min | 30 min | 20 min | | | |
| 1 h | Langres (FR) | 47.8°N 5.3°E | 59.4 mm | 05 June 20 h | | | | | | |
| 1 h | Castanet-le-Haut (FR) | 43.7°N 3.0°E | 56.2 mm | 30 May 14 h | | | | | | |
| 1 h | Erlbach-Eubabrunn (GE) | 50.3°N 12.4°E | 55.6 mm | 31 May 17 h | 1 h | 50 min | 35 min | 25 | 4.4 | 190 |
| 1 h | Rouvroy-en-Santerre (FR) | 49.8°N 2.7°E | 54.3 mm | 28 May 22 h | | | | | | |
| 3 h | Prades-le-Lez (FR) | 43.7°N 3.9°E | 86.8 mm | 11 June 15 h | | | | | | |
| 3 h | Bad Elster-Sohl (GE) | 50.3°N 12.3°E | 86.3 mm | 24 May 15 h | 3 h | 25 min | 0 min | 16.5 | 4.6 | 105 |
| 3 h | Puchberg am Schneeberg (AT) | 47.8°N 15.9°E | 86.3 mm | 12 June 15 h | | | | | | |
| 3 h | Dietenhofen (GE) | 49.4°N 10.7°E | 86.2 mm | 31 May 21 h | ∼ 1 h 25 min | 45 min | 35 min | | | |
| 3 h | L'Oudon-Lieury (FR) | 49.0°N 0.0°E | 83.8 mm | 28 May 15 h | | | | | | |
| 3 h | Rocroi (FR) | 49.9°N 4.5°E | 79.4 mm | 27 May 21 h | | | | | | |
| 3 h | Leutkirch-Herlazhofen (GE) | 47.8°N 10.0°E | 79.1 mm | 08 June 18 h | ∼ 2 h 30 min | 45 min | 20 min | 8.7 | 3.2 | 42 |
| 3 h | Kleve (GE) | 51.8°N 6.1°E | 78.8 mm | 29 May 18 h | ∼ 2 h 45 min | 40 min | 20 min | 14.5 | 5.4 | 70 |
| 3 h | Sulzberg (AT) | 47.5°N 9.9°E | 78.0 mm | 04 June 18 h | | | | | | |
| 24 h | Mauth-Finsterau (GE) | 48.9°N 13.6°E | 166.5 mm | 12 June | ∼ 8 h 0 min | 55 min | 20 min | 9.2 | 3.4 | 44 |
| 24 h | Bad Elster-Sohl (GE) | 50.3°N 12.3°E | 154.9 mm | 24 May | ∼ 8 h 15 min | 20 min | 0 min | 16.5 | 4.6 | 105 |
| 24 h | Bruchweiler (GE) | 49.8°N 7.2°E | 145.0 mm | 27 May | ∼ 2 h 30 min | 1 h 5 min | 50 min | 20.5 | 5.7 | 130 |
| 24 h | Monein (FR) | 43.3°N 0.5°W | 130.0 mm | 12 June | | | | | | |
| 24 h | Ger (FR) | 43.2°N 0.1°W | 126.4 mm | 12 June | | | | | | |
| 24 h | Mont Aigoual (FR) | 44.1°N 3.6°E | 124.1 mm | 28 May | | | | | | |
| 24 h | Les Bottereaux (FR) | 48.9°N 0.7°E | 123.0 mm | 04 June | | | | | | |
| 24 h | Navarrenx (FR) | 43.3°N 0.8°W | 117.0 mm | 12 June | | | | | | |
| 24 h | Puchberg am Schneeberg (AT) | 47.8°N 15.9°E | 116.3 mm | 12 June | | | | | | |

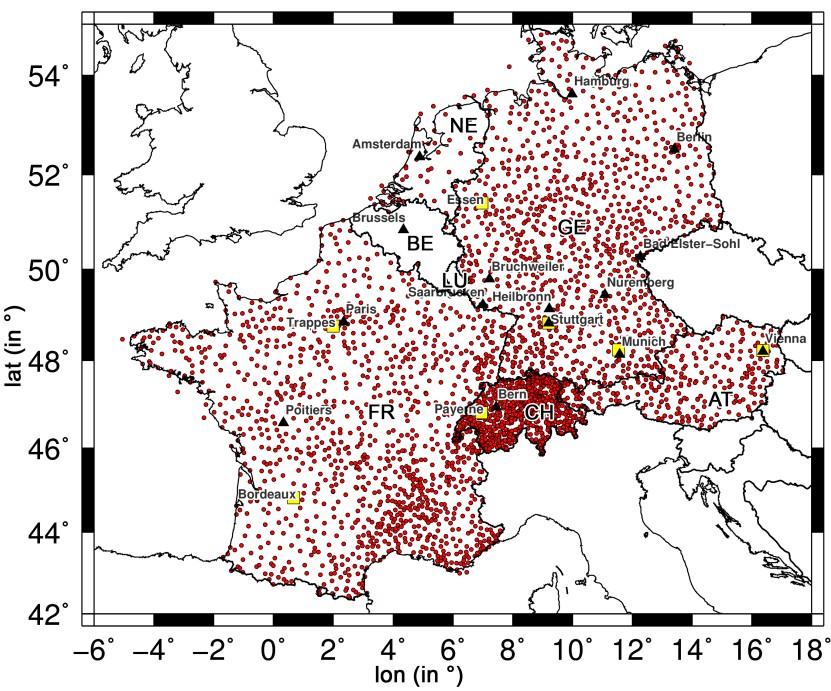

**Figure 1.** All considered precipitation stations (in red) collected from ECA&D and the three national weather services (France, Germany, Switzerland; see Sect. 2.1.3). In addition, the seven investigated sounding stations are shown (in yellow, see Sect. 2.1.5). Some relevant locations are also presented, which are used in the text. Defined country codes are FR = France, BE = Belgium, NE = Netherlands, LU = Luxembourg (the latter three: Benelux), GE = Germany, CH = Switzerland, AT = Austria.

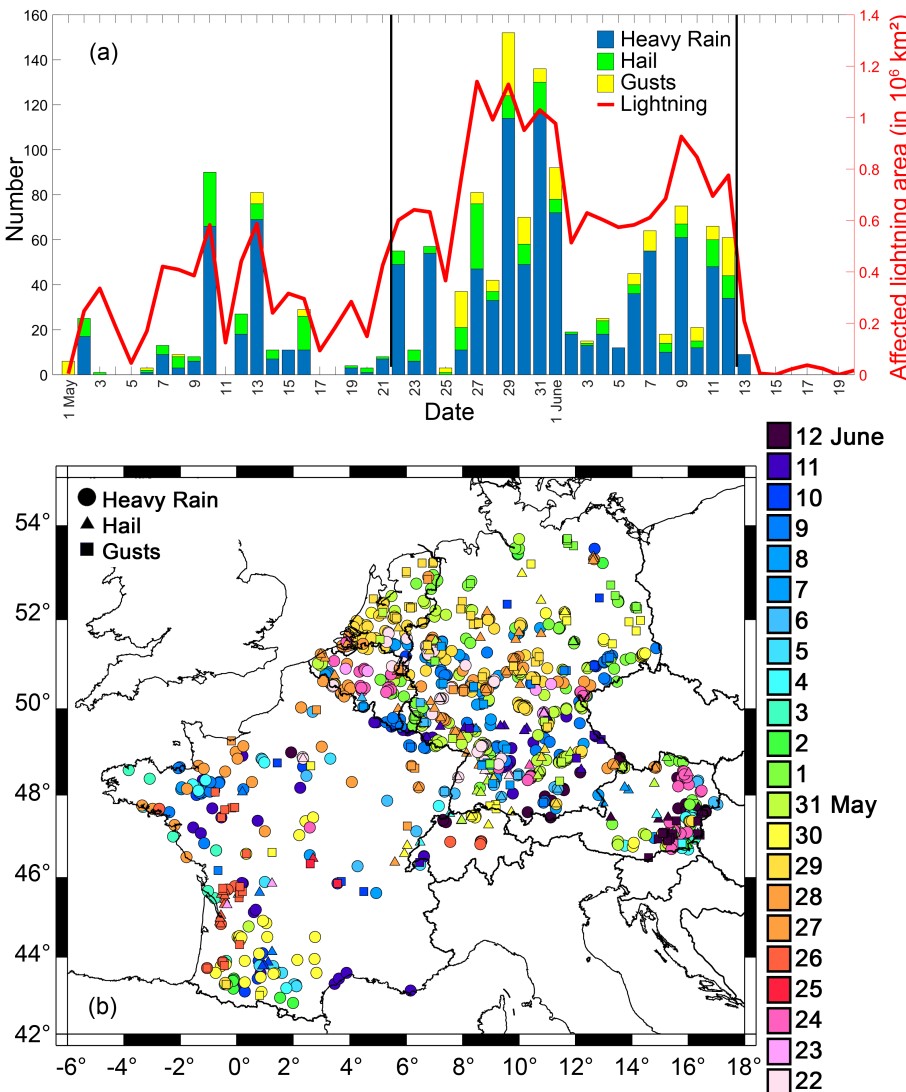

**Figure 2.** (a) Time series of all recorded ESWD reports (heavy rain in blue, hail in green, convective gusts in yellow) in the study domain during the extended study period including the daily total area affected by lightning in $km^2$ (in red). Vertical black lines indicate the study period (22 May to 12 June 2018). (b) Related regional distribution of the different phenomena (heavy rain ●, hail ▲, convective gusts ■) during the study period.

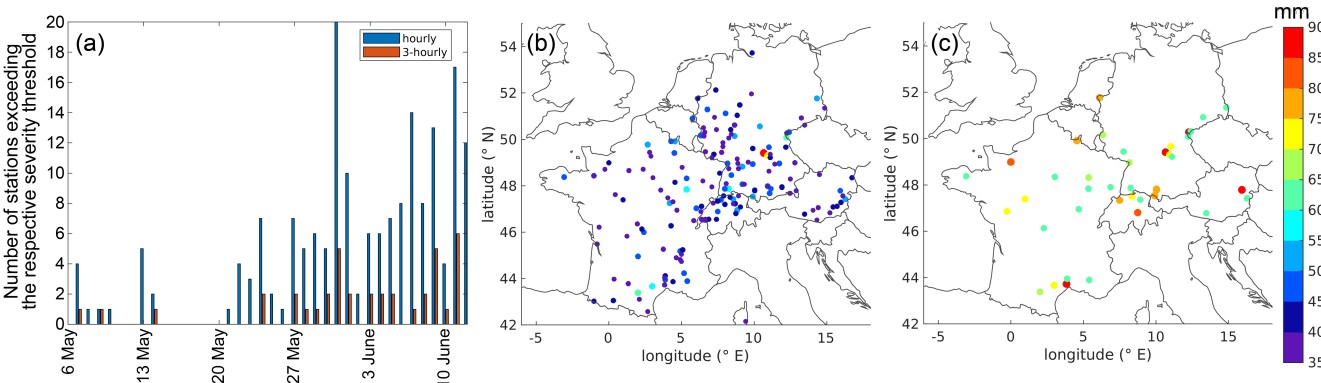

**Figure 3.** (a) Time series of the number of stations exceeding precipitation thresholds of > 35 mm hourly (blue) and > 60 mm over 3-hours (red) including the location and total maximum of (b) hourly and (c) 3-hour sums of the respective station during the study period (22 May to 12 June).

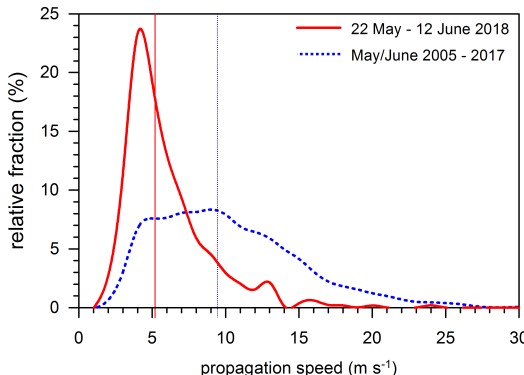

**Figure 4.** Histogram of the propagation speed of convective cells (increments of $1\,\mathrm{m\,s^{-1}}$; spline filter) detected by TRACE3D in Germany during the study period (red) and for all convective cells between 2005 and 2017 (May/June; blue); vertical lines indicate the median of the two samples.

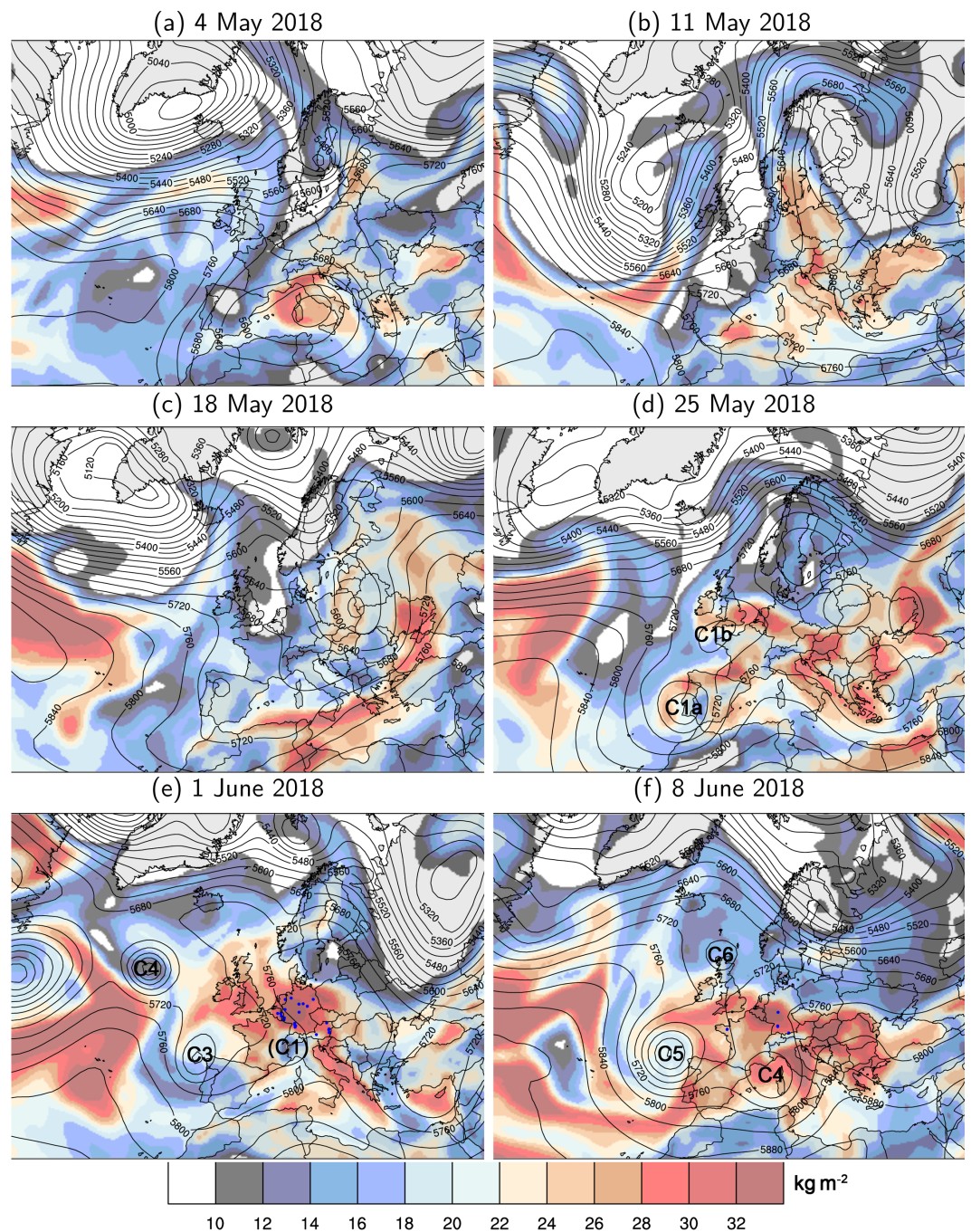

**Figure 5.** 500 hPa geopotential height (contours every 40 gpm) and vertically integrated water vapor (IWV, shaded in kg m$^{-2}$) for selected days at 00 UTC during the extended study period: (a) 4 May, (b) 11 May, (c) 18 May, (d) 25 May, (e) 1 June, and (f) 8 June (ERA-Interim). Several cut-off lows during the study period mentioned in the text are indicated with numbers (C1, ..., C6). Small blue dots (in e and f) mark the ESWD reports on heavy rain from Fig. 2. Note that there are no ESWD reports for the first four panels.

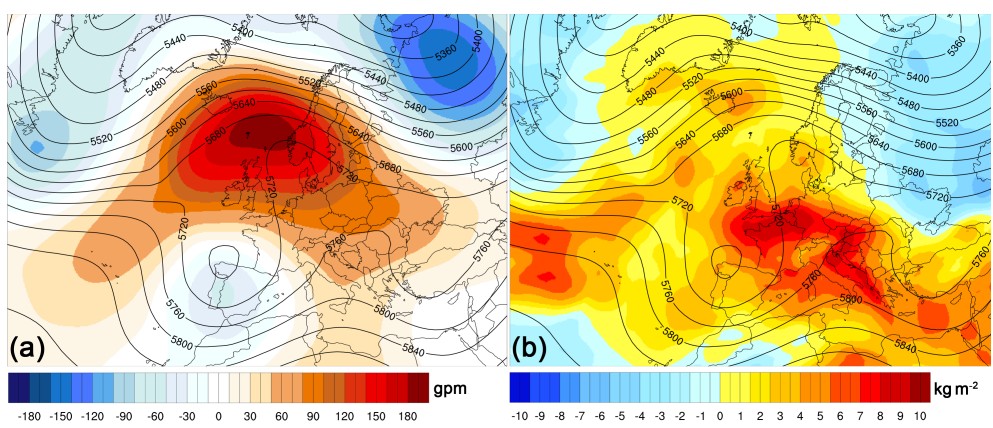

**Figure 6.** Composite mean $500\,\mathrm{hPa}$ geopotential height (contours every $40\,\mathrm{gpm}$) and in (a) anomaly with reference the climatological mean in May and June ($1981-2010$; shaded in gpm) and in (b) together with anomalies of the IWV with reference to the climatological mean in May and June ($1981-2001$; shaded in $\mathrm{kg\,m}^{-2}$ (based on ERA-Interim).

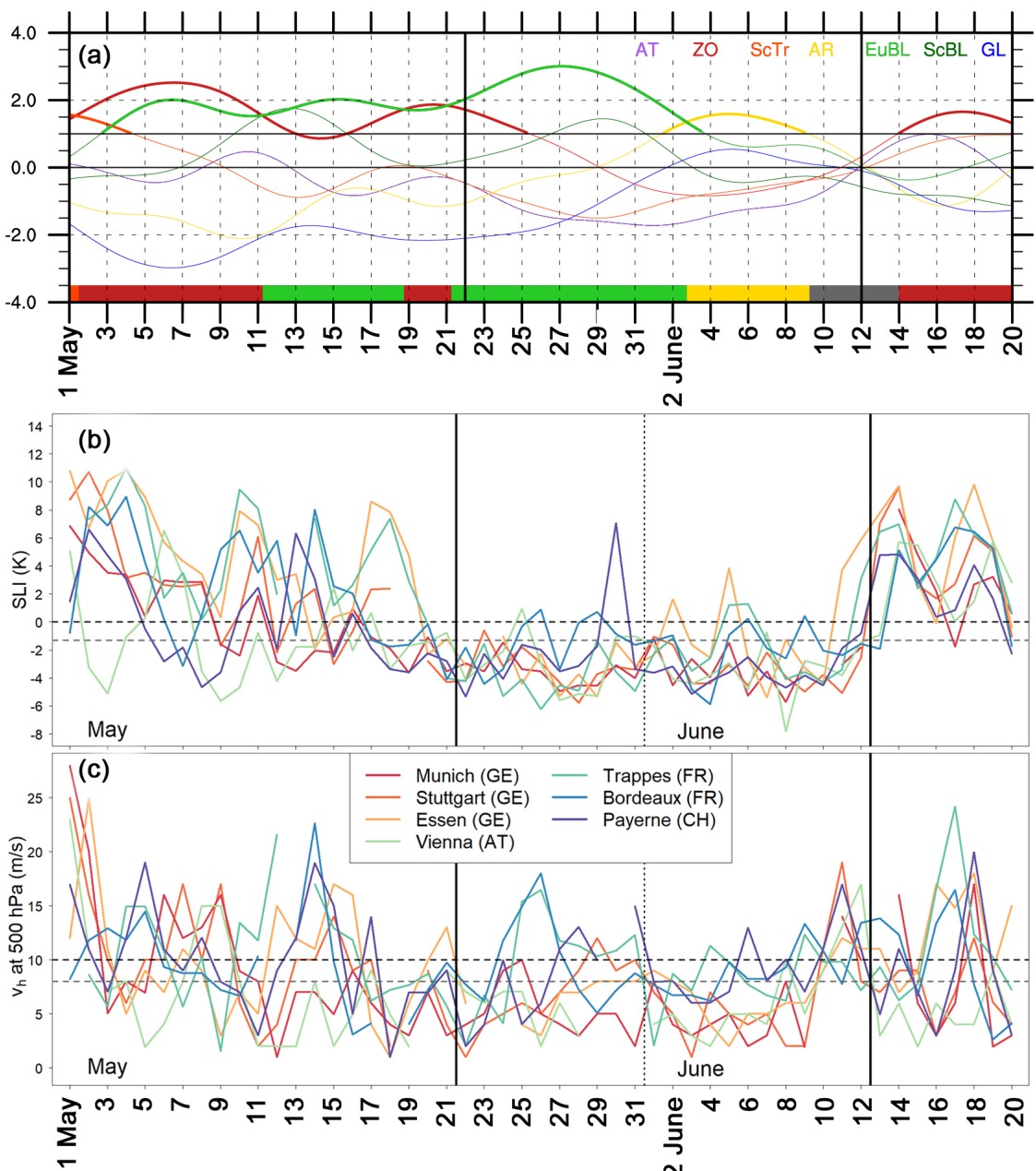

**Figure 7.** Time series of three different parameters during the extended study period from 1 May to 20 June 2018: (a) Atlantic-European weather regime life cycles including normalized projection into all seven regimes (ECMWF analysis). Active regimes (according to the life cycle definition in bold) are Zonal regime (ZO, dark red), European Blocking (EuBL, light green), Atlantic Ridge (AR, yellow), no regime (grey; see text for details). (b) Surface-based Lifted Index (SLI in K) and (c) horizontal wind speed at 500 hPa (V500 in m s$^{-1}$) for the 12 UTC sounding at seven European stations. Horizontal black/gray dashed lines indicate thresholds as defined in PIP16 (Basic criterion: 0 K & 10 m s$^{-1}$; Strict criterion: -1.3 K & 8 m s$^{-1}$; cf. Sect. 2.6). Vertical black lines indicate the study period.

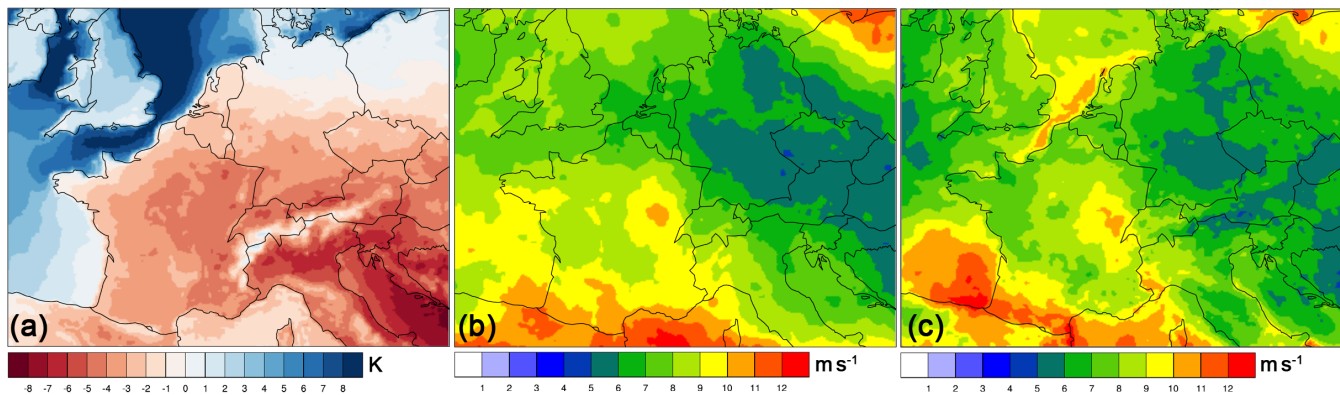

**Figure 8.** (a) Surface-based Lifted Index (SLI in K), (b) horizontal wind speed at 500 hPa (V500 in m s$^{-1}$), and (c) bulk wind shear between 500 hPa and 10 m (BWS in m s$^{-1}$) averaged over the study period from 22 May to 12 June 2018 (12 UTC; ECMWF analysis).

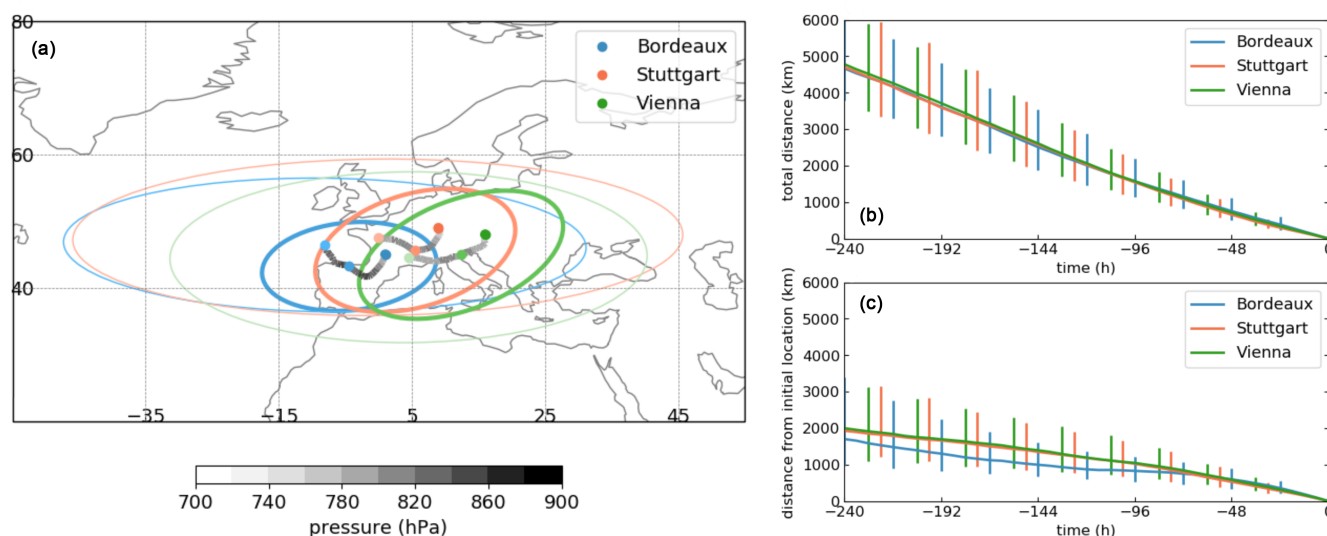

**Figure 9.** 10-day backward trajectory analysis from 22 May to 12 June 2018. (a) Median backward trajectories coloured by their median pressure (hPa) for three locations given in legend. The ellipses show the dispersion of the trajectories around their median location (dots) at 10 days (thin ellipses) and 5 days (bold ellipses) prior to arriving at the location. The ellipses enclose about 2/3 of the trajectories. (b) Temporal evolution of median distance travelled by the trajectories (km) prior to arriving at one of the locations given in legend. Bars show the interquartile range. (c) As in (b), but for distance from the initial location.

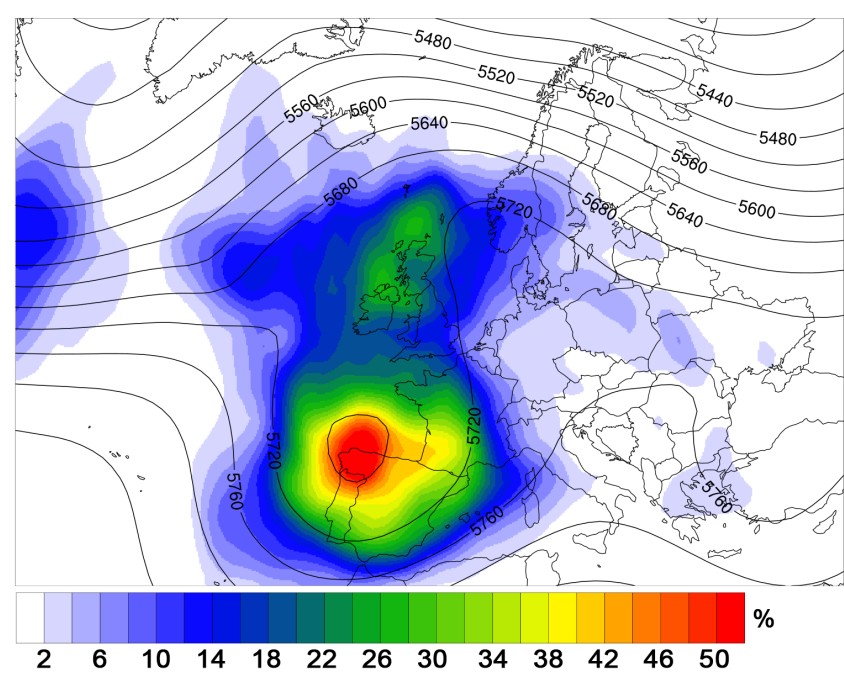

**Figure 10.** Composite mean of 500 hPa geopotential height (contours every 40 gpm) and cut-off low frequency (color shading in %) during the study period (ERA-Interim).

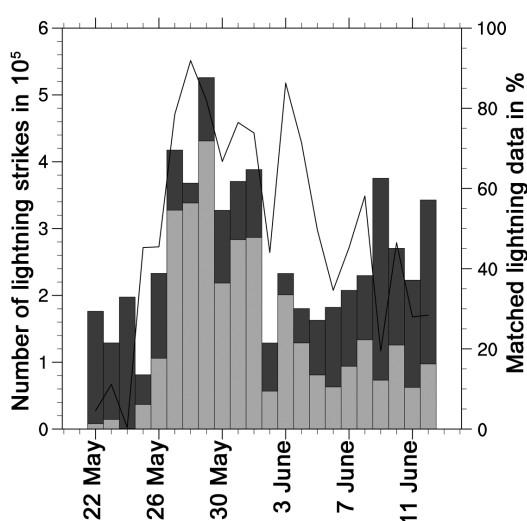

**Figure 11.** Lightning strikes per day (03 UTC – 03 UTC on the next day) during the study period for all thunderstorm events (dark grey bars) and those thunderstorms that can be linked to a cut-off low (light grey bars). The black line shows the percentage of lightning strikes per day that can be attributed to a cut-off low.

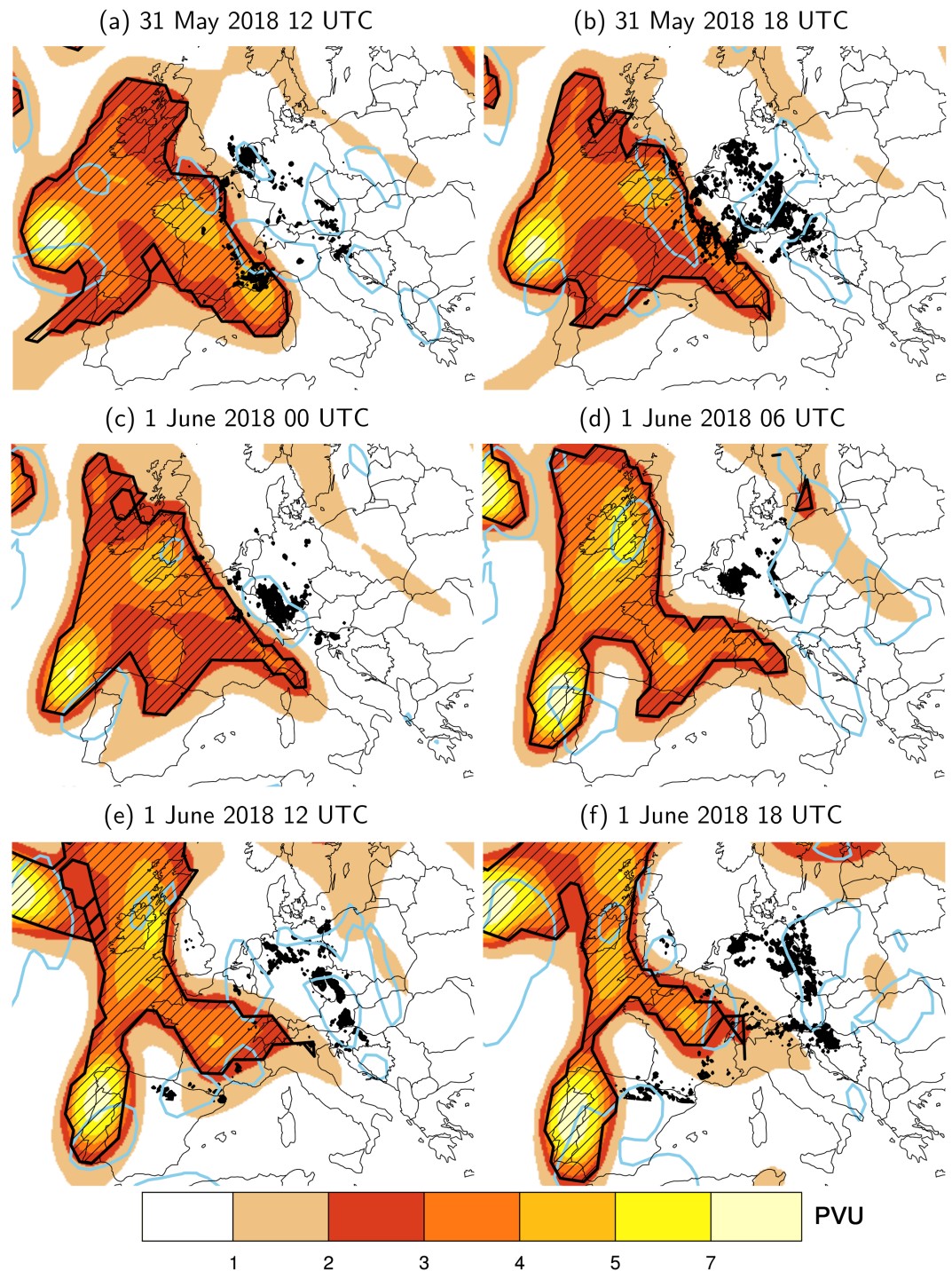

**Figure 12.** Lightning data (dark black dots) for 6-hour time spans centered around the respective time and PV on the 325 K isentropic surface (shaded in PVU; ERA-Interim). Regions of ascent at 500hPa are indicated by light blue contours ($\omega = -0.1\,\mathrm{Pa\,s^{-1}}$); ERA-Interim). Hatching indicates masks of objectively identified cut-offs on the 325 K isentropic surface (See Supplementary Fig. 2 including the buffer zone.)

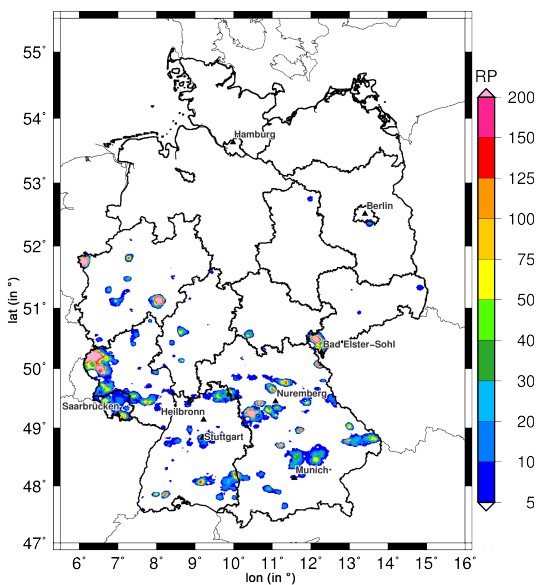

**Figure 13.** Return periods (RP) of the highest 24-hour rainfall totals that occurred during the study period at each grid point (REGNIE precipitation data; reference period: 1951 – 2017, summer half-year).

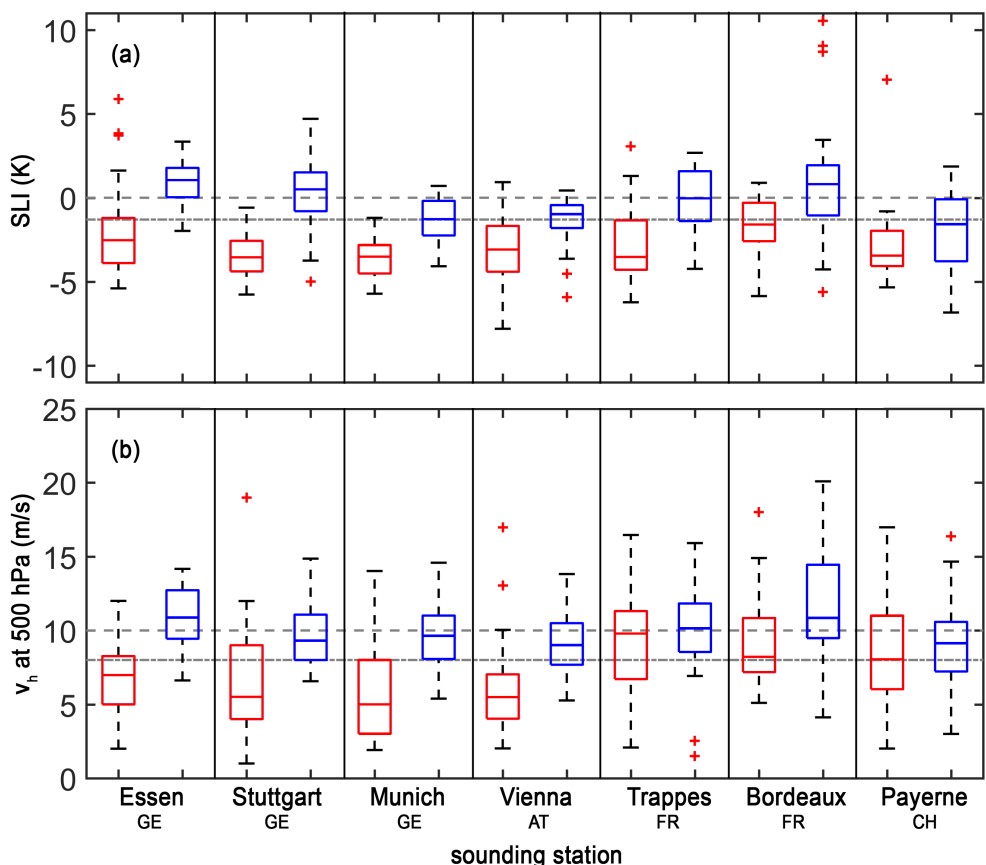

**Figure 14.** Box-and-whisker plots (median, 1st/3rd quartiles, whisker = +/–2.7$\sigma$, outliers) for the seven sounding stations. The left box-plots (in red) of each station include all values of (a) SLI and (b) V500 during the study period at 12 UTC, the right box-plots (in blue) include the annual minimum of the running mean (22 days) during May and June between 1981 and 2010. The two gray lines indicate thresholds as defined in PIP16 (Basic criterion: 0 K & 10 m s$^{-1}$; Strict criterion: -1.3 K & 8 m s$^{-1}$; cf. Sect. 2.6). Note that the median on the left box-and-whisker plots is calculated identically as all 30 values in the right box-and-whisker plots.

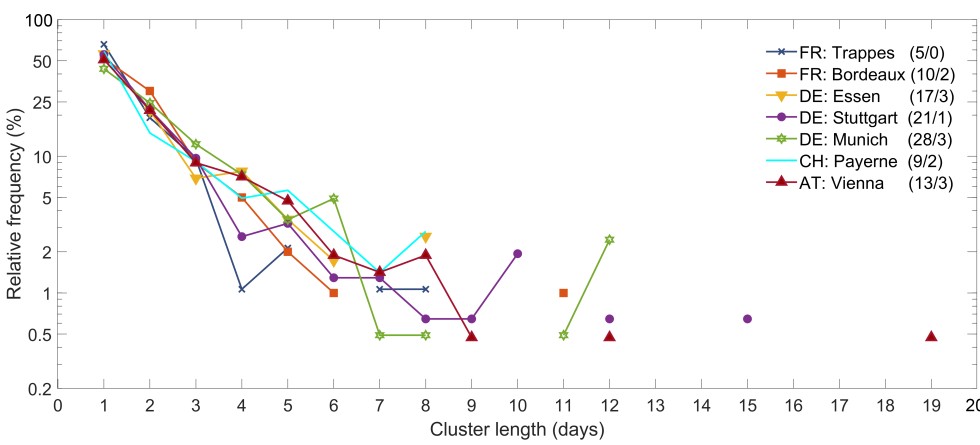

**Figure 15.** Relative frequency of clusters of consecutive days exceeding the basic criterion for concurrent events with low stability (SLI $< 0$ K) and weak flow (V500 $< 10$ m s$^{-1}$) at the seven sounding stations (Trappes, Bordeaux, Essen, Stuttgart, Munich, Payerne, Vienna) during $1981-2017$ (May/June). Maximum days with event persistence $n$ (including skip days $m$) during the extended study period in 2018 (Mai/June) are shown in the legend ($n/m$).

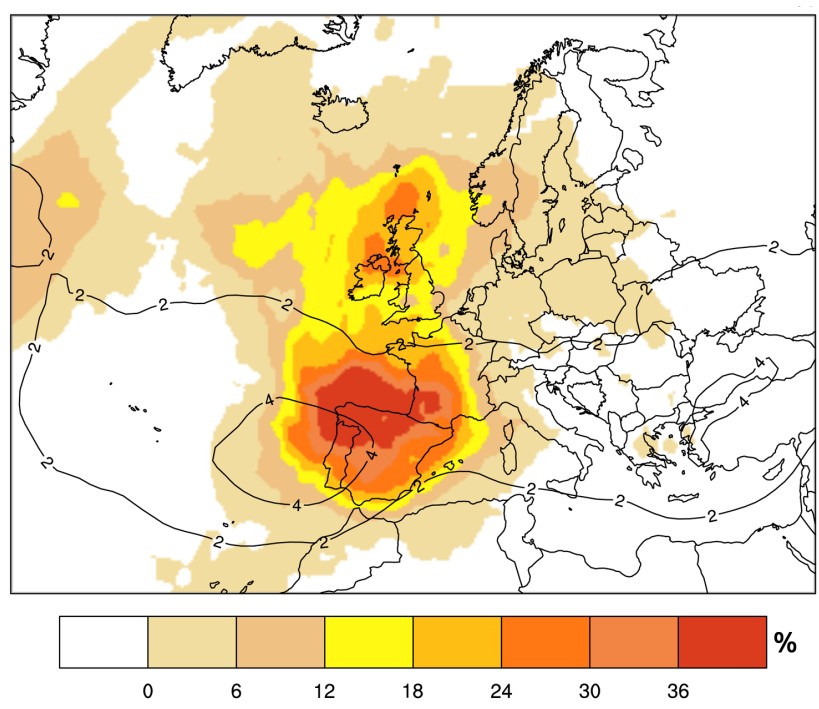

**Figure 16.** Climatological mean percentage of days with a cut-off low in May and June (black contours; every 2 %; for May and June 1981 – 2010) and anomaly percentage of days during the study period (shaded in % with reference to mean percentage of days in May and June; ERA-Interim).