# Peer review of "The role of large-scale dynamics in an exceptional sequence of severe thunderstorms in Europe May/June 2018"

_Weather and Climate Dynamics, 2020_

## Referee Comment (RC1) · Anonymous Referee #1 · 11 Feb 2020

**Review for WCD-2020-1**

**General Comments**

This study describes an exceptional period of thunderstorm activity in western Europe during summer 2018 and explores the associated synoptic-scale conditions; in particular, the role of blocking and associated upstream cut-off lows. The event is also placed in a climatological context using long-term records from surface stations, upper-air soundings, and reanalysis.

This is an interesting, thorough and well-written piece of work, which I have no hesitation in recommending for publication, subject to a few minor revisions as detailed below. I would like to thank the authors for their efforts, which made for an easy and enjoyable review.

**Specific Comments**

1.  My most significant comment relates to your conclusion regarding the slow movement of convective systems during the event. You provide clear evidence (in Fig. 9) that storm motion was, on average, much lower than is typically observed in this region and at this time of year and suggest that this was key to the extreme rainfall totals. However, storm motion is not the only relevant factor for heavy precipitation. As discussed by Doswell et al. (1996), accumulated rainfall depends on two things: average rain rate and total rainfall duration. Slow cell motion contributes to long rainfall durations, but one must also consider system size (in the direction or storm motion) and the existence of back-building convection, both of which may lead to echo training. Rainfall rates must also be considered. Given that you have hourly gauge data and radar observations, it should be possible to assess all of these things. While this might not be practical for the whole study period, you should consider doing so for the most extreme events noted in Table 1. One option would be to add an extra column to the table, providing a brief description of the storm(s) that caused the rainfall totals (including their estimated motion). At the very least, it would be good to demonstrate that the general characteristic of slow storm motion applies to some of the individual extreme events.

2.  You state in your abstract that low vertical wind shear "prevented thunderstorms from developing into severe organized systems". However, reports of hail up to 5 cm in diameter (L234; L239) suggest that this is not entirely true. Clearly, wind shear in certain areas and on certain days was sufficient for the development of organised convection (probably supercells). As such I think this statement needs revising.

3.  Please provide some citations for reports of storm impacts (in the opening paragraph of the introduction and section 3.1). These could simply be links to online news or social media reports.

4. The second paragraph of the introduction doesn't really fit and has only limited direct relevance to your study. As such I would suggest removing it (although parts of it could potentially be incorporated elsewhere in the introduction).

5. I know that 1981–2010 is a standard 30-year climatology period, but for this study you should consider using the full ERA-Interim record (1979–2017) in order to provide a more complete historical context for the 2018 event. I believe the sounding data will go back this far as well.

6. On Line 140 you claim that surface-based lifted index (SLI) offers "the best representation of convective environmental conditions in central Europe". I would expect CAPE to provide a more robust measure of surface-based instability, given that it considers the full column rather than just a single level (see discussion in Doswell and Schultz 2006). Certainly it has been shown to usefully discriminate between severe and non-severe convective environments in various parts of the world, including Europe (Pucik et al. 2015; Taszarek et al. 2017). It is also available as a diagnostic from ERA-Interim, so could easily be analysed spatially as well as from the point soundings. I appreciate that repeating all of your instability analysis using CAPE would be time consuming and is likely to show comparable results, so I will not request this. However, you should provide some further justification for why you chose to use SLI over CAPE.

7. You use a very high reflectivity threshold (55 dBZ) for identifying and tracking convective storms. Such a value is more characteristic of hail than intense rainfall. As such I wonder if the majority of storms went undetected, leading to an unrepresentative velocity estimate. One simple way to check this would be to see if the storms that produce some of the extreme rain accumulations listed in Table 1 were detected. At the very least this should be noted as a limitation of your radar-based analysis.

8. It would be good to include a figure showing the different weather regimes discussed in section 2.3 (or, at least, the ZO, EuBL, and AR regimes that dominated during the study period). Perhaps this can be found elsewhere. If so please refer to the specific figure(s) in the relevant paper(s).

9. In discussing the persistence analysis in section 2.5, and the associated results in section 5.2, I found the reference to "cluster length" confusing, in part because K-Means clustering is used in the analysis of weather regimes. I would change this to just "event duration" or "event persistence".

10. Could you provide a few more details on the origin of the "basic" and "strict" criteria for SLI and mid-tropospheric wind speed, so that the reader doesn't have to go to PIP16 for this information? Also, I'm not a fan of the notation $TH_{BC}$ and $TH_{SC}$ for these and would argue that they can be eliminated (you can just refer to the basic/strict criterion).

11. I recommend using V500 (rather than $v_{500hPa}$) to indicate the 500hPa wind speed.

12. I suggest using "total" or "accumulation" when referring to precipitation amounts rather than "sum" (e.g. in Table 1 and on L260–261).

13. I'm confused as to why you focus the opening paragraph of section 3.3.1 and the first three panels of Fig. 4 on the two weeks before your main study period. In my view it would make more sense for Fig. 4 to show more regular snapshots from the study period, so that the reader can more easily see the temporal evolution described in the subsequent two paragraphs. In particular, it would be good to include one snapshot that shows the second cut-off low (C2) and one close to the end of the study period.

14. In section 5.1, you note that several of the rainfall return period maxima in Fig. 13 "have an almost circular shape with the highest value located in the center" and suggest that this characteristic "reflects the very slow propagation of the thunderstorms". However, could it instead be an artefact of insufficient gauge density? If only only one gauge recorded the event, this information would be spread laterally by the gridding procedure, giving the impression of a small circular shape. It might be worth overlaying the gauge locations on this plot to check how many gauges are associated with each maximum.

15. The description of Fig. 14 at the start of section 5.2 is rather confusing and should be revised. In particular, I found it hard to understand how the distributions for the climatological period were derived.

16. Several of the figures could be improved in a few ways. Specifically, I recommend the following changes:
    ○ Fig. 3: The top and bottom rows could arguably be combined. In this case, rather than colouring the symbols on the map by rainfall amount you could just make them blue for > 35 mm/h and red for > 60 mm/3h; then use these same colours for the bar plot (with red bars overlaid on blue bars).
    ○ Fig. 7: I don't think it's necessary to state the two thresholds within the plots; this information can be provided in the caption (with reference to the dashed lines).
    ○ Fig. 12: I would get rid of the hatching showing the objectively identified cut-offs and use a darker contour for the pressure vertical velocity.
    ○ Fig. 14: It would be helpful to use different colours for the box-and-whisker plots corresponding to the study period and the 1981–2010 climatology. Use the same colour convention for Fig. 9.
    ○ Fig. 15: In the legend, rather than putting "(2018: N days incl. M skip days)" for each station I would just put "(N/M)" and then explain what these numbers indicate in the caption. So, for example, for Essen you would put "(17/3)" instead of "(2018: 17 days incl. 3 skip days)".
    ○ Fig. 16: Rather than plotting the percentage difference from the climatological frequency (which is confusing because it is a percentage of a percentage), I recommend expressing this difference in terms of the standard deviation of the climatological frequency. This will highlight whether the 2018 frequencies were exceptional in the context of typical year-to-year variability.

**Technical Corrections**

1.  There are a few issues with tenses in the text. For example, the opening sentence of the introduction is written in the present perfect tense (use of "has been"), but should be in the past tense ("was"). The same goes for L261. The last sentence of the opening paragraph of section 3.3.3 is written in the simple present tense but should be in the past tense.

2.  L23: Parentheses are (are not) for references and clarification (saving space) (Robock 2010). Please modify this sentence accordingly.

3.  L36: "...serve to precondition the thermodynamic environment."

17. L39–42: To which event are you referring here: 2018 or 2016? I suggest rewording this paragraph to make this clear. Similarly, you should state explicitly the event you are referring to on L71.

18. Line 60: Lifting will only lead to the release of CAPE (i.e. convective initiation) if it is sufficient for parcels of air to reach their level of free convection; however, it may still act to destabilise the column (increase CAPE) and erode lids (reduce CIN).

19. Line 90–91: Suggest revising the end of this sentence as follows: "...based on reports from storm chasers, eyewitnesses, voluntary observers, meteorological services, and news media."

20. L97–98: "... Météo-France (1223/1935 stations with hourly/daily data)..."

21. Line 100: These are the national meteorological services of all the countries in your study so I don't think you need this last part of the sentence.

22. Line 132: "wind speed and direction"

23. L146: What is the altitude of the lowest level?

24. L199: Change "fewer" to "less".

25. L239–240: Suggest revising this sentence as follows: "Many of the record-breaking 1h and 3h rain totals occurred within this period (see Sect. 3.2)."

26. L255–256: "...the latter on the day with the second most ESWD severe weather reports (cf. Sect. 3.1)." Rather than referring to the previous section here please specify the actual date.

27. L257–258: You deal with the variations in large-scale forcing for convection later in the paper so I don't think it is necessary to include this here (unless you want to explicitly refer to the relevant sections).

28. L271: Change "were" to "where".

29. L272–273: This sentence is confusing and should be revised.

30. L357: Change "vast parts of" to "much of".

31. L362: Change"exemplarily shown for" to "exemplified by".

32. Line 380–381: Change "and was already mentioned at the end of Section 3.2" to just "(Section 3.2)".

33. L510: Advected where?

34. L526: Get rid of "(global/regional)".

---

## Referee Comment (RC2) · Anonymous Referee #2 · 23 Feb 2020

Overview: The paper presents a case study of a long-lasting thunderstorm series over France/central Europe in May/June 2018 that occurred south of a blocking high. The synoptic situation persisted for several weeks. The thunderstorms were associated with cut-off lows/potential vorticity filaments that formed on the south-west of the blocking high. As a result, numerous severe convective events such as flash floods, hail and wind gusts were recorded. The authors use multiple different data sets and methods to show how the large-scale dynamics contribute to the thunderstorm series and that this event was exceptional.

Overall, I like the author's idea of studying this event from synoptic down to the convective scales. Moreover, it is an interesting case. However, I think the manuscript was difficult to follow and can still be improved considerably. In the current form, it was unfortunately no pleasure to read through the study. My main criticism is that large parts of the paper (chapter 2,3,4, but also in the introduction) read like a collection of single parts which are not really connected with one another. A central theme seems to be missing. I think, the authors should restructure the paper or parts of it and follow a clear path, e.g. from large-scale to the small scale or the other way around. If this is not possible, they should at least clarify the purpose of each (!) chapter at the beginning (as it is done in chapter 5) to facilitate the reading. Moreover, in my opinion, the writing can be improved, too. Some sentences are too long, which makes the text hard to read. Just write necessary information and just reference to papers that are relevant for your topic. Make the sentences clear and concise. Please connect the single chapters and (sub)sections with one another!

I will explain my criticism in more detail in the following:

(i) In the introduction, the authors switch strongly between different topics: first they introduce the case and its impacts in a few sentences. Then they describe convective development due to scale interactions (mainly lifting processes). Afterwards they describe the case again with focus on blocking which is described more general thereafter. In the successive part, the authors explain cut-off lows in the potential vorticity framework. Afterwards they switch back to the topic of blocking. However, these single parts are often unconnected with one another which is confusing for the reader!

(ii) In the data and methods chapter (chapter 2), data sources are often introduced without clarifying why the authors will need the data. At least some overview at the beginning of this section – how the study was designed and/or what data satisfies which purpose – would help the reader tremendously! Are there any new methods? Please clarify!

(iii) The same applies for chapters 3,4, and partly 5! Try to connect the single parts, try

not to jump unnecessarily between topics. In the current version, it is really confusing for the reader.

With respect to the methods my main concern is the usage of the 500hPa-wind instead of vertical wind shear. At least additionally analyzing shear had the advantage that your work can be compared more easily to the existing literature of convective events. Furthermore, I am missing evidence, that the thunderstorms have been single cells rather than multicells, MCS or slow-moving (HP) supercells.

Specific comments:

Abstract, p.1 line 2: "80mm" - what is the temporal range? a few hours?

p.2, line 25-36: I am missing the general ingredients of convection here: instability, moisture, lift and shear. The ingredients-based concept is first mentioned in the Discussion chapter (chapter 6), I think it would be fitting in the introduction, too. Moreover e.g. Markowski and Richardson, 2010 (their chapter 10.4) and Doswell III, C. A., Brooks, H. E., & Maddox, R. A. (1996) (Flash flood forecasting: An ingredients-based methodology. Weather and Forecasting, 11(4), 560-581) treat flash flood events. Especially in the Markowski and Richardson book, you can find a very similar synoptic pattern that led to flash flood events in the US (please refer to the publications mentioned therein).

p.2, line 34: "all these mechanisms" - which ones are meant here?

p.2, lines 51-61: There are some publications concerning the PV framework and convection: e.g. Russell A, Vaughan G, Norton EG. 2012. Large-scale potential vorticity anomalies and deep convection. Q. J. R. Meteorol. Soc. 138: 1627–1639. DOI:10.1002/qj.1875 ; Morcrette CJ, Lean H, Browning KA, Nicol J, Roberts N, Clark PA, Russell A, Blyth AM. 2007. Combination of mesoscale and synoptic mechanisms for triggering of an isolated thunderstorm: a case study of CSIP IOP 1. Mon. Weather Rev. 135: 3728–3749. Can you please put your work in context with the existing literature?

[Figure]

p.3, line 62: "A connection between atmospheric blocking and heavy precipitation events..." - Why again blocking? The sentence is almost identical to that on page 2, lines 49-50. Why don't you merge these parts?

p.3, line 68-70: "[..] such situations are usually associated with weak wind speeds at mid-tropospheric levels (cf. PIP16), so that thunderstorms become almost stationary and usually do not develop into organized structures such as large mesoscale convective systems or supercells." - first: where is the wind weak? in the high, the low, at the western flanks? second: what about HP-supercells (high precipitation supercells)? Can you please comment on HP-supercells.

p.3, line 83: What do you mean with "secondary effects"? Please elaborate.

p.3, line 85: "(May/June)" - These are the whole months (1.5-30.6)? It is confusing since you already stated two different periods in the text before.

p.4, line 87-93: Please clarify what the purpose of the ESWD data is. Do you use different quality levels or all? Why don't you show the reports also in e.g. Belgium or Italy?

p.4, line 88: It is good to know that the ESWD collects data about heavy rain, hail and wind gusts. However, what data did you use for the analysis?

p.4, line 90: better: "[..] mainly based on reports of storm chasers, [..]"

p.4, line 108: Is there a description of the REGNIE data in English for non-Germans, too?

p.4, line 104-111: Why did you decide to use the REGNIE data. The data seems to interpolate measured precipitation on a regulare grid. Is the REGNIE data suitable to analyse extreme convective precipitation which might be short in duration and small in scale? Or might these extremes be smoothed during the interpolation process? Did you consider to use a highly-resolved reanalysis data set for comparison reasons?

p.4, line 109-111: "Note that the REGNIE time series are affected by temporal changes in the number of rain gauges considered by the regionalization. For our purpose, the homogeneity of the data are sufficient." - Can you please give a reference here? Did the number of stations change in the analysed period?

p.4, line 115: "[..] appropriate for precipitation statistics [..]" - can you please give a reference here and explain a bit more in detail what was done in the previous literature with the Gumbel distribution.

p.4, line 112-123: General comment: Is this method new? If so, please state here, otherwise, please write something like: "we follow the methodology used in..."

p.4, line 116: R is not explained.

p.5, line 117: Can you please give a reference for the "Method of Moments". If it is also explained in the Wilks-book, maybe you can add the chapter to the reference here.

p.5, chapter 2.1.3: What will you use the data for?

p.5, line 132: what parameters will be taken into account to estimate the "atmospheric conditions"?

p.5/6, chapter 2.1.5: Can you conclude from the radar data, if the thunderstorms rotated? For example by comparing the direction of the mean tracks to the investigated severe thunderstorms?

p.6., chapter 2.2: What fields will you use?

p.6, line 172/173: "[..] but reflects important seasonal differences." - What do you mean here? A figure showing the weather regimes would be nice, at least later in the text, where you analyse the data, you could show the typical patterns of the prevailing regimes.

p.7, chapter 2.4: Is this method new or does it already exist? Please clarify.

p.7, line 197: general comment: The Brunt-Vaisala frequency is smaller in summer, too, due to decreased stability.

p.7, line 199-203: You could add a table to the supplementary material showing the change in associated lightning. Moreover, it would be nice to see this "buffer zone" in the figures.

p.8, line 211-214: Why do you use the wind speed at 500hPa instead of the deep-layer shear? Additionally, deep-layer shear is a widely used variable and the results would be better comparable to the existing literature. I do not understand the motivation here, especially since the authors later in the paper discuss the importance of shear on the organization of thunderstorms.

p.8, line 216: "Overview" - Can you please be more precise, there is another chapter which is also called overview. What is you intention of this whole chapter?

p.8, line 222/223: "The three-week period from 22 May until 12 June was the most active thunderstorm episode with a total of 868 heavy rain, 144 hail, and 145 convective wind gust reports based on the ESWD." - do you mean "the most active thunderstorm episode" in the year 2018 or inother period?

p.8, line 223/224: "An average area of 715,000 km2 was affected by lightning per day" - is that much, what is the average value for Europe?

p.8, line 227/228: "As shown in Figure 2b, most of the severe weather reports came from the western part of France, Benelux, central and southern Germany, and the easternmost part of Austria." - Can you explain the gap in central/eastern France? From your Fig. 8 lifte index was negative, too. Moreover, the mean wind was not much different from western France?

p. 8, line 241: Isn't the number of ESWD reports depending on the number of people reporting events? Is there a difference if you just use some of the quality levels?

p.9, line 252: "low wind speed [..] slow propagation" - You could mention here, that you

will give more details later in the text. While first reading through the text, I wondered if these statements will be verified later or just stated as a fact here?

p.9, line 257: What is meant with "The strength and spatial extent of the lifting forcing varied from day to day, [..]"? Can we see this in one of the figures?

p.9, line 260-273: Just write about the events that are explained in more detail. All other numbers will just lead to confusion and can be seen in the table.

p.10, chapter 3.3: It would be reader-friendly if you explained what the intention of this chapter is. Please give an introductory sentence.

p.10, lines 292-303: It would be a helpful addition if you overlayed the ESWD data. This would make it easier to follow your arguments.

p.11, lines 312-313: Can you please plot the typical patterns of the Zonal regime and the European Blocking.

p.11, lines 315-323/line 330: Can you plot in Fig 6/7a+b additionally to the regimes/sounding data, the lightning activity (out of Fig. 2a) for easier comparisons.

p.12, line 348/349: "Because of the low wind speed in the mid-troposphere, most of the thunderstorms moved very slowly or even became stationary." - The motion of thunderstorms is not necessarily determined by the wind at 500 hPa - can you please give a reference that shows that the storm motion correlates with 500hPa winds.

p. 12, lines 358-360: "The fact that relatively high PV cut-off frequencies expand over a larger region of western Europe underlines that multiple individual PV cut-offs form on the upstream flank of the blocking ridge, and intermittently move across Iberia, France, the British Isles, the North Sea, and Germany [..]" - How do you distinguish between a stationary cut-off low and newly-formed moving ones in Fig. 10?Please clarify.

p. 13, line 396: better: " To estimate the severity of the rainfall with respect to the rainfall climatology, [..]"

p. 13/14, chapter 5.1: I wonder if the return periods are dependent on the REGNIE data and how it is designed. Is it possible to get higher precipitation amounts than observed at the stations? Can you please comment on this?

p. 14, chapter 5.2: If I understand it correctly, the only thing one can directly compare in Fig. 14 - left vs. right boxes-and-whiskers - is the median on the left with the complete box-and-whiskers on the right? Maybe you could add the median of the actual period as an extra symbol to the right box-and whiskers.

p. 14/15, lines 435-448: Although, your main intention is presumably, that the investigated storm period is a rare event. From your text, I could not understand how Fig. 15 was produced. Can you please rewrite the text passage and clarify. What is meant by skip days and why do you use 3 instead of 1 as in the referenced paper? Please explain.

p. 15, lines: 463-466: ". A further relevant condition for the evolution of deep moist convection is the vertical wind shear or, more generally, the wind at mid-tropospheric levels, which is decisive not only for the organizational form, the longevity and thus the severity of the convective storms (e.g., Weisman and Klemp, 1982; Thompson et al., 2007; Dennis and Kumjian, 2017), but also for their propagation (Corfidi, 2003)." - As far as I know, all the cited papers talk about the vertical wind shear, but not about the wind at mid-tropospheric levels (although they might mention storm-relative winds, but this can be quite different from the mid-tropospheric wind). Of course, I can be mistaken, hence, please cite the text passages of the papers, where the mid-tropospheric wind is mentioned in your authors's response.

p. 16, lines 475/476: "[..] air masses were trapped [..]" - Is it possible to show, that the air masses were trapped over several weeks (e.g. by using trajectories)?

p. 16, lines 484-485: "In our investigated case, thunderstorms were often triggered by large-scale lifting associated with upper-level cut-off lows or filaments of high PV that separate from the main PV cut-off" - I am convinced that the cut-off lows provided good

environmental conditions for convection, however I doubt that the cut-off lows triggered the thunderstorms directly. What about (older) outflow boundaries? Can you please comment on that?

p. 16, lines 490-496: Especially since the precipitation amounts are so high, how do you know that the thunderstorms were mainly single cells? Moreover, did you mention at any point in your paper, how you differentiate between single cells and other convective thunderstorm types like multicells? Maybe you can put the radar movies for one of the extreme cases you talked about to the supplemental material?

Figures:

Fig. 2b: Please do not use the rainbow color scale. It is hard to differentiate between some days. Maybe if you switch to a sequential scale, it might be possible to see some temporal clustering? Are there really no events in northern Italy, the Czech republic or Poland?

Fig. 3b: I cannot see any difference between the blue colors here.

Fig. 4: Is it possible to add the locations of the ESWD reports of the associated day to maps?

Fig. 6: It is impossible to differentiate between ZO/SCTr, EuBL/SCBL and AT/GL. Can you add the affected lightning area (from Fig 2a) to the curves.

Fig. 7: Is it possible to add the lightning data from Fig 2a?

Fig. 12: There is no red hatching (in my print it looks black?). Is it possible to add the buffer zone?

Fig. 14: Can you please add the median from the left box-and-whiskers as an extra symbol to the right ones? Please also plot the deep-layer shear.

---

## Author Comment (AC1) · 25 Mar 2020

**1 Review for WCD-2020-1 (RC1 from 11 Feb 2020)**

This study describes an exceptional period of thunderstorm activity in western Europe during summer 2018 and explores the associated synoptic-scale conditions; in particular, the role of blocking and associated upstream cut-off lows. The event is also placed in a climatological context using long-term records from surface stations, upper-air soundings, and reanalysis. This is an interesting, thorough and well-written piece of work, which I have no hesitation in recommending for publication, subject to a few minor revisions as detailed below. I would like to thank the authors for their efforts,

which made for an easy and enjoyable review.

AC: We thank the reviewer for the time taken to review our manuscript and for his useful comments. We are pleased that the reviewer finds the manuscript an interesting, thorough and well-written piece of work.

Specific Comments: 1. My most significant comment relates to your conclusion regarding the slow movement of convective systems during the event. You provide clear evidence (in Fig. 9) that storm motion was, on average, much lower than is typically observed in this region and at this time of year and suggest that this was key to the extreme rainfall totals. However, storm motion is not the only relevant factor for heavy precipitation. As discussed by Doswell et al. (1996), accumulated rainfall depends on two things: average rain rate and total rainfall duration. Slow cell motion contributes to long rainfall durations, but one must also consider system size (in the direction or storm motion) and the existence of back-building convection, both of which may lead to echo training. Rainfall rates must also be considered. Given that you have hourly gauge data and radar observations, it should be possible to assess all of these things. While this might not be practical for the whole study period, you should consider doing so for the most extreme events noted in Table 1. One option would be to add an extra column to the table, providing a brief description of the storm(s) that caused the rainfall totals (including their estimated motion). At the very least, it would be good to demonstrate that the general characteristic of slow storm motion applies to some of the individual extreme events.

AC: The reviewer is right; the slow propagation speed is only one factor for locally high rainfall; we will add a comment to the conclusions. We will also add an additional column in Table 1 with storm motion and size / duration of the convective systems as suggested. However, a detailed analysis of the reasons for heavy rainfall would go beyond the scope of this paper.

2. You state in your abstract that low vertical wind shear "prevented thunderstorms

from developing into severe organized systems". However, reports of hail up to 5 cm in diameter (L234; L239) suggest that this is not entirely true. Clearly, wind shear in certain areas and on certain days was sufficient for the development of organised convection (probably supercells). As such I think this statement needs revising.

AC: That is partly correct; the correct formulation would have been that "low vertical wind shear prevented "very often" thunderstorms from developing into severe organized systems". We will modify this linguistically. Furthermore, we will look again at the cases with 5 cm hail (time/location, associated V500/wind shear) and address this part/exceptions separately. Note, even if there are many days with hail reports (Fig. 2), this does not necessarily mean that high shear was necessary, since hail – but usually smaller hail – is also observed with low shear situations (see Kunz et al., 2020). Note 86 % of the hail events during the long investigation period are <= 3 cm.

3. Please provide some citations for reports of storm impacts (in the opening paragraph of the introduction and section 3.1). These could simply be links to online news or social media reports.

AC: Here, we will put together a few examples from the media (however, mostly in German). (First examples are WetterOnline: https://www.wetteronline.de/extremwetter/unwetterserie-ende-mai-ganze-ortschaften-verwuestet-2018-05-31-us & https://www.wetteronline.de/extremwetter/unwetter-treffen-suedwesten-regenfluten-spuelen-autos-weg-2018-06-01-ju; MDR Sachsen: https://www.mdr.de/sachsen/chemnitz/vogtland/unwetter-sturm-im-vogtland-100.html)

4. The second paragraph of the introduction doesn't really fit and has only limited direct relevance to your study. As such I would suggest removing it (although parts of it could potentially be incorporated elsewhere in the introduction).

AC: In the course of the revision of the introduction (see also comments of Reviewer 2) we will take these into account (delete or include elsewhere).

5. I know that 1981–2010 is a standard 30-year climatology period, but for this study you should consider using the full ERA-Interim record (1979–2017) in order to provide a more complete historical context for the 2018 event. I believe the sounding data will go back this far as well.

AC: Our intention was to use a consistent period of 30 year, which is homogenous in the different analyses in the paper, to calculate the climatological mean, as is common in several studies. When calculating the return periods (Fig. 13/15), where this aspect is important, we have already taken into account longer time series (more or less what was available). We will test this aspect regarding Fig. 14 & Fig. 16 (for 1979/1981–2017) and clarify whether this results in significant differences.

6. On Line 140 you claim that surface-based lifted index (SLI) offers "the best representation of convective environmental conditions in central Europe". I would expect CAPE to provide a more robust measure of surface-based instability, given that it considers the full column rather than just a single level (see discussion in Doswell and Schultz 2006). Certainly it has been shown to usefully discriminate between severe and non-severe convective environments in various parts of the world, including Europe (Pucik et al. 2015; Taszarek et al. 2017). It is also available as a diagnostic from ERA-Interim, so could easily be analysed spatially as well as from the point soundings. I appreciate that repeating all of your instability analysis using CAPE would be time consuming and is likely to show comparable results, so I will not request this. However, you should provide some further justification for why you chose to use SLI over CAPE.

AC: For Europe, there are many studies showing that SLI can be used as well as CAPE (e.g., Huntrieser et al., 1997; Westermayer et al., 2017; Rädler et al., 2018; Sanchez et al., 2009). Some studies also showed that the skill to predict thunderstorms and/or their sub-peril can better as using CAPE (e.g., Kunz, 2007; Mohr and Kunz, 2013; Haklander and van Delden, 2003; Manzato, 2003; ). In addition, CAPE has the disadvantage that its distribution function is skewed and that the CAPE values can be zero (or small) despite (high) instability in the atmosphere. Our experience (in our last

studies) has shown that SLI is more robust in our different analyses, so we prefer to use it. Furthermore, we assume that the key messages in this paper do not change significantly when analyses are performed with CAPE. But we will test this sporadically. In addition, we will add further justification in text as desired.

7. You use a very high reflectivity threshold (55 dBZ) for identifying and tracking convective storms. Such a value is more characteristic of hail than intense rainfall. As such I wonder if the majority of storms went undetected, leading to an unrepresentative velocity estimate. One simple way to check this would be to see if the storms that produce some of the extreme rain accumulations listed in Table 1 were detected. At the very least this should be noted as a limitation of your radar-based analysis.

AC: That's right, a threshold of 55 dBZ prevents weaker cells from entering the sample. However, as we are focusing here on heavy rainfall, this threshold is appropriate; we will explain this in the text. Note that during the study period from 22 May to 12 June the tracking algorithm identified 480 individual storm tracks in Germany, whereas only 84 reports (several from the same storm) are archived in the ESWD. We assume that the largest part of the storm tracks was not associated with hail. However, we will follow the suggestion and we will check the samples listed in Table 1. Furthermore, we will add some comments on this in the next version.

8. It would be good to include a figure showing the different weather regimes discussed in section 2.3 (or, at least, the ZO, EuBL, and AR regimes that dominated during the study period). Perhaps this can be found elsewhere. If so please refer to the specific figure(s) in the relevant paper(s).

AC: The illustration of the Atlantic-European weather regimes can be found in Grams et al., 2017 – however in the Supplementary information (Supplementary Figure 1) and only for the winter season. We will provide the typical patterns (for the summer season) as supplementary material.

9. In discussing the persistence analysis in section 2.5, and the associated results

in section 5.2, I found the reference to "cluster length" confusing, in part because K-Means clustering is used in the analysis of weather regimes. I would change this to just "event duration" or "event persistence".

AC: I can understand your point. However, we have already introduced the wording in PIP16 and I would like to keep it for reasons of consistency. However, you suggestion with "event duration" or "event persistence" would be a good alternative.

10. Could you provide a few more details on the origin of the "basic" and "strict" criteria for SLI and mid-tropospheric wind speed, so that the reader doesn't have to go to PIP16 for this information? Also, I'm not a fan of the notation TH BC and TH SC for these and would argue that they can be eliminated (you can just refer to the basic/strict criterion).

AC: TH BC and TH SC are introduced in PIP16 – this is the reason why we used this notation here. We will add a comment to the definition and check if we can only use the formulation "basic/strict criterion" in the revised script.

11. I recommend using V500 (rather than v_500hPa ) to indicate the 500hPa wind speed.

AC: We will implement this suggestion.

12. I suggest using "total" or "accumulation" when referring to precipitation amounts rather than "sum" (e.g. in Table 1 and on L260–261).

AC: We will implement this suggestion. We will also unify the wording on this.

13. I'm confused as to why you focus the opening paragraph of section 3.3.1 and the first three panels of Fig. 4 on the two weeks before your main study period. In my view it would make more sense for Fig. 4 to show more regular snapshots from the study period, so that the reader can more easily see the temporal evolution described in the subsequent two paragraphs. In particular, it would be good to include one snapshot that shows the second cut-off low (C2) and one close to the end of the study period.

AC: We think it is important to highlight the synoptic situation also prior to the event in order to emphasise that the severe convection during the study period was embedded in a longer lasting unusual large-scale flow situation. The synoptic conditions prior to the event featured blocking, the formation of cut-off lows, and facilitated the advection of warm-moist air into central Europe, which was a key ingredient for the high-impact event. Therefore, we decided to keep the synoptic discussion of May/June along with the characterisation in terms of unusual geopotential height anomalies and IWV (Fig. 5) and the evolution of weather regimes (Fig. 6). As C2 did not strongly affect central Europe we compromised not to show it (now we state in line 296 "(C2, not shown)". Instead we provide a detailed synoptic discussion of the impact of C3 in section 4.

14. In section 5.1, you note that several of the rainfall return period maxima in Fig. 13 "have an almost circular shape with the highest value located in the center" and suggest that this characteristic "reflects the very slow propagation of the thunderstorms". However, could it instead be an artefact of insufficient gauge density? If only one gauge recorded the event, this information would be spread laterally by the gridding procedure, giving the impression of a small circular shape. It might be worth overlaying the gauge locations on this plot to check how many gauges are associated with each maximum.

AC: REGNIE gridded data are based on approx. 2,000 stations (more or less all ground-based observation networks of DWD). Comparing the distribution (cf. Fig. 2b in Rauthe et al., 2013, MZ) with our estimated RPs shows that the majority of all events are captured by several stations and not by only one. However, we will follow the suggestion and will plot a Figure with including all climate station to test your hypothesis and we will include a comment on that (or probably delete the relation to the slow propagation).

15. The description of Fig. 14 at the start of section 5.2 is rather confusing and should be revised. In particular, I found it hard to understand how the distributions for the climatological period were derived.

AC: We will rewrite this description to make this aspect more precise and to make it more understandable and less confusing.

16. Several of the figures could be improved in a few ways. Specifically, I recommend the following changes:

- Fig. 3: The top and bottom rows could arguably be combined. In this case, rather than colouring the symbols on the map by rainfall amount you could just make them blue for > 35 mm/h and red for > 60 mm/3h; then use these same colours for the bar plot (with red bars overlaid on blue bars).

AC: We will combine Figs. 3a and 3c in one histogram (good suggestion), so that the number is also easier to compare. However, we would like to keep Fig. 3b and 3d separate, since this way a spatial representation of the maximum precipitation per station (especially for the 3h panel) is maintained.

- Fig. 7: I don't think it's necessary to state the two thresholds within the plots; this information can be provided in the caption (with reference to the dashed lines).

AC: Personally, I'm a friend of putting a lot of information in figures (if they don't irritate too much), so that – if the figure is used elsewhere – all information is included. However, we can also remove it here – as requested – and provide the info only in the caption of the figure.

- Fig. 12: I would get rid of the hatching showing the objectively identified cut-offs and use a darker contour for the pressure vertical velocity.

AC: We will revise the figure. Definitely we will change the current green line (omega) to a darker and thicker contour. As the second reviewer asks for the buffer zone to be marked in the figure, we will check different variations (with/without hatching PV; with/without buffer zone), which is best suited to illustrate the relevant results.

- Fig. 14: It would be helpful to use different colours for the box-and-whisker plots corresponding to the study period and the 1981–2010 climatology. Use the same colour

convention for Fig. 9.

AC: We will implement this suggestion.

- Fig. 15: In the legend, rather than putting "(2018: N days incl. M skip days)" for each station I would just put "(N/M)" and then explain what these numbers indicate in the caption. So, for example, for Essen you would put "(17/3)" instead of "(2018: 17 days incl. 3 skip days)".

AC: Good suggestion; it reduce double information/text in the figure. We will implement this suggestion.

- Fig. 16: Rather than plotting the percentage difference from the climatological frequency (which is confusing because it is a percentage of a percentage), I recommend expressing this difference in terms of the standard deviation of the climatological frequency. This will highlight whether the 2018 frequencies were exceptional in the context of typical year-to-year variability.

AC: We concur, that a percentage of a percentage would be confusing. However, in Fig. 16 we show in shading the ((absolute frequency in May/June 2018)-climatological frequency May/June 1981-2010)). Thus, over Northern Spain absolute frequencies exceed 50 %, which would be relatively speaking a change of more than 1000 % compared to climatology. We can clarify this in the figure caption. Additionally, we will check the suggestion with the standard deviation of the cut-off low frequency and possibly implement a second panel to Fig. 16.

Technical Corrections:

AC: We will consider and will implement all following (small) technical suggestion and questions.

1. There are a few issues with tenses in the text. For example, the opening sentence of the introduction is written in the present perfect tense (use of "has been"), but should be in the past tense ("was"). The same goes for L261. The last sentence of the opening

paragraph of section 3.3.3 is written in the simple present tense but should be in the past tense.

AC: The manuscript was checked by an editor service; but we will check the tenses in the text again. Thanks for the careful reading.

2. L23: Parentheses are (are not) for references and clarification (saving space) (Robock 2010). Please modify this sentence accordingly.

AC: We will change this as proposed.

3. L36: "...serve to precondition the thermodynamic environment."

AC: We will change this.

17. L39–42: To which event are you referring here: 2018 or 2016? I suggest rewording this paragraph to make this clear. Similarly, you should state explicitly the event you are referring to on L71.

AC: 2016. The whole paragraph is focused on the 2016 event. In the course of the revision of the introduction (see comments of Reviewer 2) we will take these into account and formulate them more clearly.

18. Line 60: Lifting will only lead to the release of CAPE (i.e. convective initiation) if it is sufficient for parcels of air to reach their level of free convection; however, it may still act to destabilise the column (increase CAPE) and erode lids (reduce CIN).

AC: We will rewrite this.

19. Line 90–91: Suggest revising the end of this sentence as follows: "...based on reports from storm chasers, eyewitnesses, voluntary observers, meteorological services, and news media."

AC: We will change this as proposed.

20. L97–98: "... the Météo-France (1223/1935 stations with hourly/daily data)..."

[Figure]

AC: We will change this as proposed.

21. Line 100: These are the national meteorological services of all the countries in your study so I don't think you need this last part of the sentence.

AC: We will delete the last part.

22. Line 132: "wind speed and direction"

AC: We will change this.

23. L146: What is the altitude of the lowest level?

AC: 1 km above ground is the lowest level and 12 km the highest.

24. L199: Change "fewer" to "less".

AC: We will change this.

25. L239–240: Suggest revising this sentence as follows: "Many of the record-breaking 1h and 3h rain totals occurred within this period (see Sect. 3.2)."

AC: We will implement this suggestion.

26. L255–256: "...the latter on the day with the second most ESWD severe weather reports (cf. Sect. 3.1)." Rather than referring to the previous section here please specify the actual date.

AC: We will implement this (31 May).

27. L257–258: You deal with the variations in large-scale forcing for convection later in the paper so I don't think it is necessary to include this here (unless you want to explicitly refer to the relevant sections).

AC: We will include a reference for the later part in the paper.

28. L271: Change "were" to "where".

AC: Thanks. We will change this.

29. L272–273: This sentence is confusing and should be revised.

AC: We will revise this sentence to avoid confusion.

30. L357: Change "vast parts of" to "much of".

AC: We will change this.

31. L362: Change "exemplarily shown for" to "exemplified by".

AC: Good suggestion. We will use this formulation.

32. Line 380–381: Change "and was already mentioned at the end of Section 3.2" to just "(Section 3.2)".

AC: We will change this.

33. L510: Advected where?

AC: That's right, that's too inaccurate. We mean that mostly on the upstream side of the blocking, we observed the advection of warm, moist and unstable air masse favouring thunderstorm development. We will rewrite this and make it clearer.

34. L526: Get rid of "(global/regional)".

AC: We will delete this.

---

## Author Comment (AC2) · 25 Mar 2020

**2 Review for WCD-2020-1 (RC1 from 23 Feb 2020) Overview: The paper presents a case study of a long-lasting thunderstorm series over France/central Europe in May/June 2018 that occurred south of a blocking high. The synoptic situation persisted for several weeks. The thunderstorms were associated with cut-off lows/potential vorticity filaments that formed on the south-west of the blocking high. As a result, numerous severe convective events such as flash floods, hail and wind gusts were recorded. The authors use multiple different data sets and methods to show how the large-scale dynamics contribute to the thunderstorm series and that this event was exceptional.**

AC: We thank the reviewer for the time taken to review our manuscript and for his useful comments.

Overall, I like the author's idea of studying this event from synoptic down to the convective scales. Moreover, it is an interesting case. However, I think the manuscript was difficult to follow and can still be improved considerably. In the current form, it was unfortunately no pleasure to read through the study. My main criticism is that large parts of the paper (chapter 2,3,4, but also in the introduction) read like a collection of single parts which are not really connected with one another. A central theme seems to be missing. I think, the authors should restructure the paper or parts of it and follow a clear path, e.g. from large-scale to the small scale or the other way around. If this is not possible, they should at least clarify the purpose of each (!) chapter at the beginning (as it is done in chapter 5) to facilitate the reading.

AC: We will edit several text passages (linguistic and structural revision), in particular rewrite the introduction, better link the individual chapters and better highlight the interesting point of our study (description of a case study of a thunderstorm episode, which was particularly influenced by large-scale processes (blocking and enhanced cut-off frequency) resulting in the long lasting event). We believe that this will help to improve these points.

Moreover, in my opinion, the writing can be improved, too. Some sentences are too long, which makes the text hard to read. Just write necessary information and just reference to papers that are relevant for your topic. Make the sentences clear and concise. Please connect the single chapters and (sub)sections with one another!

AC: Long sentences are typical for Germans. We will take this criticism into account in the next script version and will endeavour to reduce long sentences, to make sentences clearer and more concise, and to connect the individual chapters better.

I will explain my criticism in more detail in the following:

(i) In the introduction, the authors switch strongly between different topics: first they introduce the case and its impacts in a few sentences. Then they describe convective development due to scale interactions (mainly lifting processes). Afterwards they describe the case again with focus on blocking which is described more general thereafter. In the successive part, the authors explain cut-off lows in the potential vorticity framework. Afterwards they switch back to the topic of blocking. However, these single parts are often unconnected with one another which is confusing for the reader!

AC: We will rewrite the introduction to clarify the points criticised and to better structure aspects necessary for the paper.

(ii) In the data and methods chapter (chapter 2), data sources are often introduced without clarifying why the authors will need the data. At least some overview at the beginning of this section – how the study was designed and/or what data satisfies which purpose – would help the reader tremendously!

AC: We will take this point better into account and integrate it in the relevant parts.

Are there any new methods? Please clarify!

AC: Please, see comments below.

(iii) The same applies for chapters 3,4, and partly 5! Try to connect the single parts, try not to jump unnecessarily between topics. In the current version, it is really confusing for the reader.

AC: We will edit several text passages to better structure aspects necessary for the individual chapters.

(iv) With respect to the methods my main concern is the usage of the 500hPa-wind instead of vertical wind shear. At least additionally analyzing shear had the advantage that your work can be compared more easily to the existing literature of convective events.

AC: We had made a conscious decision to show V500 and not the shear (although we had already examined both). One reason for V500 was the connection with the propagation speed of the cells (see also the comment on propagation speed below), since our focus in the work was not to analyse the organizational structure of the cells using wind shear. Since we have observed very low near-surface wind speeds in several cases, the wind at mid-tropospheric levels is more or less identical to wind shear. However, we will check this again. Perhaps, we will add the results for wind shear to one of the figures (where it will also become clear that there are only very few differences). A first idea would be Fig. 7, 8, or 14.

(v) Furthermore, I am missing evidence, that the thunderstorms have been single cells rather than multicells, MCS or slow-moving (HP) supercells.

AC: Separating the convective cells among their organization form might be an interesting issue, but is far beyond the content and aim of our paper. Furthermore, we do not have an algorithm that allows us to adequately identify this. Instead, we suggest weakening the statement about the single cells. In addition, we will check the cases in 2018 (especially those mentioned in table 1) with radar images and add some comments in the text.

Specific minor comments:

Abstract, p.1 line 2: "80mm" - what is the temporal range? a few hours?

AC: See Table 1 Theoretically several time scales are possible (1 h, but also in 3 h). We will add an information.

p.2, line 25-36: I am missing the general ingredients of convection here: instability, moisture, lift and shear. The ingredients-based concept is first mentioned in the Discussion chapter (chapter 6), I think it would be fitting in the introduction, too. Moreover e.g. Markowski and Richardson, 2010 (their chapter 10.4) & Doswell III, C. A., Brooks, H. E., & Maddox, R. A. (1996): Flash flood forecasting: An ingredients-based methodology. Weather and Forecasting, 11(4), 560-581. treat flash flood events. Especially in the Markowski and Richardson book, you can find a very similar synoptic pattern that led to flash flood events in the US (please refer to the publications mentioned therein).

AC: We will include the suggested literature and some comments on this.

p.2, line 34: "all these mechanisms" - which ones are meant here?

AC: All those mentioned before; this paragraph will be completely revised anyway. We will take care not to make this point too general.

p.2, lines 51-61: There are some publications concerning the PV framework and convection: e.g. Russell A, Vaughan G, Norton EG. 2012. Large-scale potential vorticity anomalies and deep convection. Q. J. R. Meteorol. Soc. 138: 1627–1639. DOI:10.1002/qj.1875; Morcrette CJ, Lean H, Browning KA, Nicol J, Roberts N, Clark PA, Russell A, Blyth AM. 2007. Combination of mesoscale and synoptic mechanisms for triggering of an isolated thunderstorm: a case study of CSIP IOP 1. Mon. Weather Rev. 135: 3728–3749. Can you please put your work in context with the existing literature?

AC: Thank you for your literature suggestions; we will put these works in the context of our study (or introduction).

p.3, line 62: "A connection between atmospheric blocking and heavy precipitation events..." - Why again blocking? The sentence is almost identical to that on page 2, lines 49-50. Why don't you merge these parts?

AC: We will rewrite the introduction.

p.3, line 68-70: "[..] such situations are usually associated with weak wind speeds at mid-tropospheric levels (cf. PIP16), so that thunderstorms become almost stationary and usually do not develop into organized structures such as large mesoscale convective systems or supercells." – First: where is the wind weak? in the high, the low, at the western flanks?

AC: Over the investigation area in the mentioned paper, we will clarify this.

Second: What about HP-supercells (high precipitation supercells)? Can you please comment on HP-supercells.

AC: HP-supercells might have occurred. As we haven't investigated that, I'd suggest to change the sentence into "usually associated with weak wind speed at mid-tropospheric levels and, thus, weak vertical wind shear with the consequence that thunderstorms become almost stationary and rarely develop into organized convective systems."

p.3, line 83: What do you mean with "secondary effects"? Please elaborate.

AC: Their sub-perils (e.g., hail, heavy rain, wind gusts); we will clarify this.

p.3, line 85: "(May/June)" - These are the whole months (1.5-30.6)? It is confusing since you already stated two different periods in the text before.

AC: Yes, the whole month; we will add this information (1 May to 30 June).

p.4, line 87-93: Please clarify what the purpose of the ESWD data is. Do you use different quality levels or all? Why don't you show the reports also in e.g. Belgium or Italy?

AC: The purpose of the ESWD data is to show the sub-perils associated with the thunderstorms, and that these were preferably heavy rain events. We use all data above QC0+ (we will add this). ESWD Data from Belgium have already been used (see Fig. 2). Maybe you mean the precipitation data from Belgium (here we were not able to get any). Data from Italy are not included because Italy is not included in our study area (see L: 81/82 "The study area includes parts of central and western Europe - France, Benelux (Belgium, Netherlands, Luxembourg), Germany, Switzerland and Austria (see Fig. 1) – for which data were available").

p.4, line 88: It is good to know that the ESWD collects data about heavy rain, hail and

wind gusts. However, what data did you use for the analysis?

AC: All these three sub-perils. We will add a comment.

p.4, line 90: better: "[..] mainly based on reports of storm chasers, [..]"

AC: We will correct this.

p.4, line 108: Is there a description of the REGNIE data in English for non-Germans, too?

AC: Not really, but about HYRAS, which uses the same methodology and which is already cited (Rauthe et al., 2013).

p.4, line 104-111: Why did you decide to use the REGNIE data. The data seems to interpolate measured precipitation on a regular grid. Is the REGNIE data suitable to analyse extreme convective precipitation which might be short in duration and small in scale? Or might these extremes be smoothed during the interpolation process? Did you consider to use a highly-resolved reanalysis data set for comparison reasons?

AC: We used REGNIE data only for the estimation of return periods because of their long-term availability of approx. 70 years, and the large number of approx. 2,000 climate stations used in the regionalization method. RADOLAN data (merger between Radar and station data) would be a better choice, but are available only for 20 years and, thus, not suitable to estimate return periods. Reanalysis also tend to underestimate precipitation totals. We will explain in the new version of the text, why we used REGNIE data (in addition to other station data).

p.4, line 109-111: "Note that the REGNIE time series are affected by temporal changes in the number of rain gauges considered by the regionalization. For our purpose, the homogeneity of the data are sufficient." – Can you please give a reference here? Did the number of stations change in the analysed period?

AC: We will included the reference Rauthe et al. (2013). Unfortunately, there is no

recent reference available; also the exact number of stations for the regionalization is not known (it was approx. 2,000 stations in 2011).

p.4, line 115: "[..] appropriate for precipitation statistics [..]" - can you please give a reference here and explain a bit more in detail what was done in the previous literature with the Gumbel distribution.

AC: That's right, "precipitation statistics" is a little too generic. We will change this sentence into: "The Fisher-Tippett Type I distribution, also known as the Gumbel distribution (Gumbel, 1958; Wilks, 2006), has been extensively used in various fields including hydrology for modelling extreme events, i.e. to estimate statistical return periods or return values (Sivapalan and Blöschl, 1998; Rasmussen and Gautam, 2003). The Gumbel cumulative distribution function (CDF) is given by:"

p.4, line 112-123: General comment: Is this method new? If so, please state here, otherwise, please write something like: "we follow the methodology used in..."

AC: No, is not new. We will clarify this and add a reference (e.g. Wilks, 2006).

p.4, line 116: R is not explained.

AC: That's correct. We will add this (R is the investigated variable, here precipitation values.)

p.5, line 117: Can you please give a reference for the "Method of Moments". If it is also explained in the Wilks-book, maybe you can add the chapter to the reference here.

AC: Yes; it is explained in Wilks (2006) in Chapter 4 (Parametric Probability Distributions).

p.5, chapter 2.1.3: What will you use the data for?

AC: To show the affected area in our study region. Lightning data are the best direct observation data for the thunderstorm detection, as they provide the best spatial (complete) coverage (but without reference to the respective sub-peril and no direct relation

to intensity of the event. For this, we use ESWD data and rain measurements. We will add a comment on this.

p.5, line 132: what parameters will be taken into account to estimate the "atmospheric conditions"?

AC: As already mentioned the SLI; others are V500 (perhaps shear). We will add a comment on this.

p.5/6, chapter 2.1.5: Can you conclude from the radar data, if the thunderstorms rotated? For example by comparing the direction of the mean tracks to the investigated severe thunderstorms?

AC: For proper detection of rotation in radar data you need the Dual-Doppler wind fields, which we do not have. And even with that the detection or rotation is very difficult (we didn't even detect rotation in the radar data of the severe supercell on 28 July 2013; cf. Kunz et al., 2018). Because the track direction depends – in addition to vertical pressure disturbances - on various effects such as the vertical extent of the cells relative to the wind shear or the formation of new cells (particularly in case of multicells), a comparison between the two as suggested would not allow to give any conclusions about the rotation of the cells.

p.6., chapter 2.2: What fields will you use?

AC: We will add a more detailed description (e.g., SLI, V500, PV, Z500, IWV, . . .).

p.6, line 172/173: "[..] but reflects important seasonal differences." - What do you mean here?

AC: Classical weather regime definitions typically distinguish summer and winter regimes. Our year-round definition contains these seasonal different patterns and therefore has more regimes than classical definitions. Still these patterns kann occur in any season and are important for local weather conditions.

A figure showing the weather regimes would be nice, at least later in the text, where you analyse the data, you could show the typical patterns of the prevailing regimes.

AC: The illustration of the Atlantic-European weather regimes can be found in Grams et al., 2017 – in the Supplementary information (Supplementary Figure 1) for the winter season. We will provide the typical patterns (for the summer season) as supplementary material for faster availability.

p.7, chapter 2.4: Is this method new or does it already exist? Please clarify.

AC: Similar approaches have been used to relate surface weather to weather objects in earlier work (e.g., Pfahl and Wernli 2012, 2014; Pfahl et al. 2014). We will clarify this now in the text. Still as far as we know this is the first study matching lightning data to cut-off cyclones. Pfahl, S., and H. Wernli, 2012: Quantifying the Relevance of Cyclones for Precipitation Extremes. J. Climate, 25, 6770–6780, doi:10.1175/JCLI-D-11-00705.1. Pfahl, S., and H. Wernli, 2012: Quantifying the relevance of atmospheric blocking for co-located temperature extremes in the Northern Hemisphere on (sub-)daily time scales. Geophys. Res. Lett., 39, L12807, doi:10.1029/2012GL052261. Pfahl, S., E. Madonna, M. Boettcher, H. Joos, and H. Wernli, 2014: Warm Conveyor Belts in the ERA-Interim Dataset (1979–2010). Part II: Moisture Origin and Relevance for Precipitation. Journal of Climate, 27, 27–40, doi:10.1175/JCLI-D-13-00223.1.

p.7, line 197: general comment: The Brunt-Vaisala frequency is smaller in summer, too, due to decreased stability.

AC: Here we aim to provide a physical justification for the scale of our buffer radius not an exact estimation which would depend on each specific case. We think our scale analysis yields a reasonable estimate of a remote influence and that seasonal variation in stability would not change the order of magnitude. In addition, we tested the sensitivity to a range of buffer radii with now impact on our qualitative interpretation.

p.7, line 199-203: You could add a table to the supplementary material showing the

change in associated lightning.

AC: Good suggestion; we will include a table in the supplementary materials.

Moreover, it would be nice to see this "buffer zone" in the figures.

AC: However, showing the additional buffer zone can be problematic – it is probably only possible as an example. We will check this.

p.8, line 211-214: Why do you use the wind speed at 500hPa instead of the deep-layer shear? Additionally, deep-layer shear is a widely used variable and the results would be better comparable to the existing literature. I do not understand the motivation here, especially since the authors later in the paper discuss the importance of shear on the organization of thunderstorms.

AC: We had made a conscious decision to show V500 and not the shear (although we had already examined both). One reason for V500 was the connection with the propagation speed of the cells (see also the comment on propagation speed below), since our focus in the work was not to analyse the organizational structure of the cells using wind shear. Since we have observed very low near-surface wind speeds in several cases, the wind at mid-tropospheric levels is more or less identical to wind shear. However, we will check this again. Perhaps, we will add the results for wind shear to one of the figures (where it will also become clear that there are only very few differences). A first idea would be Fig. 7 or 8.

p.8, line 216: "Overview" - Can you please be more precise, there is another chapter which is also called overview. What is you intention of this whole chapter?

AC: A first description if the event episode in 2018 based on direct observation (lightning data, ESWD) to demonstrate the severity. We will rename the section title. Perhaps, we will combine the Section with Section 3.3.2 and called it "(Overview of) direct observations" during the thunderstorm episode.

p.8, line 222/223: "The three-week period from 22 May until 12 June was the most

active thunderstorm episode with a total of 868 heavy rain, 144 hail, and 145 convective wind gust reports based on the ESWD." - do you mean "the most active thunderstorm episode" in the year 2018 or inother period?

AC: During our extended study period (1 May to 20 June) or also in May / June 2018. We will clarify this point.

p.8, line 223/224: "An average area of 715,000 km2 was affected by lightning per day" - is that much, what is the average value for Europe?

AC: Yes, this is twice the area of Germany per day, that's much! Do you really think it's necessary to give an average value? I'd suggest to include the relation to the area of Germany.

p.8, line 227/228: "As shown in Figure 2b, most of the severe weather reports came from the western part of France, Benelux, central and southern Germany, and the easternmost part of Austria." - Can you explain the gap in central/eastern France? From your Fig. 8 Lifted index was negative, too. Moreover, the mean wind was not much different from western France?

AC: This could depend on the availability of storm chasers, eyewitnesses, or voluntary observers. In addition, the orography could also have an influence in Central France (Massif Central) on thunderstorm activity and/or the reports possibilities. We will verify this by investigating the (spatial) lightning density (or thunderstorm days) during the whole study period/area and add a comment.

p. 8, line 241: Isn't the number of ESWD reports depending on the number of people reporting events? Is there a difference if you just use some of the quality levels?

AC: Right. And the quality levels can't fix this. The reports in the ESWD are absolutely controlled by the activity of the different people. Here, the population density plays a significant role or where the most active "reporters" live and what their area of investigation is. There are very severe events, which have smaller number of reports that

less severe events. But what/how exactly defines severe events?

p.9, line 252: "low wind speed [..] slow propagation" - You could mention here, that you will give more details later in the text. While first reading through the text, I wondered if these statements will be verified later or just stated as a fact here?

AC: We will follow the suggestion and add a reference.

p.9, line 257: What is meant with "The strength and spatial extent of the lifting forcing varied from day to day, [..]"? Can we see this in one of the figures?

AC: Not directly. We will delete the sentence.

p.9, line 260-273: Just write about the events that are explained in more detail. All other numbers will just lead to confusion and can be seen in the table.

AC: Ok, we will follow the suggestion.

p.10, chapter 3.3: It would be reader-friendly if you explained what the intention of this chapter is. Please give an introductory sentence.

AC: We will do this (between Sect. 3.3 and Sect. 3.3.1).

p.10, lines 292-303: It would be a helpful addition if you overlayed the ESWD data. This would make it easier to follow your arguments.

AC: We will check whether this is graphically implementable (e.g., only precipitation reports as small points for each time step/panel).

p.11, lines 312-313: Can you please plot the typical patterns of the Zonal regime and the European Blocking.

AC: We will provide the typical patterns (for the summer season) as supplementary material.

p.11, lines 315-323/line 330: Can you plot in Fig 6/7a+b additionally to the regimes/sounding data, the lightning activity (out of Fig. 2a) for easier comparisons.

AC: We will implement this.

p.12, line 348/349: "Because of the low wind speed in the mid-troposphere, most of the thunderstorms moved very slowly or even became stationary." - The motion of thunderstorms is not necessarily determined by the wind at 500 hPa – can you please give a reference that shows that the storm motion correlates with 500 hPa winds.

AC: The propagation of convective cells is driven by various factors such as gust-front lifting in case of multicells and vertical pressure gradients extending over a deep in case of supercells. For single cells, however, no such processes occur or are relevant and the propagation is related to the mean flow, i.e. the vertically average winds. We took the 500 hPa wind as a proxy because proper determination of the vertical extent – even though possible – would be out of the context of this paper. We will modify this statement and include a reference. For example Houston and Wilhelmson (2012) & Markowski and Richardson, 2010). @Article{houston12, author = {Houston, Adam L and Wilhelmson, Robert B}, title = {The impact of airmass boundaries on the propagation of deep convection: A modeling-based study in a high-CAPE, low-shear environment}, journal = {Mon. Wea. Rev.}, year = {2012}, volume = {140}, pages = {167–183}, doi = {doi:10.1175/MWR-D-10-05033.1},}

p. 12, lines 358-360: "The fact that relatively high PV cut-off frequencies expand over a larger region of western Europe underlines that multiple individual PV cut-offs form on the upstream flank of the blocking ridge, and intermittently move across Iberia, France, the British Isles, the North Sea, and Germany [..]" - How do you distinguish between a stationary cut-off low and newly-formed moving ones in Fig. 10? Please clarify.

AC: We concur that this statement was misunderstandable. We here only refer in the first half of the sentence to Fig. 10 (now reference included after "… western Europe (Fig. 10) underlines that …". The occurrence of multiple cut-offs was explained in the synoptic overview Fig. 4. The references "(see Fig. 4)" is now earlier in the sentence "…of the blocking ridge (see Fig. 4), and intermittently …"

p. 13, line 396: better: "To estimate the severity of the rainfall with respect to the rainfall climatology, [..]"

AC: We will change this.

p. 13/14, chapter 5.1: I wonder if the return periods are dependent on the REGNIE data and how it is designed. Is it possible to get higher precipitation amounts than observed at the stations? Can you please comment on this?

AC: Yes of course, all extreme value estimates depend upon the used data set. REGNIE certainly underestimate the precipitation peaks, but this is the case for both the observation period and the reference period of 50 years. We will add a comment on this.

p. 14, chapter 5.2: If I understand it correctly, the only thing one can directly compare in Fig. 14 - left vs. right boxes-and-whiskers - is the median on the left with the complete box-and-whiskers on the right? Maybe you could add the median of the actual period as an extra symbol to the right box-and whiskers.

AC: We will implement the suggestion in Fig. 14.

p. 14/15, lines 435-448: Although, your main intention is presumably, that the investigated storm period is a rare event. From your text, I could not understand how Fig. 15 was produced. Can you please rewrite the text passage and clarify.

AC: We will rewrite this description to make it more understandable.

What is meant by skip days and why do you use 3 instead of 1 as in the referenced paper? Please explain.

AC: One skip day per (started) week; already PIP16 used 2 skip days for two weeks. Since in this study, the persistence is longer – at least three weeks – three skip days are possible.

p. 15, lines: 463-466: "A further relevant condition for the evolution of deep moist

convection is the vertical wind shear or, more generally, the wind at mid-tropospheric levels, which is decisive not only for the organizational form, the longevity and thus the severity of the convective storms (e.g., Weisman and Klemp, 1982; Thompson et al., 2007; Dennis and Kumjian, 2017), but also for their propagation (Corfidi, 2003)." - As far as I know, all the cited papers talk about the vertical wind shear, but not about the wind at mid-tropospheric levels (although they might mention storm-relative winds, but this can be quite different from the mid-tropospheric wind). Of course, I can be mistaken, hence, please cite the text passages of the papers, where the mid-tropospheric wind is mentioned in your authors's response.

AC: We will clarify that when in cases with very low near-surface wind speed the wind at mid-tropospheric levels is more or less identical to the wind shear. This is clumsily expressed in the sentence quoted.

p. 16, lines 475/476: "[..] air masses were trapped [..]" - Is it possible to show, that the air masses were trapped over several weeks (e.g. by using trajectories)?

AC: Good suggestion; we will check this (by working with the Lagrangian Analysis Tool, LAGRANTO from Wernli & Davies, 1997) and comment/include the results. Wernli, H., H. C. Davies (1997): A Lagrangian-based analysis of extratropical cyclones. I: The method and some applications, Q. J. R. Meteorol. Soc., 123, 467–489, doi:10.1002/qj.49712353811.

p. 16, lines 484-485: "In our investigated case, thunderstorms were often triggered by large-scale lifting associated with upper-level cut-off lows or filaments of high PV that separate from the main PV cut-off" - I am convinced that the cut-off lows provided good environmental conditions for convection, however I doubt that the cut-off lows triggered the thunderstorms directly. What about (older) outflow boundaries? Can you please comment on that?

AC: That's correct. Thunderstorms are triggered by a variety of mechanisms. Large-scale uplift by itself won't bring air-packets up to LFC height, speeds are too low (resp.

time scales would be too long for that). Rather, it is the decrease in CIN and the increase in CAPE that are relevant here. We will specify the wording, e.g., we will not speak of trigger (in the direct sense), but of large-scale conditions.

p. 16, lines 490-496: Especially since the precipitation amounts are so high, how do you know that the thunderstorms were mainly single cells? Moreover, did you mention at any point in your paper, how you differentiate between single cells and other convective thunderstorm types like multicells? Maybe you can put the radar movies for one of the extreme cases you talked about to the supplemental material?

AC: Only by visual observation from radar data (we do not have an algorithm which allow to identify the various organisational structures (see also comment above (v)). But we will comment the cases in Table 1 (by naming the type observed). Furthermore, we will check how many days (during the study period) we observed mainly single cells (but also only visually).

Figures:

Fig. 2b: Please do not use the rainbow color scale. It is hard to differentiate between some days. Maybe if you switch to a sequential scale, it might be possible to see some temporal clustering?

AC: We will modify the colorbar.

Are there really no events in northern Italy, the Czech republic or Poland?

AC: Data from Northern Italy, the Czech republic or Poland are not included because these are not part of our study area (Homogeneity reasons) (see L: 81/82 "The study area includes parts of central and western Europe - France, Benelux (Belgium, Netherlands, Luxembourg), Germany, Switzerland and Austria (see Fig. 1) – for which data were available").

Fig. 3b: I cannot see any difference between the blue colors here.

AC: We will modify the colorbar.

Fig. 4: Is it possible to add the locations of the ESWD reports of the associated day to maps?

AC: We will check whether this is graphically implementable (e.g., only precipitation reports as small points for each time step/panel).

Fig. 6: It is impossible to differentiate between ZO/SCTr, EuBL/SCBL and AT/GL.

AC: We will change the colours to make these regimes better distinguishable in Figure 6.

Can you add the affected lightning area (from Fig 2a) to the curves.

AC: We will implement this.

Fig. 7: Is it possible to add the lightning data from Fig 2a?

AC: We will implement this.

Fig. 12: There is no red hatching (in my print it looks black?). Is it possible to add the buffer zone?

AC: Oh, sorry in an older version of the Figure the PV on the 325 K isentropic surface was red. Thanks, we will change this. However, showing the buffer zone can be problematic / to much information in one figure– it is probably only possible as an example. We will check this (for example, a contour filled with small points).

Fig. 14: Can you please add the median from the left box-and-whiskers as an extra symbol to the right ones?

AC: We will implement the suggestion in Fig. 14.

Please also plot the deep-layer shear.

AC: The point with the deep layer wind shear we will check if there is an added value

in this figure (or is the same/to similar; see comments above/beginning if the review)
* * *

---

## Author Response (AR1)

**#1 Review for WCD-2020-1 (RC1 from 11 Feb 2020)**

This study describes an exceptional period of thunderstorm activity in western Europe during summer 2018 and explores the associated synoptic-scale conditions; in particular, the role of blocking and associated upstream cut-off lows. The event is also placed in a climatological context using long-term records from surface stations, upper-air soundings, and reanalysis.

This is an interesting, thorough and well-written piece of work, which I have no hesitation in recommending for publication, subject to a few minor revisions as detailed below. I would like to thank the authors for their efforts, which made for an easy and enjoyable review.

AC: We thank the reviewer for the time taken to review our manuscript and for the useful comments. We are pleased that the reviewer finds the manuscript an interesting, thorough and well-written piece of work.

**Specific Comments:**

1. My most significant comment relates to your conclusion regarding the slow movement of convective systems during the event. You provide clear evidence (in Fig. 9) that storm motion was, on average, much lower than is typically observed in this region and at this time of year and suggest that this was key to the extreme rainfall totals. However, storm motion is not the only relevant factor for heavy precipitation. As discussed by Doswell et al. (1996), accumulated rainfall depends on two things: average rain rate and total rainfall duration. Slow cell motion contributes to long rainfall durations, but one must also consider system size (in the direction or storm motion) and the existence of back-building convection, both of which may lead to echo training. Rainfall rates must also be considered. Given that you have hourly gauge data and radar observations, it should be possible to assess all of these things. While this might not be practical for the whole study period, you should consider doing so for the most extreme events noted in Table 1. One option would be to add an extra column to the table, providing a brief description of the storm(s) that caused the rainfall totals (including their estimated motion). At the very least, it would be good to demonstrate that the general characteristic of slow storm motion applies to some of the individual extreme events.
   AC: We responded this point of criticism by reviewing all German events in Table 1 again with RADOLAN data (merger between radar and station data), so that we were able to investigate the rain rate and duration in more detail (see Supplement Sect. 4 and Supplementary Figure 5). We added these values to Table 1, which impressively shows the particular high rain rate (e.g., 50 min over 60 mm/h for Dietenhofen). Furthermore, we included a discussion in Sect. 5 on that (also citing Doswell et al., 1996). Additionally, we added further information based on the TRACE3D algorithm in Table 1, which shows that many of the cells has a propagation speed below 7 m/s (only one of the sample has a propagation speed of more than 10 m/s). Note that due to data availability, these detailed analyses were only available for Germany.

2. You state in your abstract that low vertical wind shear "prevented thunderstorms from developing into severe organized systems". However, reports of hail up to 5 cm in diameter (L234; L239) suggest that this is not entirely true. Clearly, wind shear in certain areas and on certain days was sufficient for the development of organised convection (probably supercells). As such I think this statement needs revising.
   AC: The correct formulation would have been that " the low-pressure gradient led predominantly to weak flow conditions in the mid-troposphere and thus to low vertical wind shear that prevented thunderstorms from developing into severe organized systems." We changed that.
   Furthermore, we added a comment on the 5 cm hail stone (during the discussion of the ESWD reports): "…the wind shear values over the study area were predominantly very low. Individual cases with hail stones of 5 cm were feasible, because in the border area of our study area above the Pyrenees twice areas with high shear (up to 20 m/s) was transported, which led to the large hail in southwest France (26 May/9 June) or southern Germany (11 June). However, these were exceptional cases."
   Note, even if there are many days with hail reports (Fig. 2), this does not necessarily mean that high shear was necessary, since hail – but usually smaller hail – is also observed with low shear situations (see Kunz et al., 2020). Note 86 % of the hail events during the long investigation period are <= 3 cm.

2. Please provide some citations for reports of storm impacts (in the opening paragraph of the introduction and section 3.1). These could simply be links to online news or social media reports.
AC: We included some references for the storm impacts based on WetterOnline and DWD online media reports (however, only in German):
Tornado wütet bei Viersen: Dutzende Häuser stark beschädigt (17.05.2018)
https://www.wetteronline.de/extremwetter/tornado-wuetet-bei-viersen-dutzende-haeuser-stark-beschaedigt-2018-05-17-tv
Unwetterserie Ende Mai: Ganze Ortschaften verwüstet
https://www.wetteronline.de/extremwetter/unwetterserie-ende-mai-ganze-ortschaften-verwuestet-2018-05-31-us
Unwetterserie im Juni: Überflutungen und Hagelmassen
https://www.wetteronline.de/extremwetter/unwetterserie-im-juni-ueberflutungen-und-hagelmassen-2018-06-14-js
Schadensrückblick des Deutschen Wetterdienstes: Gefährliche Wetterereignisse und Wetterschäden in Deutschland 2018:
https://www.dwd.de/DE/presse/pressemitteilungen/DE/2018/20181213_schadensrueckblick2018_news.html

3. The second paragraph of the introduction doesn't really fit and has only limited direct relevance to your study. As such I would suggest removing it (although parts of it could potentially be incorporated elsewhere in the introduction).
AC: We fundamentally revised the introduction. On this occasion, there were some textual restructuring. The second paragraph was also rewritten and shortened. The aspect of thunderstorm development (we moved the ingredient-based theory from Sect. 6 into the introduction as suggested by the another reviewer) and the scale interactions (between local and large-scale) was still considered; but the link to teleconnection was removed. The latter is not relevant for this paper, and thus rather confusing at this point.

4. I know that 1981–2010 is a standard 30-year climatology period, but for this study you should consider using the full ERA-Interim record (1979–2017) in order to provide a more complete historical context for the 2018 event. I believe the sounding data will go back this far as well.

AC: We wanted to use a consistent period of 30 years, which is homogenous in the different analyses in the paper calculating the climatological mean, as is common in several studies. When calculating the return periods (Fig. 13/15), where this aspect is important, we have already taken into account longer time series (more or less what was available).
We tested this aspect regarding Fig. 14 & Fig. 16 (for 1981-2017). However, no significant differences were found, meaning that we kept to the previous period with a fixed annual period of 30 years.

5. On Line 140 you claim that surface-based lifted index (SLI) offers "the best representation of convective environmental conditions in central Europe". I would expect CAPE to provide a more robust measure of surface-based instability, given that it considers the full column rather than just a single level (see discussion in Doswell and Schultz 2006 ). Certainly it has been shown to usefully discriminate between severe and non-severe convective environments in various parts of the world, including Europe (Pucik et al. 2015; Taszarek et al. 2017). It is also available as a diagnostic from ERA-Interim, so could easily be analysed spatially as well as from the point soundings. I appreciate that repeating all of your instability analysis using CAPE would be time consuming and is likely to show comparable results, so I will not request this. However, you should provide some further justification for why you chose to use SLI over CAPE.

AC: For Europe, there are many studies showing that SLI can be used as well as CAPE (e.g., Huntrieser et al., 1997; Westermayer et al., 2017; Rädler et al., 2018; Sanchez et al., 2009). Some studies also showed that the skill to predict thunderstorms and/or their sub-peril can better as using CAPE (e.g., Kunz, 2007; Mohr and Kunz, 2013; Haklander and van Delden, 2003; Manzato, 2003; ). In addition, CAPE has the disadvantage that its distribution function is skewed and that the CAPE values can be zero (or small) despite (high) instability in the atmosphere. Our experience (in our last studies) has shown that SLI is more robust in our different analyses, so we prefer to use it.
We added a comment on this. Additionally, we observed that the key messages in this paper do not change significantly when analyses are performed with CAPE.

6. You use a very high reflectivity threshold (55 dBZ) for identifying and tracking convective storms. Such a value is more characteristic of hail than intense rainfall. As such I wonder if the majority of storms went undetected, leading to an unrepresentative velocity estimate. One simple way to check this would be to see if the storms that produce some of the extreme rain accumulations listed in Table 1 were detected. At the very least this should be noted as a limitation of your radar-based analysis.
AC: That's right, a threshold of 55 dBZ prevents weaker cells from entering the sample. However, as we are focusing here on heavy rainfall, this threshold is appropriate: We explained this in the text. "Note that we have only used tracking to estimate the propagation speed and direction of the cells (Sect. 3 and Sect. 5.1). Even if weaker cells are not detected using the 55 dBZ thresholds, it can be assumed that they cannot move at higher speeds."
In addition, we checked if the German events in Table 1 can be linked with a track derived with TRACE3D and included the information of track length, propagation speed, and total track area in the Table. However, two tracks could not be identified by TRACE3D due to the overlapping of several cells, which were relatively quasi-stationary.

7. It would be good to include a figure showing the different weather regimes discussed in section 2.3 (or, at least, the ZO, EuBL, and AR regimes that dominated during the study period). Perhaps this can be found elsewhere. If so, please refer to the specific figure(s) in the relevant paper(s).
AC: The illustration of the Atlantic-European weather regimes can be found in Grams et al., 2017 – however in the Supplementary information (Supplementary Figure 1); but only for the winter season, but which are very similar to the summer season.
We now provided the typical patterns (for May/June) as supplementary material.

8. In discussing the persistence analysis in section 2.5, and the associated results in section 5.2, I found the reference to "cluster length" confusing, in part because K-Means clustering is used in the analysis of weather regimes. I would change this to just "event duration" or "event persistence".
AC: We followed the suggestion and changed the formulation "cluster length" in "event persistence".

9. Could you provide a few more details on the origin of the "basic" and "strict" criteria for SLI and mid-tropospheric wind speed, so that the reader doesn't have to go to PIP16 for this information? Also, I'm not a fan of the notation TH BC and TH SC for these and would argue that they can be eliminated (you can just refer to the basic/strict criterion).
AC: TH BC and TH SC are introduced in PIP16 – this is the reason why we used this notation here.
We added a comment to the definition "Both thresholds were originally determined by choosing the maximum of the daily minima in the case study to capture the prevailing (exceptional) atmospheric conditions." and deleted the abbreviations/notation TH BC and TH SC in the text.

10. I recommend using V500 (rather than v 500hPa ) to indicate the 500hPa wind speed.
AC: We implemented this suggestion.

11. I suggest using "total" or "accumulation" when referring to precipitation amounts rather than "sum" (e.g. in Table 1 and on L260–261).

AC: We implemented this suggestion and also unify the wording on this.

12. I'm confused as to why you focus the opening paragraph of section 3.3.1 and the first three panels of Fig. 4 on the two weeks before your main study period. In my view it would make more sense for Fig. 4 to show more regular snapshots from the study period, so that the reader can more easily see the temporal evolution described in the subsequent two paragraphs. In particular, it would be good to include one snapshot that shows the second cut-off low (C2) and one close to the end of the study period.
AC: We think it is important to highlight the synoptic situation also prior to the event in order to emphasise that the severe convection during the study period was embedded in a longer lasting unusual large-scale flow situation. The synoptic conditions prior to the event featured blocking, the formation of cut-off lows, and facilitated the advection of warm-moist air into central Europe, which was a key ingredient for the high-impact event. Therefore, we decided to keep the synoptic discussion of May/June along with the characterisation in terms of unusual geopotential height anomalies and IWV (Fig. 5) and the evolution of weather regimes (Fig. 6). As C2 did not strongly affect central Europe we compromised not to show it (now we state in line 296 "(C2, not shown)". Instead we provide a detailed synoptic discussion of the impact of C3 in section 4. We made this aspect clearer in the text.

13. In section 5.1, you note that several of the rainfall return period maxima in Fig. 13 "have an almost circular shape with the highest value located in the center" and suggest that this characteristic "reflects the very slow propagation of the thunderstorms". However, could it instead be an artefact of insufficient gauge density? If only one gauge recorded the event, this information would be spread laterally by the gridding procedure, giving the impression of a small circular shape. It might be worth overlaying the gauge locations on this plot to check how many gauges are associated with each maximum.
AC: REGNIE gridded data are based on approx. 2,000 stations (more or less all ground-based observation networks of DWD; see black points in the Fig.).

[Figure]

Comparing the stations with our estimated RPs shows that the majority of all events are captured by several stations and not by only one. We added a comment on that.

14. The description of Fig. 14 at the start of section 5.2 is rather confusing and should be revised. In particular, I found it hard to understand how the distributions for the climatological period were derived.
AC: We rewrote the description and hope that it is now more understandable and less confusing.

15. Several of the figures could be improved in a few ways. Specifically, I recommend the following changes:

○ Fig. 3: The top and bottom rows could arguably be combined. In this case, rather than colouring the symbols on the map by rainfall amount you could just make them blue for > 35 mm/h and red for > 60 mm/3h; then use these same colours for the bar plot (with red bars overlaid on blue bars).
AC: We combined Figs. 3a and 3c in one histogram (good suggestion), so that the number of stations is easier to compare. However, we kept Fig. 3b and 3d separate, since this way a spatial representation of the maximum precipitation (especially for the 3 h panel) is maintained.

○ Fig. 7: I don't think it's necessary to state the two thresholds within the plots; this information can be provided in the caption (with reference to the dashed lines).
AC: Personally, I'm a friend of putting a lot of information in figures (if they don't irritate too much), so that – if the figure is used elsewhere – all information is included. However, we removed it and provide the info only in the figure caption.

○ Fig. 12: I would get rid of the hatching showing the objectively identified cut-offs and use a darker contour for the pressure vertical velocity.
AC: We revised the figure and hope that by leaving the blue colours (< 1PVU) it is now clearer.

○ Fig. 14: It would be helpful to use different colours for the box-and-whisker plots corresponding to the study period and the 1981–2010 climatology. Use the same colour convention for Fig. 9.
AC: We implemented this suggestion.

○ Fig. 15: In the legend, rather than putting "(2018: N days incl. M skip days)" for each station I would just put "(N/M)" and then explain what these numbers indicate in the caption. So, for example, for Essen you would put "(17/3)" instead of "(2018: 17 days incl. 3 skip days)".
AC: We implemented this suggestion.

○ Fig. 16: Rather than plotting the percentage difference from the climatological frequency (which is confusing because it is a percentage of a percentage), I recommend expressing this difference in terms of the standard deviation of the climatological frequency. This will highlight whether the 2018 frequencies were exceptional in the context of typical year-to-year variability.

AC: We concur, that a percentage of a percentage would be confusing. However, in Fig. 16 we show in shading the ((absolute frequency in May/June 2018)-climatological frequency May/June 1981-2010)). Thus, over Northern Spain absolute frequencies exceed 50 %, which would be relatively speaking a change of more than 1000 % compared to climatology. We clarified this in the figure caption.
"Climatological mean percentage of days with a cut-off low in May and June (black contours; every 2 %; for May and June 1981 – 2010) and anomaly percentage of days during the study period (shaded in % with reference to mean percentage of days} in May and June; ERA-Interim)".
Furthermore, we checked the suggestion with the standard deviation of the cut-off low frequency and added a comment on that (however the added value of an extra figure for the std is too small to include; see Fig. here).

[Figure]

**Technical Corrections**

1. There are a few issues with tenses in the text. For example, the opening sentence of the introduction is written in the present perfect tense (use of "has been"), but should be in the past tense ("was"). The same goes for L261. The last sentence of the opening paragraph of section 3.3.3 is written in the simple present tense but should be in the past tense.
AC: The manuscript was checked by an editor service; we included the improvements. Thanks for the careful reading.

2. L23: Parentheses are (are not) for references and clarification (saving space) (Robock 2010). Please modify this sentence accordingly.
AC: We changed this as proposed.

3. L36: "…serve to precondition the thermodynamic environment."
   AC: We changed this.

16. L39–42: To which event are you referring here: 2018 or 2016? I suggest rewording this paragraph to make this clear. Similarly, you should state explicitly the event you are referring to on L71.
    AC: 2016. We clarified this in the text "During the episode in 2016…"

17. Line 60: Lifting will only lead to the release of CAPE (i.e. convective initiation) if it is sufficient for parcels of air to reach their level of free convection; however, it may still act to destabilise the column (increase CAPE) and erode lids (reduce CIN).
    AC: That's right. We added some words on this.
    "When such a positive PV filament moves over air masses that are conditionally or potentially unstably stratified, they trigger lifting and thereby release convective available potential energy (if the air parcel reach its level of free convection)} and facilitate/cause deep moist convection."

18. Line 90–91: Suggest revising the end of this sentence as follows: "…based on reports from storm chasers, eyewitnesses, voluntary observers, meteorological services, and news media."
    AC: We changed this as proposed.

19. L97–98: "… the Météo-France (1223/1935 stations with hourly/daily data)…"
    AC: We changed this as proposed.

20. Line 100: These are the national meteorological services of all the countries in your study so I don't think you need this last part of the sentence.
    AC: We deleted the last part.

21. Line 132: "wind speed and direction"
    AC: We changed this.

22. L146: What is the altitude of the lowest level?
    AC: 1 km above ground is the lowest level and 12 km the highest.

23. L199: Change "fewer" to "less".
    AC: We changed this.

24. L239–240: Suggest revising this sentence as follows: "Many of the record-breaking 1h and 3h rain totals occurred within this period (see Sect. 3.2)."
    AC: We implemented this suggestion.

25. L255–256: "…the latter on the day with the second most ESWD severe weather reports (cf. Sect. 3.1)." Rather than referring to the previous section here please specify the actual date.
    AC: We implemented this (31 May).

26. L257–258: You deal with the variations in large-scale forcing for convection later in the paper so I don't think it is necessary to include this here (unless you want to explicitly refer to the relevant sections).
    AC: We deleted the sentence.

27. L271: Change "were" to "where".
    AC: Thanks. We changed this.

28. L272–273: This sentence is confusing and should be revised.
    AC: We fundamentally revised and restructured the first part of Section 3 and deleted some passages. In doing so, the sentence was also deleted.

29. L357: Change "vast parts of" to "much of".
    AC: We changed this.

30. L362: Change "exemplarily shown for" to "exemplified by".
    AC: Good suggestion. We now used this formulation.

31. Line 380–381: Change "and was already mentioned at the end of Section 3.2" to just "(Section 3.2)".
    We changed this.

32. L510: Advected where?
    AC: "into Europe". We added this.

33. L526: Get rid of "(global/regional)".
    AC: We deleted this.

**#2 Review for WCD-2020-1 (RC1 from 23 Feb 2020)**

Overview: The paper presents a case study of a long-lasting thunderstorm series over France/central Europe in May/June 2018 that occurred south of a blocking high. The synoptic situation persisted for several weeks. The thunderstorms were associated with cut-off lows/potential vorticity filaments that formed on the south-west of the blocking high. As a result, numerous severe convective events such as flash floods, hail and wind gusts were recorded. The authors use multiple different data sets and methods to show how the large-scale dynamics contribute to the thunderstorm series and that this event was exceptional.

Overall, I like the author's idea of studying this event from synoptic down to the convective scales. Moreover, it is an interesting case. However, I think the manuscript was difficult to follow and can still be improved considerably. In the current form, it was unfortunately no pleasure to read through the study. My main criticism is that large parts of the paper (chapter 2,3,4, but also in the introduction) read like a collection of single parts which are not really connected with one another.

A central theme seems to be missing.
AC: See Introduction Lines 76-80:
"The primary objective of this paper is to examine the conditions and processes that made this particular thunderstorm episode in 2018 unique. We focus on the process interaction across scales, i.e., from the large-scale dynamics such as atmospheric blocking to meso-scale PV cut-off lows and/or small meso-scale PV filaments to modifications of the convective environment to local-scale thunderstorm occurrences." For this purpose, it is also necessary to describe the "synoptic framework during the thunderstorm episode, to show the severity of the events and to place event in a historical context".

I think, the authors should restructure the paper or parts of it and follow a clear path, e.g. from large-scale to the small scale or the other way around. If this is not possible, they should at least clarify the purpose of each (!) chapter at the beginning (as it is done in chapter 5) to facilitate the reading.

AC: We edited and restructured several text passages, in particular we rewrote the introduction, restructured the sub-sections in Section 2 and rewrote the beginning of Section 3. Furthermore, we tried to link the text passages better with each other. We also split long sentences.

Moreover, in my opinion, the writing can be improved, too. Some sentences are too long, which makes the text hard to read. Just write necessary information and just reference to papers that are relevant for your topic. Make the sentences clear and concise. Please connect the single chapters and (sub)sections with one another!
AC: We split long sentences. Additionally, we connected the individual chapters better and included a more detailed description at the end of the introduction, so this should help the reader for the storyline.

I will explain my criticism in more detail in the following:

(i) In the introduction, the authors switch strongly between different topics: first they introduce the case and its impacts in a few sentences. Then they describe convective development due to scale interactions (mainly lifting processes). Afterwards they describe the case again with focus on blocking which is described more general thereafter. In the successive part, the authors explain cut-off lows in the potential vorticity framework. Afterwards they switch back to the topic of blocking. However, these single parts are often unconnected with one another which is confusing for the reader!
AC: We fundamentally revised the introduction. On this occasion, there were some textual restructuring, shortening, and additions and hope that all these (justified) criticised points are thus been eliminated.

 (ii) In the data and methods chapter (chapter 2), data sources are often introduced without clarifying why the authors will need the data. At least some overview at the beginning of this section – how the study was designed and/or what data satisfies which purpose – would help the reader tremendously!

AC: We took this criticism into account and integrated the necessary information in the corresponding text passages (see also comments below).

Are there any new methods? Please clarify!

AC: Please, see comments below.

(iii) The same applies for chapters 3,4, and partly 5! Try to connect the single parts, try not to jump unnecessarily between topics. In the current version, it is really confusing for the reader.

AC: We edited several text passages to better structure aspects necessary for the individual chapters.

(iv) With respect to the methods my main concern is the usage of the 500hPa-wind instead of vertical wind shear. At least additionally analyzing shear had the advantage that your work can be compared more easily to the existing literature of convective events.

AC: We made a conscious decision to show V500 and not the shear (although we had already examined both). One reason for V500 was the connection with the propagation speed of the cells (see also the comment on propagation speed below), since our focus in the work was not to analyse the organizational structure of the cells using wind shear. Furthermore, the interesting point for us was the really extraordinary low wind speeds in the mid-troposphere.

In the panel of Figure 7 we now additionally included the bulk wind shear between 10 m and 500 hPa – showing that the values averaged over the study period are almost similar to V500. We added also some comments on that.

(v) Furthermore, I am missing evidence, that the thunderstorms have been single cells rather than multicells, MCS or slow-moving (HP) supercells.

AC: Separating the convective cells among their organization form might be an interesting issue, but is far beyond the content and aim of our paper. Furthermore, we do not have an algorithm that allows us to adequately identify this. We added animated images of radar reflectivity for two representative days in the video supplementary. These animations show how both single cells and multicells develop throughout the day. A brief discussion is included in Section 3. Furthermore, we weakened the statement about single cells and changed the wording into isolated cells (which clearly can be seen in the radar animations). Finally, we added some more information about the dimensions of the storms detected by the tracking algorithm to Table 1.

p. 16, lines 490-496: Especially since the precipitation amounts are so high, how do you know that the thunderstorms were mainly single cells? Moreover, did you mention at any point in your paper, how you differentiate between single cells and other convective thunderstorm types like multicells? Maybe you can put the radar movies for one of the extreme cases you talked about to the supplemental material?

AC: We followed the suggestion and included radar movies for two days (the most severe ones) in the supplementary. These animations show how both single cells and multicells develop throughout the day. A brief discussion is included in Section 3.1. Furthermore, we weakened the statement about single cells and changed the wording into isolated cells (which clearly can be seen in the radar animations). Finally, we added some more information about the dimensions of the storms detected by the tracking algorithm to Table 1.

**Specific minor comments:**

Abstract, p.1 line 2: "80mm" - what is the temporal range? a few hours?

AC: See Table 1. Theoretically several time scales are possible (1 h, but also in 3 h). We added an information in the abstract.

p.2, line 25-36: I am missing the general ingredients of convection here: instability, moisture, lift and shear. The ingredients-based concept is first mentioned in the Discussion chapter (chapter 6), I think it would be fitting in the introduction, too.

We moved the text passage on the ingredient-based theory of Sect. 6 to the introduction.

Moreover e.g.

Markowski and Richardson, 2010 (their chapter 10.4) &

Doswell III, C. A., Brooks, H. E., & Maddox, R. A. (1996): Flash flood forecasting: An ingredients-based methodology. Weather and Forecasting, 11(4), 560-581.
treat flash flood events. Especially in the Markowski and Richardson book, you can find a very similar synoptic pattern that led to flash flood events in the US (please refer to the publications mentioned therein).

AC: We decided to use the suggested references in the discussion section, because it fits best there.

p.2, line 34: "all these mechanisms" - which ones are meant here?

AC: In the course of a revision of the "introduction" this sentence was deleted.

p.2, lines 51-61: There are some publications concerning the PV framework and convection: e.g.

Russell A, Vaughan G, Norton EG. 2012. Large-scale potential vorticity anomalies and deep convection. Q. J. R. Meteorol. Soc. 138: 1627–1639. DOI:10.1002/qj.1875;

Morcrette CJ, Lean H, Browning KA, Nicol J, Roberts N, Clark PA, Russell A, Blyth AM. 2007. Combination of mesoscale and synoptic mechanisms for triggering of an isolated thunderstorm: a case study of CSIP IOP 1. Mon. Weather Rev. 135: 3728–3749.

Can you please put your work in context with the existing literature?

AC: Thank you for your literature suggestions; we put these works in our introduction:

"The effect of large-scale PV anomalies accompanied by cut-off lows on deep moist convection (in relation to severe precipitation events) has already been observed in other studies showing for Europe that this is an important mechanism for convection due to the associated patterns of advection and vertical motion (Roberts, 2000; Morcrette et al., 2007; Browning et al., 2007; Russell et al., 2012). But the effect is complex and not well understood."

p.3, line 62: "A connection between atmospheric blocking and heavy precipitation events..." - Why again blocking? The sentence is almost identical to that on page 2, lines 49-50. Why don't you merge these parts?

AC: We fundamentally revised the introduction and considered (or rewrote) this point.

"At first, the relationship between atmospheric blocking and severe convection seems counterintuitive because heat-waves and associated droughts are frequently associated with such patterns (e.g., Pfahl and Wernli, 2012a; Bieli et al.,752015; Schaller et al., 2018; Röthlisberger and Martius, 2019). But in peripheral locations upstream and downstream blocks can also create environmental conditions conducive for deep moist convection development. For example, the link to heavy precipitation events (including flood events) has already been established in the last decade (e.g., Martius et al., 2013; Grams et al., 2014; Piaget et al., 2015; Sousa et al., 2017; Lenggenhager et al., 2018; Lenggenhager and Martius, 2019)."

p.3, line 68-70: "[..] such situations are usually associated with weak wind speeds at mid-tropospheric levels (cf. PIP16), so that thunderstorms become almost stationary and usually do not develop into organized structures such as large mesoscale convective systems or supercells." – First: where is the wind weak? in the high, the low, at the western flanks?

AC: Over the area, where we observed / investigated the thunderstorms.

Second: What about HP-supercells (high precipitation supercells)? Can you please comment on HP-supercells.

AC: HP-supercells might have occurred. As we haven't investigated that, we rewrote the sentence "In addition, such situations are usually associated with weak wind speed at mid-tropospheric levels and thus weak vertical wind shear over the thunderstorm area with the consequence that thunderstorms become often stationary and rarely develop into large organized convective systems."

p.3, line 83: What do you mean with "secondary effects"? Please elaborate.

AC: We mean their sub-perils (e.g., hail, heavy rain, wind gusts); we added an explanation:

"thunderstorms and secondary effects such as heavy rain, hail and convective wind gusts"

p.3, line 85: "(May/June)" - These are the whole months (1.5-30.6)? It is confusing since you already stated two different periods in the text before.

AC: Yes, the whole month; we added this information (1 May to 30 June).

p.4, line 87-93: Please clarify what the purpose of the ESWD data is. Do you use different quality levels or all? Why don't you show the reports also in e.g. Belgium or Italy?

AC: The purpose of the ESWD data is to show the sub-perils associated with the thunderstorms, and that these were preferably heavy rain events. We use all data above QC0+ (we added this).
ESWD Data from Belgium have already been used (see Fig. 2) – since it is in our study area. Maybe you mean the precipitation data from Belgium (here we were not able to get any). Data from Italy are not included because Italy is not included in our study area (see lines 92-93 "The study area includes parts of central and western Europe - France, Benelux (Belgium, Netherlands, Luxembourg), Germany, Switzerland and Austria (see Fig. 1)" – for which data were available.

p.4, line 88: It is good to know that the ESWD collects data about heavy rain, hail and wind gusts. However, what data did you use for the analysis?

AC: All these three sub-perils. We clarified this.

p.4, line 90: better: "[..] mainly based on reports of storm chasers, [..]"

AC: We corrected this.

p.4, line 108: Is there a description of the REGNIE data in English for non-Germans, too?

AC: Not really, but about HYRAS, which uses the same methodology and which is already cited (Rauthe et al., 2013).

p.4, line 104-111: Why did you decide to use the REGNIE data. The data seems to interpolate measured precipitation on a regular grid. Is the REGNIE data suitable to analyse extreme convective precipitation which might be short in duration and small in scale? Or might these extremes be smoothed during the interpolation process? Did you consider to use a highly-resolved reanalysis data set for comparison reasons?

AC: We used REGNIE data only for the estimation of return periods because of their long-term availability of approx. 70 years, and the large number of approx. 2,000 climate stations used in the regionalization method. RADOLAN data (merger between Radar and station data) would be a better choice, but are available only for 20 years and, thus, not suitable to estimate return periods. Reanalysis also tend to underestimate precipitation totals. We included an explanation on this.

p.4, line 109-111: "Note that the REGNIE time series are affected by temporal changes in the number of rain gauges considered by the regionalization. For our purpose, the homogeneity of the data are sufficient." - Can you please give a reference here? Did the number of stations change in the analysed period?

AC: We included the reference Rauthe et al. (2013). Unfortunately, there is no recent reference available; also the exact number of stations for the regionalization is not known (it was approx. 2,000 stations in 2011).

p.4, line 115: "[..] appropriate for precipitation statistics [..]" - can you please give a reference here and explain a bit more in detail what was done in the previous literature with the Gumbel distribution.

AC: That's right, "precipitation statistics" is a little too generic. We changed this sentence into: "The Fisher-Tippett Type I distribution, also known as the Gumbel distribution (Gumbel, 1958; Wilks, 2006), has been extensively used in various fields including hydrology for modelling extreme events, i.e. to estimate statistical return periods or return values (Sivapalan and Blöschl, 1998; Rasmussen and Gautam, 2003). The Gumbel cumulative distribution function (CDF) is given by:"

p.4, line 112-123: General comment: Is this method new? If so, please state here, otherwise, please write something like: "we follow the methodology used in..."

AC: No, is not new. We clarified this.

p.4, line 116: R is not explained.

AC: That's correct. We added this (R is the investigated variable, here precipitation values.)

p.5, line 117: Can you please give a reference for the "Method of Moments". If it is also explained in the Wilks-book, maybe you can add the chapter to the reference here.
AC: Yes; it is explained in Wilks (2006) in Chapter 4 (Parametric Probability Distributions).

p.5, chapter 2.1.3: What will you use the data for?
AC: (Note now chapter 2.1.1) "Lightning data offer the best spatially homogeneous coverage for a full thunderstorm detection, but does not distinguish according to the severity. For this purpose, we use eyewitness reports of the European Severe Weather Database and precipitation observation (station-based and gridded-based)". Furthermore, the lightning data are used to compare these with the PV cut-offs/filaments. We added this information.

p.5, line 132: what parameters will be taken into account to estimate the "atmospheric conditions"?
AC: As already mentioned the SLI; the second is the V500. We added a comment on this to clarify.

p.5/6, chapter 2.1.5: Can you conclude from the radar data, if the thunderstorms rotated? For example by comparing the direction of the mean tracks to the investigated severe thunderstorms?
AC: For proper detection of rotation in radar data you need the Dual-Doppler wind fields, which we do not have. And even with that the detection or rotation is very difficult (we didn't even detect rotation in the radar data of the severe supercell on 28 July 2013; cf. Kunz et al., 2018). Because the track direction depends – in addition to vertical pressure disturbances - on various effects such as the vertical extent of the cells relative to the wind shear or the formation of new cells (particularly in case of multicells), a comparison between the two as suggested would not allow to give any conclusions about the rotation of the cells.
Kunz, M., Blahak, U., Handwerker, J., Schmidberger, M., Punge, H. J., Mohr, S., Fluck, E., and Bedka, K. M.: The severe hailstorm in SW Germany on 28 July 2013: Characteristics, impacts, and meteorological conditions, Q. J. R. Meteorol. Soc., 144, 231–250, https://doi.org/10.1002/qj.3197, 2018.

p.6., chapter 2.2: What fields will you use?
AC: See Lines 193-195: "Beside the atmospheric stability (based on SLI), we examine in the study V500, the bulk wind shear (BWS; e.g., Thompson et al., 2007), 500 hPa geopotential height (Z500) and the vertically integrated water vapor (IWV)."

p.6, line 172/173: "[..] but reflects important seasonal differences." - What do you mean here?
AC: Classical weather regime definitions typically distinguish summer and winter regimes. Our year-round definition contains these seasonal different patterns and therefore has more regimes than classical definitions. Still these patterns can occur in any season and are important for local weather conditions.
A figure showing the weather regimes would be nice, at least later in the text, where you analyse the data, you could show the typical patterns of the prevailing regimes.
AC: The illustration of the Atlantic-European weather regimes can be found in Grams et al., 2017 – however in the Supplementary information (Supplementary Figure 1); but only for the winter season, but which are very similar to the summer season. We now provided the typical patterns (for May/June) as supplementary material (see Supplementary Figure 1).

p.7, chapter 2.4: Is this method new or does it already exist? Please clarify.
AC: Similar approaches have been used to relate surface weather to weather objects in earlier work (e.g., Pfahl and Wernli 2012, 2014; Pfahl et al. 2014). We now clarified this in the text.
Still as far as we know this is the first study matching lightning data to cut-off cyclones.

p.7, line 197: general comment: The Brunt-Vaisala frequency is smaller in summer, too, due to decreased stability.
AC: Here we aim to provide a physical justification for the scale of our buffer radius not an exact estimation which would depend on each specific case. We think our scale analysis yields a reasonable estimate of a remote influence and that seasonal variation in stability would not change the order of magnitude. In addition, we tested the sensitivity to a range of buffer radii with now impact on our qualitative interpretation.

p.7, line 199-203: You could add a table to the supplementary material showing the change in associated lightning.

AC: We included a table in the supplementary materials (and also the results/Fig. for a buffer radius of 400 km and 600 km).

As stated in the main text, the buffer radius of 500 km is chosen subjectively. We know that there is an uncertainty involved but that a radius below 400 km or above 600 km is not appropriate from a meteorological perspective. Changing the buffer Radius to 400 or 600 km yields the following conclusions:

1.) The percentage of lightning strikes that can be linked to a cut-off low does not change in the first five days of the study period. This is due to the fact that only a few cut-off lows were identified at that point.

2.) The percentage of lightning strikes that are matched with cut-off lows changes in the most active period from 27 May to 1 June. For a radius of 400 km 60 % of the lightning data can be matched, for 500 km 75 % and for 600 km 88 %. This shows that there is a relatively high sensitivity for this main area to the chosen radius. Following our arguments in the paper, we believe that 500 km is a conservative radius and 600 km are probably even more appropriate. This can also be seen in the overview plots for the period from 31 May to 1 June with the buffer radius (see supplementary). Moreover, even with a low radius of 400 km more than half of all the lightning strikes in this most active period can be linked to a cut-off low.

3.) The overall matched lightning percentage increases from 42.2 % to 64.3 %, which shows that there is a sensitivity to the chosen radius but that these results do not change our conclusions.

Moreover, it would be nice to see this "buffer zone" in the figures.

AC: We decided not to include a figure including the buffer zone in the main text (otherwise it would be too obvious), but we included it in the supplement for the interested reader.

p.8, line 211-214: Why do you use the wind speed at 500hPa instead of the deep-layer shear? Additionally, deep-layer shear is a widely used variable and the results would be better comparable to the existing literature. I do not understand the motivation here, especially since the authors later in the paper discuss the importance of shear on the organization of thunderstorms.

AC: We made a conscious decision to show V500 and not the shear (although we had already examined both). One reason for V500 was the connection with the propagation speed of the cells (see also the comment on propagation speed below), since our focus in the work was not to analyse the organizational structure of the cells using wind shear. Furthermore, the interesting point for us was the really extraordinary low wind speeds in the mid-troposphere.

In the panel of Figure 7 we now additionally included the bulk wind shear between 10 m and 500 hPa – showing that the values averaged over the study period are almost similar to V500. We added also a comment on that (Lines 403-408).

p.8, line 216: "Overview" - Can you please be more precise, there is another chapter which is also called overview. What is you intention of this whole chapter?

AC: We restructured the headings in Chapter 3.

p.8, line 222/223: "The three-week period from 22 May until 12 June was the most active thunderstorm episode with a total of 868 heavy rain, 144 hail, and 145 convective wind gust reports based on the ESWD." - do you mean "the most active thunderstorm episode" in the year 2018 or inother period?

AC: During our extended study period (1 May to 20 June) or also in May / June 2018. We clarified this point.

p.8, line 223/224: "An average area of 715,000 km2 was affected by lightning per day" - is that much, what is the average value for Europe?

AC: Yes, this is twice the area of Germany per day, that's much! We included the relation to the area of Germany.

p.8, line 227/228: "As shown in Figure 2b, most of the severe weather reports came from the western part of France, Benelux, central and southern Germany, and the easternmost part of Austria." - Can you explain the gap in central/eastern France? From your Fig. 8 Lifted index was negative, too. Moreover, the mean wind was not much different from western France?

AC: This primarily depend on the availability of storm chasers, eyewitnesses, or voluntary observers. We added a comment on this: "While the spatial composition of the ESWD reports shows regional gaps due to an under-representation of eyewitness reports, for example, in Central and south-eastern France (cf. Groenemeijer et al., 2017; Kunz et al., 2020), thunderstorm days are observed throughout the study area". We included a thunderstorm day map of the study period in the Supplement (SFig.2).

p. 8, line 241: Isn't the number of ESWD reports depending on the number of people reporting events? Is there a difference if you just use some of the quality levels?
AC: Right. And the quality levels can't fix this. The reports in the ESWD are absolutely controlled by the activity of the different people. Here, the population density plays a significant role or where the most active "reporters" live and what their area of investigation is. There are very severe events, which have smaller number of reports that less severe events.

p.9, line 252: "low wind speed [..] slow propagation" - You could mention here, that you will give more details later in the text. While first reading through the text, I wondered if these statements will be verified later or just stated as a fact here?
AC: Note that the Section 3 (Part 1) is completely revised. We followed the suggestion and added a reference and further comment on that (now Lines 287-297; text passage with tracking/propagation speed was moved).

p.9, line 257: What is meant with "The strength and spatial extent of the lifting forcing varied from day to day, [..]"? Can we see this in one of the figures?
AC: We deleted the sentence.

p.9, line 260-273: Just write about the events that are explained in more detail. All other numbers will just lead to confusion and can be seen in the table.
AC: Ok, we followed the suggestion. The Section 3 (Part 1) is completely revised (and was shortened).

p.10, chapter 3.3: It would be reader-friendly if you explained what the intention of this chapter is. Please give an introductory sentence.
AC: We included a more detailed description at the end of the introduction, so this should help the reader for the storyline. Additionally, we included a sentence at the beginning of Section (now) 3.1 for the storyline.

p.10, lines 292-303: It would be a helpful addition if you overlayed the ESWD data. This would make it easier to follow your arguments.
AC: We implemented the suggestion. Note, however, that the first four days of the panel are even days on which no ESWD reports were reported (cf. Fig. 2).

p.11, lines 312-313: Can you please plot the typical patterns of the Zonal regime and the European Blocking.
AC: The illustration of the Atlantic-European weather regimes can be found in Grams et al., 2017 – however in the Supplementary information (Supplementary Figure 1); but only for the winter season.
We now provided the typical patterns (for the May/June season) as supplementary material.

p.11, lines 315-323/line 330: Can you plot in Fig 6/7a+b additionally to the regimes/sounding data, the lightning activity (out of Fig. 2a) for easier comparisons.
AC: We did not integrate the lightning data into the two figures, which made it too confusing. Instead, we combined the two figures and noted that the vertical black bars represent the study period, which was the period with the highest lightning activity (see Fig. 2).

p.12, line 348/349: "Because of the low wind speed in the mid-troposphere, most of the thunderstorms moved very slowly or even became stationary." - The motion of thunderstorms is not necessarily determined by the wind at 500 hPa - can you please give a reference that shows that the storm motion correlates with 500hPa winds.
AC: It's right, the propagation of convective cells is driven by various factors such as gust-front lifting in case of multicells and vertical pressure gradients extending over a deep layer in case of supercells. We took the 500 hPa wind as a proxy because proper determination of the vertical extent – even though possible – would be out of the context of this paper. We have overworked and extended the whole paragraph, included to references and moved it into Section 3 (Description of the thunderstorm episode 2018).

p. 12, lines 358-360: "The fact that relatively high PV cut-off frequencies expand over a larger region of western Europe underlines that multiple individual PV cut-offs form on the upstream flank of the blocking ridge, and intermittently move across Iberia, France, the British Isles, the North Sea, and Germany [..]" - How do you distinguish between a stationary cut-off low and newly-formed moving ones in Fig. 10? Please clarify.
AC: We concur that this statement was misunderstandable. We here only refer in the first half of the sentence to Fig. 10 (now reference included after "… western Europe (Fig. 10) underlines that …". The occurrence of multiple cut-offs was explained in the synoptic overview Fig. 4. The references "(see Fig. 4)" is now earlier in the sentence "…of the blocking ridge (see Fig. 4), and intermittently …"

p. 13, line 396: better: " To estimate the severity of the rainfall with respect to the rainfall climatology, [..]"
AC: We changed this.

p. 13/14, chapter 5.1: I wonder if the return periods are dependent on the REGNIE data and how it is designed. Is it possible to get higher precipitation amounts than observed at the stations? Can you please comment on this?
AC: Yes of course, all extreme value estimates depend upon the used data set. REGNIE certainly underestimate the precipitation peaks, but this is the case for both the observation period and the 67-years reference period. We added a comment in this section and a further comment in the data description.

p. 14, chapter 5.2: If I understand it correctly, the only thing one can directly compare in Fig. 14 - left vs. right boxes-and-whiskers - is the median on the left with the complete box-and-whiskers on the right? Maybe you could add the median of the actual period as an extra symbol to the right box-and whiskers.
AC: We decided against this proposal, because we do not want to overload the plot unnecessarily with symbols, which is already very voluminous. In addition, the two box-and-whiskers are very close together, so they are easy to compare. However, we added a comment in the figure caption, which should help.

p. 14/15, lines 435-448: Although, your main intention is presumably, that the investigated storm period is a rare event. From your text, I could not understand how Fig. 15 was produced. Can you please rewrite the text passage and clarify.
AC: We rewrote the description and hope that it is now more understandable and less confusing.
What is meant by skip days and why do you use 3 instead of 1 as in the referenced paper? Please explain.
AC: "Within a cluster of seven event days, we allow one day to be a non-event one (skip day), which is not considered in the total length n.  For example, clusters with a length of up to 7 (14/21) days may contain at most 1 skip day (2/3 skip days)."  Means one skip day per (started) week.

p. 15, lines: 463-466: "A further relevant condition for the evolution of deep moist convection is the vertical wind shear or, more generally, the wind at mid-tropospheric levels, which is decisive not only for the organizational form, the longevity and thus the severity of the convective storms (e.g., Weisman and Klemp, 1982; Thompson et al., 2007; Dennis and Kumjian, 2017), but also for their propagation (Corfidi, 2003)." - As far as I know, all the cited papers talk about the vertical wind shear, but not about the wind at mid-tropospheric levels (although they might mention storm-relative winds, but this can be quite different from the mid-tropospheric wind). Of course, I can be mistaken, hence, please cite the text passages of the papers, where the mid-tropospheric wind is mentioned in your authors's response.
AC: In the panel of Figure 7 we additionally included bulk wind shear between 10 m and 500 hPa. The values averaged over the study period are almost similar to V500. We added a further paragraph that briefly discusses this issue. Furthermore, we deleted wind speed in the quoted sentence.

p. 16, lines 475/476: "[..] air masses were trapped [..]" - Is it possible to show, that the air masses were trapped over several weeks (e.g. by using trajectories)?

AC: Good suggestion; we implemented a Lagrange-based analysis of the air masses, which supports this statement (new Section 3.2 and Figure 9).

p. 16, lines 484-485: "In our investigated case, thunderstorms were often triggered by large-scale lifting associated with upper-level cut-off lows or filaments of high PV that separate from the main PV cut-off" – I am convinced that the cut-off lows provided good environmental conditions for convection, however I doubt that the cut-off lows triggered the thunderstorms directly. What about (older) outflow boundaries? Can you please comment on that?

AC: That's correct. Thunderstorms are triggered by a variety of mechanisms. Large-scale uplift by itself won't bring air-packets up to LFC height, speeds are too low (resp. time scales would be too long for that). Rather, it is the decrease in CIN and the increase in CAPE that are relevant here.
We reformulated this sentence and changed the statements of thunderstorm triggering by large-scale lifting throughout the manuscript.

p. 16, lines 490-496: Especially since the precipitation amounts are so high, how do you know that the thunderstorms were mainly single cells? Moreover, did you mention at any point in your paper, how you differentiate between single cells and other convective thunderstorm types like multicells? Maybe you can put the radar movies for one of the extreme cases you talked about to the supplemental material?

AC: We followed the suggestion and included radar movies for two days (the most severe ones) in the supplementary. These animations show how both single cells and multicells develop throughout the day. A brief discussion is included in Section 3. Furthermore, we weakened the statement about single cells and changed the wording into isolated cells (which clearly can be seen in the radar animations). Finally, we added some more information about the dimensions of the storms detected by the tracking algorithm to Table 1.

**Figures:**

Fig. 2b: Please do not use the rainbow color scale. It is hard to differentiate between some days. Maybe if you switch to a scale, it might be possible to see some temporal clustering?

AC: We tested different (sequential) colorbars. However, we had to find out that the one used so far is the most suitable one. With a sequential colorbars the available color spectrum is too small to divide the days decently. Furthermore, there was no clear clustering on a time scale of several days, so that the added value of the figure is to be able to identify individual days more precisely (see figure below)..

[Figure]

Are there really no events in northern Italy, the Czech republic or Poland?

AC: Data from Northern Italy, the Czech republic or Poland are not included because these are not part of our study area (homogeneity reasons).

Fig. 3b: I cannot see any difference between the blue colors here.
AC: We modified the colorbar.

Fig. 4: Is it possible to add the locations of the ESWD reports of the associated day to maps?
AC: We implemented the suggestion. Note, however, that the first four days of the panel are even days on which no ESWD reports were reported (cf. Fig. 2).

Fig. 6: It is impossible to differentiate between ZO/SCTr, EuBL/SCBL and AT/GL.
AC: We modified the colorbar.
Can you add the affected lightning area (from Fig 2a) to the curves.
Fig. 7: Is it possible to add the lightning data from Fig 2a?
AC: We did not integrate the lightning data into the two figures, which made it too confusing. Instead, we combined the two figures and noted that the vertical black bars represent the study period, which was the period with the highest lightning activity (see Fig. 2).

Fig. 12: There is no red hatching (in my print it looks black?).
AC: Oh, sorry in an older version of the Figure the PV on the 325 K isentropic surface was red. Thanks, we changed this.
Is it possible to add the buffer zone?
We decided not to include a figure including the buffer zone in the main text (otherwise it would be too obvious), but we included it in the supplement for the interested reader.

Fig. 14: Can you please add the median from the left box-and-whiskers as an extra symbol to the right ones?
AC: We decided against this proposal, because we do not want to overload the plot unnecessarily with symbols, which is already very voluminous. In addition, the two box-and-whiskers are very close together, so they are easy to compare. However, we added a comment in the figure caption, which should help.
Please also plot the deep-layer shear.
AC: We decided to include the aspect with the shear in Fig. 7. Since the results are always quite similar (V500 vs. shear; see comment above) or the core statements are not significantly influenced by the result of the shear, we decided against showing the shear for the rest of the figure with V500, since we did not see any added value.

[revised manuscript text omitted]

---

## Referee Report (RR1)

**Review for WCD-2020-1, Revision 1**

**General Comments**

The changes and additions made by the authors have further improved what was already a very good piece of work. However, reading the revised manuscript I felt that the text could be improved in a number of ways. I have tried to cover these in my suggestions below, but I would encourage all coauthors to give the paper a thorough read through before the final submission to make sure that things are clearly explained, particularly when it comes to the methodology. I also have a concern regarding the use of the word "triggering" to describe the role of cut-off lows and PV filaments in the event, which I believe needs to be addressed. As such I am recommending further minor revisions.

**Specific Comments**

My only significant comment relates to your repeated use of the word "triggering" to describe the role of the cut-off lows and PV filaments in this event. In general, convective triggering refers to the process whereby air parcels are lifted to their level of free convection and subsequently rise through buoyant accelerations. For an individual convective cell (thunderstorm) this process occurs on the scale of a few kilometres to ~100km (i.e. the meso-$\beta$ or meso-$\gamma$ scale following Orlanski 1975; see also Markowski and Richardson 2010, section 1.1). On the other hand, lifting associated with cut-off lows and other synoptic-scale disturbances occurs on length scales of 100s of kilometres to ~1000km (meso-$\alpha$ scale). It is generally accepted that this lifting contributes indirectly to convective initiation (triggering) through the generation of CAPE and the removal of CIN, via changes in lapse rate (see Markowski and Richardson 2010, section 7.1) - in other words large-scale ascent *primes* the atmosphere for convective initiation. However, the initiation process itself is typically associated with phenomena such as convergence lines, thermally driven circulations (sea/lake/vegetation breezes), orographic lifting, and boundary-layer thermals, at least for surface-based convection (some elevated MCSs may be directly triggered by large-scale ascent). I think it is important that this distinction is clearly articulated in your paper. As such you need to modify the text in several places, including L56-59, L425, L433-434, L460-462, and L584.

L27-34: You should restructure this part of the paragraph so that the three ingredients for deep moist convection are listed together. State the ingredients first and then discuss the scale of the processes they are associated with (synoptic for instability and moisture; mesoscale to storm scale for the lifting mechanism).

L31: Latent and conditional instability are one and the same (see, for example, http://glossary.ametsoc.org/wiki/Latent_instability). Also, potential instability is generally not considered to be a major factor in the preconditioning of convective environments (see section 3.1.3 of Markowski and Richardson 2010). There are also various other forms of

instability (centrifugal, inertial, symmetric, shear). As such I would just state conditional instability as the first of the three ingredients for deep, moist convection.

L62: It should be "fully" not "full" here. However, I would actually recommend deleting this sentence as it is a bit "hand wavy".

L78: I would say "mesoscale cut-off lows and PV filaments".

L82: Get rid of "and their accompanying phenomena"

L84: "prior to"

L87: Get rid of "The next" and add "then" after "Section 4" (i.e. "Section 4 then puts the results in a historical context…")

L94: Rather than "secondary effects" I would say "associated hazards".

L103: You need to say "allow us to investigate" or, alternatively, "permit/facilitate an investigation of".

L126: I think it should just be "Météo-France", not "the Météo-France".

L128-129: Here and elsewhere I would use the term "1-hour extreme rainfall events" rather than "hourly extreme rainfall events". Generally I would take "hourly" to mean "occurring every hour" rather than "lasting for 1 hour". This is also consistent with "3-hour extreme rainfall events".

L135: Since "the RR collective" isn't mentioned again, you can get rid of the statement in parentheses.

L147: "location and scale parameters, respectively"

L150: I think you mean "standard deviation" not "derivation"

L152: You can either just say "return period" here or use the symbol $t_{RP}$; you don't need both as this definition is already given in the previous sentence.

L158: Suggest changing to "12 equidistant vertical levels extending from 1 km to 12 km above ground level (AGL)".

L169-170: I'm not sure it is fair to assume weaker cells "cannot move at higher speeds" than those above 55 dBZ. I would instead simply note the caveat that your use of a high reflectivity threshold means that the resulting storm-motion estimates may not be representative of weaker convective cells.

L189-190: Suggest modifying the end of this sentence as follows: "...to describe the large-scale meteorological conditions and define weather regimes (see Sect. 2.3), perform kinematic backward trajectories (see Sect. 2.4), and identify cut-off lows (see Sect. 2.5)."

L193: Here and throughout your analysis you should say "bulk wind difference (BWD)" rather than "bulk wind shear". Shear has units of $s^{-1}$ as it is the BWD divided by the layer depth.

L198: Rather than using Z500' for 500 hPa geopotential height anomalies, I suggest using Z500 to represent 500 hPa geopotential height and explicitly stating when you are talking about an anomaly. For example, on L206 you would say "dominated by a negative Z500 anomaly".

L199: Why did you choose the first seven EOFs? What percentage of the total variance do they collectively explain?

L217-218: Presumably, the five "surrounding" grid points are the nearest grid point to the sounding site and its immediate neighbours to the north, south, east and west; however, this should be stated explicitly.

L220: Get rid of "where the air masses relevant for the thunderstorm development are located". Air below 950 hPa and above 600 hPa is certainly relevant for thunderstorms!

L224: Rather than "the literature" I would say "previous studies" (or work or research).

L245-257: This description of the persistence analysis is quite difficult to follow, particularly the first paragraph. As such I would recommend completely rewriting it. Also, as stated in my original review, you should avoid using the term "cluster" here (and in Fig. 15) to avoid confusion with the actual cluster analysis used to define weather regimes.

L257: What do you mean by "the maximum of the daily minima"? Please rephrase.

L268: Get rid of "an area" before "twice the size of Germany"

L275: I think you mean "evolution" rather than "evaluation" here.

L285: What does "(radar visibility)" indicate? Are you saying that the cells were only visible on radar for 30 minutes? Please explain or delete this if it isn't important.

L290: As stated in my original review, you should avoid using parentheses to save space as the resulting sentences are much more difficult to read and comprehend.

L312: This sentence has some grammatical errors. I suggest revising as follows: "However, this rain fell in a period of 3 hours, with 60 mm falling in just 50 min."

L320: Put "indicating wind speeds between 25 and 31 m $s^{-1}$" in parentheses.

L324-327: This sentence would also benefit from rewording. Something like "In a few cases, deep-layer shear magnitudes were sufficient (BWD up to 20 m s$^{-1}$) for the development of severe storms, with large hail up to 5 cm in diameter recorded in Southwest France on 26 and 9 June and in southern Germany on 11 June."

L359: "(Fig. 6a)"

L367: Why introduce the abbreviations "ZO" and "EuBL" here if you aren't going to use them in the text?

L379: I would say "a pronounced decrease in convective activity"

L385: Get rid of the first instance of "values"

L396: Since your analysis considers geopotential height on constant pressure surfaces you should say "weak geopotential height gradients".

L403: One way to highlight the strong relationship between V500 and BWD would be to compute the correlation coefficient between the two. You could do this both for the sounding data and the ECMWF analysis over the domain shown in Fig. 8. Just a thought.

L405: Get rid of "squall lines" (MCS covers this).

L407: Not sure what you mean by deep-layer shear. In my experience this is another name for the 0-6 km (or surface-500 hPa) BWD. As noted above, shear has units of s$^{-1}$, not m s$^{-1}$.

L409-422: I recommend using the term "air parcels" rather than "air masses" in this section, since the former is more consistent with what a back trajectory represents.

L469-470: Suggest rewording this sentence as follows: "This analysis is restricted to Germany due to the availability of long-term (> 50 years), high-resolution (1 km$^2$) gridded rainfall data."

L479: What do you mean by "partly with new all-year records"? Maybe rephrase this.

L480-481: Suggest rewording this sentence and connecting it with the next one (getting rid of the paragraph break in the process) as follows: "This does not appear to be an artefact of insufficient gauge density, as most events are represented by multiple gauges (not shown). Instead, it likely reflects the very slow propagation of storms..."

L493-494: Suggest getting rid of (or moving) the sentence beginning "Recall that…" as it breaks up the flow between the preceding and following sentences. Also, in the next sentence, suggest changing "In doing so" to "Thus".

L504: Here and elsewhere in this section, change "event persistence(s)" to "CE duration".

L508-509: Change "event persistences of CE with long duration" to "long-duration CEs". You might consider completely rewording this sentence as follows: "To put these numbers in context, Fig. 15 shows the relative frequency of CEs in May/June as a function of their duration for the period 1981 to 2010."

L510-515: This additional explanation of the procedure is confusing and unnecessary. I recommend getting rid of it and the subsequent paragraph break.

L541: Change "convective" to "convection"

L581: Change "due to several reason" to "in several respects"

L611: I'm not sure what you mean by "(e.g. jet stream)". Please either elaborate or delete this.

Table 1: I don't think the track length and area are particularly informative, so these can probably be removed. The information on rainfall intensity and storm speed are more useful, but you don't actually discuss them anywhere in the text.

Figure 2: Rather than saying "the extended study period" I would give the dates explicitly (i.e. 1 May to 20 June).

Figure 3: What do you mean by "total maximum"? Also, as stated in my original review you should say "accumulation" rather than "sum" when referring to rainfall amounts in mm.

Figure 4: I don't think it is necessary or appropriate to apply a spline filter to the distributions here. Just plot the raw data as histograms (c.f. Fig. 11).

Figure 6: For some reason your color bars have more ticks than colours. Also the number of ticks per color varies and the tick labels don't always align with changes in the color level. Please make it so that the ticks and labels occur at the boundaries between the colors; otherwise it is difficult for the reader to extract quantitative information from the figure. The caption for this figure also needs revising for clarity. Here is my suggestions: "Mean anomalies during May/June 2018 of (a) 500 geopotential height anomaly (shaded in gpm) and (b) integrated water vapour anomaly (shaded in kg m$^{-2}$), together with the mean 500 hPa geopotential height (contours every 40 gpm). Data are from ERA-Interim and anomalies are computed with respect to the 1981-2010 climatology."

Figure 7: Panel (a) needs to be better explained in the caption. In particular you should state the meaning of the bold sections of the curves and the colours along the x axis. Also, in panel (c) the y axis should be labelled as "V500 (m s$^{-1}$)".

Figure 8: Add "at 12 UTC" before "averaged over the study period" and get rid of "12 UTC" from the parentheses at the end.

Figure 9: Please explain either in the caption or the main text how the ellipses were defined. Also, I suggest using "distance along trajectory" instead of "total distance".

Figure 13: Change "Return periods" to "Return period" (singular).

Figure 14: I think you mean "2nd and 3rd quartiles" (not "1st and 3rd"). Also, I would recommend making the whiskers and outliers the same colour as the boxes and using solid rather than dashed lines for the whiskers.

Figure 15: As noted above (and in my original review) you should avoid using the term "cluster" in this analysis. Instead I recommend using "CE duration" (this should also replace "event persistence"). Also, it should be "May", not "Mai".

Figure 16: I believe this information could be better presented in Fig. 10, by replacing the Z500 contours (which are already shown in Fig. 6) with the climatological cut-off low frequency. This way the reader can directly compare the climatological and 2018 cut-off low frequencies without the need to consider anomalies of percentages. You can then get rid of Fig. 16.

---

## Author Response (AR2)

**#1 Review for WCD-2020-1, Revision 1 (RC1 from 01 June 2020)**

**General Comments**

The changes and additions made by the authors have further improved what was already a very good piece of work. However, reading the revised manuscript I felt that the text could be improved in a number of ways. I have tried to cover these in my suggestions below, but I would encourage all co-authors to give the paper a thorough read through before the final submission to make sure that things are clearly explained, particularly when it comes to the methodology. I also have a concern regarding the use of the word "triggering" to describe the role of cut-off lows and PV filaments in the event, which I believe needs to be addressed. As such I am recommending further minor revisions.

**Specific Comments**

My only significant comment relates to your repeated use of the word "triggering" to describe the role of the cut-off lows and PV filaments in this event. In general, convective triggering refers to the process whereby air parcels are lifted to their level of free convection and subsequently rise through buoyant accelerations. For an individual convective cell (thunderstorm) this process occurs on the scale of a few kilometres to ~100 km (i.e. the meso-$\beta$ or meso-$\gamma$ scale following Orlanski 1975; see also Markowski and Richardson 2010, section 1.1). On the other hand, lifting associated with cut-off lows and other synoptic-scale disturbances occurs on length scales of 100 s of kilometres to ~1000 km (meso-$\alpha$ scale). It is generally accepted that this lifting contributes indirectly to convective initiation (triggering) through the generation of CAPE and the removal of CIN, via changes in lapse rate (see Markowski and Richardson 2010, section 7.1) - in other words large-scale ascent primes the atmosphere for convective initiation. However, the initiation process itself is typically associated with phenomena such as convergence lines, thermally driven circulations (sea/lake/vegetation breezes), orographic lifting, and boundary-layer thermals, at least for surface-based convection (some elevated MCSs may be directly triggered by large-scale ascent). I think it is important that this distinction is clearly articulated in your paper. As such you need to modify the text in several places, including L56-59, L425, L433-434, L460-462, and L584.

AC: We followed this recommendation, delete the word "trigger" and modified all sentence in the lines mentioned. For example, we explained in Sect. 5 that large-scale lifting provides weak yet persistent ascent, which serve to precondition the thermodynamic environment via adiabatic cooling and thus increase in CAPE and reduction in CIN.

L27-34: You should restructure this part of the paragraph so that the three ingredients for deep moist convection are listed together. State the ingredients first and then discuss the scale of the processes they are associated with (synoptic for instability and moisture; mesoscale to storm scale for the lifting mechanism).

AC: We changed this as desired. "In general, the development of convective storms results from scale interactions of different processes in the atmosphere. It is well known that deep moist convection depends on three necessary but not sufficient ingredients: (i) convective instability over a layer of sufficient depth, (ii) sufficient moisture in the lower troposphere and (iii) a suitable lifting mechanism for the triggering of convection. The first two requirements are usually controlled by processes on the synoptic scale. The latter one can occur at different scale ranges."

L31: Latent and conditional instability are one and the same (see, for example, http://glossary.ametsoc.org/wiki/Latent_instability). Also, potential instability is generally not considered to be a major factor in the preconditioning of convective environments (see section 3.1.3 of Markowski and Richardson 2010). There are also various other forms of instability (centrifugal, inertial, symmetric, shear). As such I would just state conditional instability as the first of the three ingredients for deep, moist convection.

AC: We changed the sentence by deleting the parenthesis with "conditional, latent, potential" to "(i) convective instability over a layer of sufficient depth and (ii)…"

L62: It should be "fully" not "full" here. However, I would actually recommend deleting this sentence as it is a bit "hand wavy".
AC: That's right. Thanks for the careful reading. However, as suggested, we now deleted the sentence.

L78: I would say "mesoscale cut-off lows and PV filaments".
AC: We deleted one meso-scale.

L82: Get rid of "and their accompanying phenomena"
AC: We deleted this part.

L84: "prior to"
AC: Thanks for the careful reading.

L87: Get rid of "The next" and add "then" after "Section 4" (i.e. "Section 4 then puts the results in a historical context")
AC: We changed this.

L94: Rather than "secondary effects" I would say "associated hazards".
AC: We changed this.

L103: You need to say "allow us to investigate" or, alternatively, "permit/facilitate an investigation of".
AC: Thanks for the careful reading.

L126: I think it should just be "Météo-France", not "the Météo-France".
AC: Thanks for the careful reading.

L128-129: Here and elsewhere I would use the term "1-hour extreme rainfall events" rather than "hourly extreme rainfall events". Generally, I would take "hourly" to mean "occurring every hour" rather than "lasting for 1 hour". This is also consistent with "3-hour extreme rainfall events".
AC: We added this suggestion.

L135: Since "the RR collective" isn't mentioned again, you can get rid of the statement in parentheses.
AC: We changed this.

L147: "location and scale parameters, respectively"
AC: We added this suggestion.

L150: I think you mean "standard deviation" not "derivation"
AC: Thanks for the careful reading.

L152: You can either just say "return period" here or use the symbol t RP; you don't need both as this definition is already given in the previous sentence.
AC: We added this suggestion.

L158: Suggest changing to "12 equidistant vertical levels extending from 1 km to 12 km above ground level (AGL)".
AC: We added this suggestion.

L169-170: I'm not sure it is fair to assume weaker cells "cannot move at higher speeds" than those above 55 dBZ. I would instead simply note the caveat that your use of a high reflectivity threshold means that the resulting storm-motion estimates may not be representative of weaker convective cells.
AC: We modified the sentence: "Even if weaker cells are not detected using the 55 dBZ thresholds, it can be assumed (cf. Video Supplement for two representative days) that they cannot move with higher speeds."

L189-190: Suggest modifying the end of this sentence as follows: "...to describe the large-scale meteorological conditions and define weather regimes (see Sect. 2.3), perform kinematic backward trajectories (see Sect. 2.4), and identify cut-off lows (see Sect. 2.5)."
AC: We added the suggestion.

L193: Here and throughout your analysis you should say "bulk wind difference (BWD)" rather than "bulk wind shear". Shear has units of s$^{-1}$ as it is the BWD divided by the layer depth.

AC: Basically this is right. But in the literature, such as in Weisman and Klemp (1982), Bunkers (2002), Markowsky and Richardson (2010), or Trapp (2013), the wind difference is termed to as "vertical wind shear", even though it does not have units of shear.

Bunkers, M. J. (2002): Vertical wind shear associated with left-moving supercells. Weather Forecast., 17, 845-855, https://doi.org/10.1175/1520-0434(2002)017<0845:VWSAWL>2.0.CO;2.

L198: Rather than using Z500' for 500 hPa geopotential height anomalies, I suggest using Z500 to represent 500 hPa geopotential height and explicitly stating when you are talking about an anomaly. For example, on L206 you would say "dominated by a negative Z500 anomaly".

We changed this as desired.

L199: Why did you choose the first seven EOFs? What percentage of the total variance do they collectively explain?

AC: Following other work, e.g. Ferranti et al. 2015, we opted for the minimum number of EOFS explaining more than 75 % of the total variance, which for the normalised Z500 was seven and the total variance explained then is 76,7 %. See for details Grams et al. (2017).

L217-218: Presumably, the five "surrounding" grid points are the nearest grid point to the sounding site and its immediate neighbours to the north, south, east and west; however, this should be stated explicitly.

AC: We added the suggestion.

L220: Get rid of "where the air masses relevant for the thunderstorm development are located". Air below 950 hPa and above 600 hPa is certainly relevant for thunderstorms!

AC: We deleted this part.

L224: Rather than "the literature" I would say "previous studies" (or work or research).

AC: We added the suggestion.

L245-257: This description of the persistence analysis is quite difficult to follow, particularly the first paragraph. As such I would recommend completely rewriting it. Also, as stated in my original review, you should avoid using the term "cluster" here (and in Fig. 15) to avoid confusion with the actual cluster analysis used to define weather regimes.

AC: We revised the paragraph and hope that the method is now clearer and easier to understand. Furthermore, we also reduced the word cluster even more.

L257: What do you mean by "the maximum of the daily minima"? Please rephrase.

AC: We deleted the sentence, since the thresholds are defined and used (as already described) in PIP16. And the interested reader is referred to the study.

L268: Get rid of "an area" before "twice the size of Germany"

AC: We added the suggestion.

L275: I think you mean "evolution" rather than "evaluation" here.

AC: Thanks for the careful reading.

L285: What does "(radar visibility)" indicate? Are you saying that the cells were only visible on radar for 30 minutes? Please explain or delete this if it isn't important.

AC: It is important to mention because the radar visibility is shorter than the lifetime of the cell; we changed "(radar visibility, i.e., period of precipitation)"

L290: As stated in my original review, you should avoid using parentheses to save space as the resulting sentences are much more difficult to read and comprehend.

AC: We changed this as desired.

L312: This sentence has some grammatical errors. I suggest revising as follows: "However, this rain fell in a period of 3 hours, with 60 mm falling in just 50 min."
AC: Thanks for the careful reading. We changed this as suggested.

L320: Put "indicating wind speeds between 25 and 31 m/s" in parentheses.
AC: We changed this as desired.

L324-327: This sentence would also benefit from rewording. Something like "In a few cases, deep-layer shear magnitudes were sufficient (BWD up to 20 m/s) for the development of severe storms, with large hail up to 5 cm in diameter recorded in Southwest France on 26 and 9 June and in southern Germany on 11 June."
AC: Thanks for the suggestion; we changed the sentence.

L359: "(Fig. 6a)"
AC: We added this.

L367: Why introduce the abbreviations "ZO" and "EuBL" here if you aren't going to use them in the text?
AC: To make the connection to the figure quicker and easier.

L379: I would say "a pronounced decrease in convective activity"
AC: We changed this as desired.

L385: Get rid of the first instance of "values"
AC: We changed this.

L396: Since your analysis considers geopotential height on constant pressure surfaces you should say "weak geopotential height gradients".
AC: We changed this as desired.

L403: One way to highlight the strong relationship between V500 and BWD would be to compute the correlation coefficient between the two. You could do this both for the sounding data and the ECMWF analysis over the domain shown in Fig. 8. Just a thought.
AC: This is a nice thought; however, the two figures already indicate similar conditions and the key message will not change.

L405: Get rid of "squall lines" (MCS covers this).
AC: We deleted this part.

L407: Not sure what you mean by deep-layer shear. In my experience this is another name for the 0-6 km (or surface-500 hPa) BWD. As noted above, shear has units of $s^{-1}$, not m $s^{-1}$.
AC: We modified the sentence. BWS represents the directional shear; here, we mean the speed shear (difference between two wind speed values). Regarding the units see answer above.

L409-422: I recommend using the term "air parcels" rather than "air masses" in this section, since the former is more consistent with what a back trajectory represents.
AC: Basically this is right. But, when we are talking in the section about "air mass", we already mean/associate the whole.

L469-470: Suggest rewording this sentence as follows: "This analysis is restricted to Germany due to the availability of long-term (> 50 years), high-resolution (1 km 2) gridded rainfall data."
AC: We changed this as suggested.

L479: What do you mean by "partly with new all-year records"? Maybe rephrase this.
AC: We deleted the sentence.

L480-481: Suggest rewording this sentence and connecting it with the next one (getting rid of the paragraph break in the process) as follows: "This does not appear to be an artefact of insufficient gauge density, as most events are represented by multiple gauges (not shown). Instead, it likely reflects the very slow propagation of storms…"

AC: We changed this as suggested.

L493-494: Suggest getting rid of (or moving) the sentence beginning "Recall that" as it breaks up the flow between the preceding and following sentences. Also, in the next sentence, suggest changing "In doing so" to "Thus".

AC: We moved the sentence and added the second suggestion.

L504: Here and elsewhere in this section, change "event persistence(s)" to "CE duration".

AC: We added the suggestion.

L508-509: Change "event persistences of CE with long duration" to "long-duration CEs".
You might consider completely rewording this sentence as follows: "To put these numbers in context, Fig. 15 shows the relative frequency of CEs in May/June as a function of their duration for the period 1981 to 2010."

AC: We changed the formulation as suggested.

L510-515: This additional explanation of the procedure is confusing and unnecessary. I recommend getting rid of it and the subsequent paragraph break.

AC: This addition was made at the request of the second reviewer after the first draft. It is important to note how the relative frequency is calculated in case someone wants to repeat the procedure.

L541: Change "convective" to "convection"

AC: Thanks for the careful reading.

L581: Change "due to several reason" to "in several respects"

AC: We added the suggestion.

L611: I'm not sure what you mean by "(e.g. jet stream)". Please either elaborate or delete this.

AC: We deleted this comment.

Table 1: I don't think the track length and area are particularly informative, so these can probably be removed. The information on rainfall intensity and storm speed are more useful, but you don't actually discuss them anywhere in the text.

AC: We have already mentioned the information on rainfall intensity, track length, and propagation speed in the last draft in connection with the discussion of the station Bruchweiler (Vers 2: Line 311–314; now Line 311–316) and the station Dietenhofen (a few lines later).
Now, we deleted the total track area in Table 1. Concerning the track length, we added a further comment in conjunction with Figure 4.

Figure 2: Rather than saying "the extended study period" I would give the dates explicitly (i.e. 1 May to 20 June).

AC: We added the information.

Figure 3: What do you mean by "total maximum"?
AC: We deleted "total". We mean the maximum during the study period (as already described).
Also, as stated in my original review you should say "accumulation" rather than "sum" when referring to rainfall amounts in mm.
AC: We changed this.

Figure 4: I don't think it is necessary or appropriate to apply a spline filter to the distributions here. Just plot the raw data as histograms (c.f. Fig. 11).
AC: We plotted the raw data in Figure 4 without spline interpolation.

Figure 6: For some reason your color bars have more ticks than colours. Also the number of ticks per color varies and the tick labels don't always align with changes in the color level. Please make it so that the ticks and labels occur at the boundaries between the colors; otherwise it is difficult for the reader to extract quantitative information from the figure. The caption for this figure also needs revising for clarity. Here is my suggestions: "Mean anomalies during May/June 2018 of (a) 500 geopotential height anomaly (shaded in gpm) and (b) integrated water vapour anomaly (shaded in kg m$^{-2}$), together with the mean 500 hPa geopotential height (contours every 40 gpm). Data are from ERA-Interim and anomalies are computed with respect to the 1981-2010 climatology."

AC: We corrected the figure legend and revised the caption according the suggestion.

Figure 7: Panel (a) needs to be better explained in the caption. In particular, you should state the meaning of the bold sections of the curves and the colours along the x axis. Also, in panel (c) the y axis should be labelled as "V500 (m/s)".

AC: We revised the caption, which now explains clearer the meaning of weather regime life cycles.

Figure 8: Add "at 12 UTC" before "averaged over the study period" and get rid of "12 UTC" from the parentheses at the end.

AC: We changed this.

Figure 9: Please explain either in the caption or the main text how the ellipses were defined. Also, I suggest using "distance along trajectory" instead of "total distance".

AC: We included such an explanation in the caption of Figure 9. The axis label in panel (b) has been changed accordingly.

Figure 13: Change "Return periods" to "Return period" (singular).

AC: We changed this.

Figure 14: I think you mean "2nd and 3rd quartiles" (not "1st and 3rd").

AC: No; we mean the 1$^{st}$ and 3$^{rd}$ quartiles. 1$^{st}$ (lower) = 25 %; 2$^{nd}$ = 50 % = Median; 3$^{rd}$ (upper) = 75 %.

Also, I would recommend making the whiskers and outliers the same colour as the boxes and using solid rather than dashed lines for the whiskers.

AC: The software that was used to produce the figure unfortunately does not support this.

Figure 15: As noted above (and in my original review) you should avoid using the term "cluster" in this analysis. Instead I recommend using "CE duration" (this should also replace "event persistence").

AC: We adapted this.

Also, it should be "May", not "Mai".

AC: Thanks for the careful reading

Figure 16: I believe this information could be better presented in Fig. 10, by replacing the Z500 contours (which are already shown in Fig. 6) with the climatological cut-off low frequency. This way the reader can directly compare the climatological and 2018 cut-off low frequencies without the need to consider anomalies of percentages. You can then get rid of Fig. 16.

AC: In addition to a parallel presentation of Z500 and the cut-off frequency (Fig. 10) for easier comparison, we also want to show the cut-off anomaly (study period vs. climatology; even if it looks quite similar); however, we combined Figure 10 and 16 into Figure 10a and 10b, so that the aspect of comparability is better considered.

**#2 Review for WCD-2020-1 Revision 2 (RC2 from 05 June 2020)**

The paper describes an extraordinary thunderstorm episode of western and central Europe in May/June 2018. The situation was related to an unusually high cut-off low activity at the upstream side of a quasi-stationary blocking. The authors use multiple data sets to study the situation. The paper improved highly from the first version: It is now an easy, nice read. The paper has a very good, easy-to-follow structure. An additional trajectory analysis gives insight to the quasi-stationarity of the situation. An interesting outcome is the potential linkage between large-scale weather patterns (European weather regimes) and thunderstorm activity which should be studied in more detail in the future.
AC: Thanks. The latter point is already the plan for further investigations :)

Abstract: I like the abstract. I think an additional last sentence would be nice that describes the benefits and aim of the paper. Something like you write in line 76-80 in your introduction:
"The primary objective of this paper is to examine the conditions and processes that made this particular thunderstorm episode in 2018 unique. We focus on the process interaction across scales, i.e., from the large-scale dynamics such as atmospheric blocking to meso-scale PV cut-off lows and/or small meso-scale PV filaments to modifications of the convective environment to local-scale thunderstorm occurrences. Further objectives are to highlight the synoptic setting during the thunderstorm episode, to demonstrate the severity of the events, and to place the event in a historical context."
AC: Thanks. We added one sentence about the objectives.

**More specific comments:**
2. Data and Method: This section improved really much. It is now very clear what kind of data you use for what purpose. And it gives a clear and concise overview. Thanks for taking into account the reviewer's comments!
AC: There were also good comments.

Section 3.1.1: I like the connection the authors made between regime changes and convective activity. This would be interesting for a future climatology!
AC: That is already the plan for further investigations :)

Section 3.1.2: Thanks for adding a comparison with bulk wind shear. The comparison nicely shows that V500 is almost identical to BWS over land in this situation. Overall the unusual situation seems to be clearer discussed (or plotted)
AC: Thanks.

Section 3.2: Thanks for adding the trajectory analysis. It is a nice addition. Especially the length of track vs. ellipse size
AC: Thanks.

Section 3.3: This is probably my only real CRITIQUE here: You discuss cut-off low C3 being responsible for the thunderstorm activity over southeast Germany, central France and the Netherlands. However, from your figure 5e it rather seems to be cut-off low C1. The cut-off low detection method seems to join the several smaller cut-offs or filaments. However, the center of C3 is still over the Atlantic Ocean? I would at least write a clear comment here or rename the identified cut-off low blob in Fig 12. In my opinion, C3 is just embedded in a larger scale system - and so is C1 (which seems to be rather associated with the thunderstorms).
AC: The reviewer is absolutely right; the way we wrote it, it sounds like C3 was responsible for the thunderstorm activity over southeast Germany, central France and the Netherlands (but we only meant that the trough arrived from the Atlantic with C3). However, C1 is responsible for this. We modified the text in Sect. 3.3 for clarification and added a reference in section 3.1 in the context of C1 (line 349).
Thanks for the careful reading!

5. Discussion: lines 540-541: "Interestingly, atmospheric blocking was key to providing the large-scale setting conducive for convective in its vicinity." - There seems to be a word missing after convective?

AC: We mean "convection". Thanks for the careful reading.

6. Summary and Conclusions: line 584: "(iii) the large cut-off low frequency that was responsible for the majority of convection triggering" - I would rather say "associated with" instead of "was responsible for", because: Could you really prove that the cut-off lows were responsible?

AC: That's right. We changed this. Thanks for the suggestion.

Fig. 5: I like the blue dots! It would be good to comment on the dots in the main text as well!

AC: We added some words in Sect. 3.1. (line 352)

Overall, the quality of the figures is good.

**The role of large-scale dynamics in an exceptional sequence of severe thunderstorms in Europe May/June 2018**

Susanna Mohr[1], Jannik Wilhelm[1], Jan Wandel[1], Michael Kunz[1,2], Raphael Portmann[3], Heinz Jürgen Punge[1], Manuel Schmidberger[1], Julian F. Quinting[1], and Christian M. Grams[1]

[1]Karlsruhe Institute of Technology (KIT), Institute of Meteorology and Climate Research (IMK), Karlsruhe, Germany
[2]Center for Disaster Management and Risk Reduction Technology (CEDIM), Karlsruhe, Germany
[3]Institute for Atmospheric and Climate Science, ETH Zurich, Switzerland

**Correspondence:** Susanna Mohr (mohr@kit.edu)

**Abstract.** Over three weeks in May and June 2018, an exceptionally large number of thunderstorms hit vast parts of western
and central Europe, causing precipitation of up to 80 mm within one hour and several flash floods. **This study examines**
**the conditions and processes that made this particular thunderstorm episode exceptional. Besides a description of the**
**synoptic setting and the severity of the convective hazards, it is shown how processes interact across scales, from large-**
**scale dynamics with atmospheric blocking to meso-scale cut-off lows to regional convective environment to local-scale**
**thunderstorm occurrences.**

[revised manuscript text omitted]

---

## Author Response (AR3)

**Co-Editor Decision: Publish subject to technical corrections** (03 Jul 2020) by Martin Singh

Comments to the Author:
The authors have implemented the reviewers suggestions satisfactorily, and I am pleased to recommend this manuscript for publication.

Non-public comments to the Author:
There are a number of small writing errors in the current version of the manuscript. I detail these below, along with suggested replacement text. I would ask the authors to consider these changes to the manuscript before publication. Further, I would ask the authors to give the manuscript a careful read to ensure that these errors are minimised before publication.

Many thanks to the co-editor for the very useful suggestions and the careful reading of the last version. We made corrections to all of them and read the manuscript again carefully.

Line 2: causing precipitation of up to 80mm -> causing precipitation accumulations of up to 80 mm

Line 3-6: Suggest shortening these two sentences into one:
This study examines the conditions and processes that made this particular thunderstorm episode exceptional, with a particular focus on the interaction of processes across scales.

Line 60: to release of Convective Available Potential Energy -> to the release of convective available potential energy.

Line 63: important mechanisms for convection -> important mechanism for producing convection

Line 68: But in peripheral locations upstream and downstream of the blocks can also create environmental conditions conducive for deep moist convection development.
-> But blocking can also create environmental conditions conducive for deep moist convection development in peripheral locations upstream and downstream of the block itself.

Line 88: accompanied weather regimes -> accompanying weather regimes

Line 89: PV cut-off -> PV cut-offs

Line 97: associated hazard -> associated hazards

Line 102: Observation data -> Observational data

Line 103: Observation data ->Observational data

Line 104: for a complete thunderstorm detection, but does not discern according to severity ->
for complete thunderstorm detection, but this data does not discern according to severity

Line 107: REGNIE has not yet been defined

Line 147: i.e. -> e.g.,

Line 162: extending from 1 km to 12 km above ground level. For the whole period between 2005 and 2018, which is used to relate the storm motions computed for the investigation period to the climatology (Sect. 4.1), data were stored in six reflectivity classes only.

-> extending from 1 km to 12 km above ground level for the whole period between 2005 and 2018. This data is used to relate the storm motions computed for the investigation period to the climatology (Sect. 4.1). Data were stored in six reflectivity classes only.

Line 173: I don't think this can be assumed, but perhaps you can say weaker cells are unlikely to move with higher speeds.

Line 187: wind speed in 500 hPa -> wind speed at 500 hPa

Line 196: the bulk wind shear (BWS; directional shear) as wind difference between 10m and 500 hPa -> the bulk wind shear (BWS; directional shear), defined as the wind difference between 10m and 500 hPa

Line 222: moist, low-tropospheric air masses -> moist, lower-tropospheric air masses

Line 251: number of days, on which a certain -> number of days on which a certain (remove comma)

Line 258: which is fulfilled if both conditions apply -> which is fulfilled if the following conditions apply

Line 259: in context with the strict criterion -> in the context of the ``strict criterion'' of PIP16

Line 270: most of them reporting heavy rainfall leading to several -> most of the reports described heavy rainfall, some of these heavy rainfall events lead to

line 281: This highest number -> The highest number

Line 295: Does the "half" here refer to the mean, median or standard deviation?
To all; we added this aspect.

Line 318: A second example is on 31 May the exceptionally high -> A second example is on 31 May; the exceptionally high

Line 323: Especially on the last day of the study period, on 12 June, the proportion of gust reports (indicating wind speeds between 25 and 31ms´1324 ) to all reports was very large.
-> The proportion of gust reports (indicating wind speeds between 25 and 31ms´1324 ) to all reports was very large, especially on the last day of the study period, on 12 June.

Line 408: V500 is almost similar to BWS -> V500 is similar to BWS

Line 411: are relying -> rely

Line 488: It is not clear what "this" is referring to in this sentence.
Here there was a doubling with the sentence before; therefore we have combined the two.

Line 536: a PV cut-off was up -> PV cut-off frequency was up

Line 550: Several studies have identified such a flow to provide convection-favouring conditions
-> Several studies have identified such a flow as providing convection-favouring conditions in this region

Line 565: Or it can generate instability, if an entire column is lifted bodily until complete saturation in case of potential instability

Suggest removing this sentence, as previous sentence is now about CAPE (instability).

line 576: In addition, a high concentration of water vapour at low levels in the presence of strong updrafts, high environmental relative humidity, significant cloud depth below the freezing level contribute to maximize rain accumulations, and potentially weak vertical wind shear, which tend to be correlated with weak mid-tropospheric winds (Markowski and Richardson, 2010). Due to the low propagation speeds, which contributes
-> In addition, a high concentration of water vapour at low levels in the presence of strong updrafts, high environmental relative humidity, significant cloud depth below the freezing level contribute to maximize rain accumulations. Furthermore, weak vertical wind shear, which tends to be correlated with weak mid-tropospheric winds (Markowski and Richardson, 2010), reduces storm propagation speeds. Due to the low propagation speeds, which contribute